# Atomically dispersed Pt–N$_4$ sites as efficient and selective electrocatalysts for the chlorine evolution reaction

Taejung Lim [1,4], Gwan Yeong Jung[1,4], Jae Hyung Kim [1], Sung O Park[1], Jaehyun Park[1], Yong-Tae Kim [2], Seok Ju Kang [1], Hu Young Jeong [3], Sang Kyu Kwak [1*] & Sang Hoon Joo [1*]

Chlorine evolution reaction (CER) is a critical anode reaction in chlor-alkali electrolysis. Although precious metal-based mixed metal oxides (MMOs) have been widely used as CER catalysts, they suffer from the concomitant generation of oxygen during the CER. Herein, we demonstrate that atomically dispersed Pt−N$_4$ sites doped on a carbon nanotube (Pt$_1$/CNT) can catalyse the CER with excellent activity and selectivity. The Pt$_1$/CNT catalyst shows superior CER activity to a Pt nanoparticle-based catalyst and a commercial Ru/Ir-based MMO catalyst. Notably, Pt$_1$/CNT exhibits near 100% CER selectivity even in acidic media, with low Cl$^-$ concentrations (0.1 M), as well as in neutral media, whereas the MMO catalyst shows substantially lower CER selectivity. In situ electrochemical X-ray absorption spectroscopy reveals the direct adsorption of Cl$^-$ on Pt−N$_4$ sites during the CER. Density functional theory calculations suggest the PtN$_4$C$_{12}$ site as the most plausible active site structure for the CER.

[1] Department of Energy Engineering and School of Energy and Chemical Engineering, Ulsan National Institute of Science and Technology (UNIST), 50 UNIST-gil, Ulsan 44919, Republic of Korea. [2] Department of Materials Science and Engineering, Pohang University of Science and Technology (POSTECH), 77 Cheongam-Ro, Pohang, Gyeongbuk 37673, Republic of Korea. [3] UNIST Central Research Facilities, Ulsan National Institute of Science and Technology (UNIST), 50 UNIST-gil, Ulsan 44919, Republic of Korea. [4] These authors contributed equally: Taejung Lim, Gwan Yeong Jung. *email: skkwak@unist.ac.kr; shjoo@unist.ac.kr

Chlorine ($Cl_2$) is one of the most important industrial chemicals with an annual production of approximately 75 million tons worldwide[1]. It is used as a key chemical in the production of polymers and pharmaceuticals, pulp and paper industries, and water treatments[2–4]. The current $Cl_2$ production is prevalently dependent on the chlor-alkali process[4,5], for which the electrochemical chlorine evolution reaction (CER) plays a pivotal role as the anodic reaction[5–8]. To ensure high efficiency in the chlor-alkali system and the production of high purity $Cl_2$ gas, CER should be operated in acidic pH saturated with $Cl^-$ (ref. [5]). The CER is also important for generating active chlorine (AC) as a disinfectant for wastewater and ship ballast water treatments on account of its effectiveness in removing harmful organisms or invasive aquatic species with a long residual time[9–12]. AC generation is typically conducted in neutral pH, where the oxygen evolution reaction (OER), the side reaction of CER, shows much lower overpotential than that in acidic pH[13,14], leading to low CER selectivity. On the potential scale of reversible hydrogen electrode (RHE), the CER and OER occur via the following reactions and their standard reversible electrode potentials ($E^0$).

$$2Cl^- \rightleftharpoons Cl_2 + 2e^-$$

$$E_{CER}^0 = \left(1.358 + \frac{RT}{F} \times 2.303 \times pH\right) V \text{ vs. RHE} \quad (1)$$

$$2H_2O \rightleftharpoons O_2 + 4H^+ + 4e^-$$
$$E_{OER}^0 = 1.229 \, V \text{ vs. RHE} \quad (2)$$

where $R$, $T$ and $F$ represent the universal gas constant, the temperature, and the Faraday constant, respectively. Moreover, under the AC-generating conditions, the $Cl^-$ concentration is below 1 M, which makes the CER thermodynamically and kinetically more challenging than that in the chlor-alkali process[5,14–16].

Mixed metal oxides (MMOs) based on precious metals (Ru and Ir), such as a dimensionally stable anode (DSA), have been predominantly used as CER catalysts irrespective of the pH of the solution[5–8]. However, computational and experimental works revealed that MMO catalysts are also highly active for the OER, exhibiting a scaling relationship between the CER and OER[13,17–20]. This relationship suggests that two reactions are catalysed on a similar active site of the MMOs or form a common surface intermediate species[20–24]. The oxidative water activation and concomitant surface oxidation on MMOs[25–27] are, therefore, unavoidable, leading to a decrease in active site density for the CER[25]. To mitigate $O_2$ generation in the condition of the chlor-alkali process, the contents of precious metals were reduced. Nevertheless, high amounts (approximately 30 at%) were required to maintain sufficient electronic conductivity for the CER[5]. As alternative strategies, doping of other metals into the MMOs[28,29], structural modification of MMOs[30–32], and the use of new compositions[12,33] have been exploited to promote activity as well as selectivity for the CER. However, MMOs still exhibit low CER selectivity at low $Cl^-$ concentration and neutral pH[5,12,15].

Atomically dispersed catalysts[34,35] or single-atom catalysts[36] have recently been actively pursued to maximise the utilisation efficiency of precious metals. The atomically dispersed catalysts often give rise to a different reaction pathway compared to that of widely used nanoparticle-based catalysts, leading to unusual selectivity and activity for many electrocatalytic reactions, including the oxygen reduction reaction, hydrogen evolution reaction and fuel oxidation reactions[37–40]. However, to the best of our knowledge, the atomically dispersed catalysts have never been exploited as an electrocatalyst for the CER, and only homogeneous electrocatalysts for generating $Cl_2$ (refs. [41,42]) or $ClO_2$ (refs. [43,44]) have sporadically been reported.

Here, we demonstrate that an electrocatalyst of atomically dispersed Pt–$N_4$ sites doped on carbon nanotube ($Pt_1$/CNT) is capable of catalysing CER with excellent activity and selectivity. The $Pt_1$/CNT catalyst exhibits superior CER activity to Pt nanoparticles on CNT (PtNP/CNT) and commercial Ru/Ir-based DSA catalysts in acidic media. Notably, $Pt_1$/CNT exhibits near 100% CER selectivity in acidic media, with $Cl^-$ concentration as low as 0.1 M, as well as in neutral media, whereas DSA shows substantially lower selectivity. In situ electrochemical X-ray absorption spectroscopy (XAS) reveals the direct interaction between $Cl^-$ reactant and Pt centre of Pt–$N_4$ sites during the CER. Density functional theory (DFT) calculations identify the $PtN_4C_{12}$ structure as the most plausible active site structure with the lowest Gibbs free energy for the CER among the possible structural configurations for the Pt–$N_4$ sites. The atomically dispersed Pt catalyst comprising Pt–$N_4$ sites reported herein may broaden the scope of CER catalysts beyond the hitherto-dominated precious metal-based MMOs, with maximised precious metal atom utilisation for the CER.

## Results

**Preparation and characterisations of catalysts.** The preparation of atomically dispersed Pt catalysts consists of mixing a Pt precursor (Pt(II) meso-tetraphenylporphine, PtTPP) with acid-treated CNT (Supplementary Fig. 1a), followed by heat treatment at desired temperatures between 500 °C and 800 °C. The resulting samples were denoted as $Pt_1$/CNT_$X$ ($X$ = annealing temperature). Among the prepared $Pt_1$/CNT_$X$ samples, the best CER activity was obtained with the sample treated at 700 °C, which is hereafter denoted as $Pt_1$/CNT. Inductively coupled plasma optical emission spectroscopy (ICP-OES) and combustion elemental analysis (EA) revealed that $Pt_1$/CNT comprised 2.7 wt% Pt and 0.7 wt% N (N/Pt atomic ratio = 3.6, Supplementary Table 1). The high-angle annular dark-field scanning transmission electron microscopy (HAADF-STEM; Fig. 1a and Supplementary Fig. 1b) images show uniformly distributed ultrasmall white dots, suggesting formation of atomically dispersed Pt species without Pt NPs. The particle size distribution histogram of $Pt_1$/CNT (Supplementary Fig. 1c) confirmed the atomic dispersion of Pt species. The electron energy loss spectrum (EELS) of $Pt_1$/CNT, obtained from a limited area (~5 Å$^2$) comprising an atomically dispersed Pt site in the HAADF-STEM image (Supplementary Fig. 2), indicated the presence of Pt and N. For comparison, the PtNP/CNT catalyst with 2.9 wt% Pt loading (Supplementary Table 1) was prepared by the conventional impregnation-reduction method using $H_2PtCl_6\cdot6H_2O$ as a precursor. The high-resolution TEM (Supplementary Fig. 1d) and HAADF-STEM images (Supplementary Fig. 1e) and particle size distribution histogram (Supplementary Fig. 1f) of PtNP/CNT revealed the formation of Pt NPs with a diameter of 1–6 nm. X-ray diffraction (XRD) results (Supplementary Fig. 3) also suggested the absence of metallic Pt species in $Pt_1$/CNT and the formation of Pt NPs in PtNP/CNT.

The detailed geometric structure around the Pt atoms was obtained by extended X-ray absorption fine structure (EXAFS) analyses. The $k^3$-weighted Pt $L_3$-edge EXAFS spectrum of $Pt_1$/CNT and its fitted curve (Fig. 1b and Supplementary Fig. 4) exhibited a major peak at 1.6 Å, which can be assigned to Pt–N/C bonding with a coordination number (CN) of 3.9 (Supplementary Table 2). A second peak at 2.6 Å could be interpreted as the second shell of Pt⋯C by way of N. The EXAFS spectrum of $Pt_1$/CNT remarkably resembles that of PtTPP, indicating that the local structure of PtTPP was nearly preserved in $Pt_1$/CNT even after the high-temperature treatment. In contrast, the EXAFS

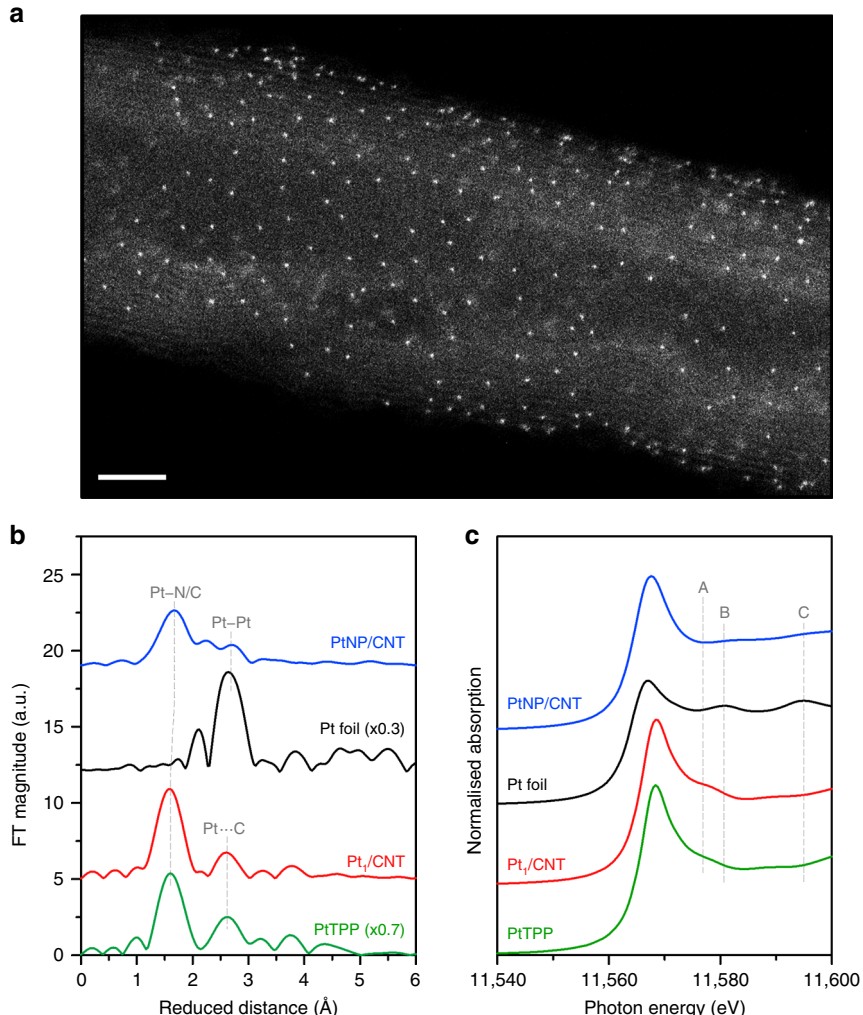

**Fig. 1 Characterisation of Pt₁/CNT catalyst comprising atomically dispersed Pt−N₄ sites on the surface of CNT. a** HAADF-STEM image of Pt₁/CNT catalyst. Scale bar: 3 nm. **b** $k^3$-weighted Pt L₃-edge EXAFS spectra and **c** XANES spectra of Pt₁/CNT and PtNP/CNT catalysts, along with PtTPP precursor and Pt foil reference.

spectrum of PtNP/CNT showed peaks for Pt–Pt and Pt–C bonds, which correspond to metallic Pt and Pt–CNT interaction, respectively. X-ray absorption near-edge structure (XANES) analyses further corroborated the EXAFS results. The Pt L₃-edge XANES spectra of Pt₁/CNT and PtTPP (Fig. 1c) similarly showed a peak at 11,577.0 eV (peak A), which is a spectral feature of square planar N₄-coordinated Pt complexes, such as Pt(II) phthalocyanine[45] and tetraamine Pt(II)[46]. For the XANES spectra of PtNP/CNT and Pt foil, peaks at 11,580.5 eV (peak B) and 11,595.0 eV (peak C) were observed, which are characteristic peaks for metallic Pt[46]. Pt 4f X-ray photoelectron spectroscopy (XPS) analyses (Supplementary Fig. 5) revealed that Pt₁/CNT consisted almost entirely of $Pt^{2+}$ species whereas $Pt^0$ was a predominant species in PtNP/CNT with some contributions from $Pt^{2+}$ and $Pt^{4+}$ species.

Next, the impact of annealing temperatures on the physico-chemical properties of Pt₁/CNT_X catalysts were investigated. The HAADF-STEM images of Pt₁/CNT_X (X = 500, 600 and 800 °C) (Supplementary Fig. 6) clearly indicated the formation of atomically dispersed Pt sites irrespective of annealing temperatures, suggesting that annealing temperatures up to 800 °C cannot induce agglomeration of the Pt. XRD patterns of Pt₁/CNT_X catalysts (Supplementary Fig. 7) suggested the absence of Pt NPs even the annealing temperature was as high

as 800 °C. Pt₁/CNT_X catalysts also showed similar EXAFS and XANES spectra (Supplementary Fig. 8) without annealing temperature-dependent peak shifts. Notably, these spectra resemble those of a physical mixture of PtTPP and CNT, indicating that the local structure of PtTPP was preserved after heat treatments up to 800 °C. Deconvoluted N 1s XPS spectra of the series of Pt₁/CNT_X samples (Supplementary Fig. 9a) indicated that a large portion of N atoms maintained its chemical state of the PtTPP precursor with a peak at 399.2 eV (Pt−$N_x$ coordination) during annealing at high temperatures. With the samples annealed above 600 °C, peaks for pyridinic N (398.1 eV), pyrrolic N (400.2 eV) and graphitic N (400.7 eV) species emerged, which could be ascribed to the decomposition of surrounding carbons following the van Veen model[47]. We note that the position of $Pt^{2+}$-ligated N species was clearly different from those of pyridinic N and pyrrolic N species, which is consistent with the XPS results of heat-treated M−N/C catalysts comprising atomically dispersed Fe−$N_x$ or Co−$N_x$ sites[47–50]. Deconvoluted Pt 4f spectra (Supplementary Fig. 9b) revealed that Pt₁/CNT consisted of $Pt^{2+}$ species irrespective of the annealing tempera-tures, which are in accordance with the EXAFS results. Taken together, TEM, XAS and XPS results indicate that the use of Pt-containing macrocyclic compound can yield atomically dispersed Pt catalysts comprising Pt−N₄ sites with high Pt loading, which is

hardly achievable with the widely used impregnation-reduction method using $H_2PtCl_6 \cdot 6H_2O$ as a precursor.

**Electrochemical CER activity and stability of Pt−N4 sites.** We investigated the electrocatalytic performances of the catalysts using a rotating ring-disk electrode (RRDE) setup[51] in 0.1 M $HClO_4$ in the presence and absence of 1.0 M of $Cl^-$. All electrochemical measurements were conducted at room temperature (RT, ~25 °C) unless otherwise specified. In the RHE scale, the equilibrium potential of CER ($E_{CER}$) is dependent on pH, temperature, concentration of $Cl^-$ ions, and partial pressure of $Cl_2$ of the electrolyte. The $E_{CER}$ was derived by the Nernst equation:[22,52]

$$E_{CER}(T, a(Cl_2), a(Cl^-)) \text{ vs. RHE}$$
$$= E^0_{CER} - \frac{RT}{F} \times \ln a(Cl^-) + \frac{RT}{2F} \times \ln a(Cl_2) \qquad (3)$$

where $R$, $T$, $F$ and $a$ represent the universal gas constant, the temperature, the Faraday constant, and the chemical activity, respectively. The value of $a(Cl_2)$ was assumed to be 0.01 for the partial pressure of evolving $Cl_2$ under Ar purging. $a(Cl^-)$ was determined by experimental conditions (i.e., $a(Cl^-) = 1.0$ for 1.0 M NaCl)[52]. The temperature dependence of $E^0_{CER}$ vs. RHE can be calculated from the following Eq. (4)[53].

$$E^0_{CER} = \left(1.358\,V + \frac{RT}{F} \times 2.303 \times pH\right)$$
$$- \left(0.001248\,\frac{dE}{dT}\right) \times (T - 298.15\,K) \qquad (4)$$

Figure 2a displays the CER polarisation curves of the catalysts in the presence of 1.0 M NaCl, which clearly indicate the superior CER activity of $Pt_1/CNT$ compared to CNT, PtNP/CNT and commercial Ru/Ir-based DSA catalysts. $Pt_1/CNT$ started to catalyse the CER at a potential of 1.38 V, which is 30 mV higher than the $E_{CER}$ (1.35 V vs. RHE for 25 °C). $Pt_1/CNT$ delivered a current density of 10 mA cm$^{-2}$ at an overpotential of 50 mV, which is much lower than those of DSA (105 mV) and PtNP/CNT (120 mV). The evolution of $Cl_2$ at this potential could be confirmed by $Cl_2$ reduction on the Pt ring (Supplementary Fig. 10). The CER activity of the catalysts was also assessed by calculating the mass activities and TOFs. At an overpotential of 70 mV, $Pt_1/CNT$ exhibited a mass activity of 1.6 A mg$_{Pt}^{-1}$, which is 6.2 times higher than that of PtNP/CNT. For the calculation of TOFs, all Pt sites in the catalyst layer were considered as active sites for $Pt_1/CNT$ and calculated as 3.46 nmol based on the quantification of Pt by ICP-OES analysis (see Eqs. 8 and 9 in Methods section and Supplementary Table 1). The CO stripping method was used to calculate the electrochemically active surface sites for PtNP/CNT, and the calculated value was 0.96 nmol (see Eqs. 7 and 9 in Methods section and Supplementary Fig. 11). As a result, $Pt_1/CNT$ showed 2.6 times higher TOF than PtNP/CNT at an overpotential of 70 mV (Fig. 2b). The CER activities were also tested on a carbon paper. $Pt_1/CNT$ coated on a carbon paper required an overpotential of 70 mV to reach a current density of 10 mA cm$^{-2}$ (Supplementary Fig. 12a). The intrinsic catalytic activity of electrocatalysts for the CER was also assessed in terms of exchange current density ($j_0$) using Tafel analyses (see Eq. 10 in Methods section). The $j_0$ of $Pt_1/CNT$ was 0.43 mA cm$^{-2}$, whereas those of PtNP/CNT and DSA were 0.23 mA cm$^{-2}$ and 0.20 mA cm$^{-2}$, respectively. $Pt_1/CNT$ loaded on a carbon paper exhibited $j_0$ of 0.44 mA cm$^{-2}$ similar to the value obtained on a RRDE, suggesting that $j_0$ was invariant on the type of electrode substrates. Because the practical chlor-alkali process is operated at temperatures of 80–90 °C, we also measured the CER activity of $Pt_1/CNT$ and DSA catalysts at 80 °C, which indicated a similar activity trend as that obtained at 25 °C

(Supplementary Fig. 13). Significantly, a comparison of CER activity data (Supplementary Table 3) reveals that $Pt_1/CNT$ outperforms previously reported DSA electrodes[25,30–33]. The CER activities of $Pt_1/CNT$ on a carbon paper were also compared in neutral pH conditions, 0.55 M NaCl and natural seawater. $Pt_1/CNT$ showed slightly better or similar activity in comparison to DSA in both 0.55 M NaCl solution and natural seawater (Supplementary Fig. 12b and c). Such a high intrinsic CER activity of $Pt_1/CNT$ is noteworthy given that $Pt_1/CNT$ showed much lower surface roughness than DSA, as observed in their cyclic voltammograms (Supplementary Fig. 14).

The CER activities of $Pt_1/CNT\_X$ catalysts and a mixture of PtTPP and CNT were also compared (Supplementary Fig. 15). It is evident that the PtTPP-CNT mixture showed substantially lower CER activity than the annealed catalysts, and the CER activity of $Pt_1/CNT\_X$ catalysts gradually increased with annealing temperatures. To clarify the annealing temperature-dependent activity trend in $Pt_1/CNT\_X$ catalysts, Nyquist plots were obtained at a potential of 1.4 V (vs. RHE) and fitted with an equivalent circuit to assess the charge transfer resistance ($R_{ct}$, Supplementary Fig. 16). As the annealing temperature increased, smaller semicircles (Supplementary Fig. 16c) and lower $R_{ct}$ of $Pt_1/CNT\_X$ catalysts were obtained (Supplementary Table 4), indicating that more facile charge transfer is achievable for the CER through higher annealing temperature. Overall, the annealing at high temperatures concomitantly enhanced the structural integrity and electrical conductivity between the Pt−N4 sites and CNT[47–50,54,55], while preserving the local structure and chemical states of the Pt−N4 sites.

The stability of $Pt_1/CNT$ was examined by chronoamperometry (CA) at an initial current density of 10 mA cm$^{-2}$ (Fig. 2c). $Pt_1/CNT$ retained 72% of its initial current after 12 h of CER operation, which is similar to that of DSA. After the long-term stability test, the polarisation curve of $Pt_1/CNT$ exhibited 75% of its initial current at 1.41 V vs. RHE (Supplementary Fig. 17). In contrast, PtNP/CNT underwent more severe degradation preserving only 50%, which could originate from the dissolution of Pt, because even a trace amount of $Cl^-$ accelerates the formation of metastable chloro-Pt complexes[56]. In contrast, the Pt−N4 sites in $Pt_1/CNT$ could mitigate the formation of chloro-Pt complexes by their strong ligation with N.

**Electrochemical CER selectivity and kinetics of Pt−N4 sites.** We next assessed the CER selectivity of the catalysts. In the absence of NaCl, $Pt_1/CNT$ delivered virtually no faradaic currents for both disk current ($Cl_2$ evolution) and ring current ($Cl_2$ reduction), suggesting near 100% CER selectivity (Fig. 2a). The high CER selectivity of $Pt_1/CNT$ was confirmed by iodometric titration (Fig. 2d and Supplementary Fig. 18a)[51], which indicated 96.6% selectivity. CER selectivity was also examined by the CA method using RRDE, which further confirmed high CER selectivity at 97.1% (Supplementary Fig. 19a). PtNP/CNT and DSA also showed high CER selectivity of above 95% (Supplementary Figs. 18 and 19). CER selectivity was also assessed at low pH with low $Cl^-$ concentration and neutral pH, where MMOs commonly exhibit low CER selectivity. In acidic pH with a $Cl^-$ concentration of 0.1 M, $Pt_1/CNT$ showed high CER selectivity (96.9%), whereas DSA exhibited only 61.3% CER selectivity, as revealed by iodometric titrations (Fig. 2d and Supplementary Fig. 20). The CA method using RRDE (Supplementary Fig. 21) further confirmed the high CER selectivity of $Pt_1/CNT$ (96.1%). In a neutral solution of 0.55 M NaCl, $Pt_1/CNT$ catalysed the CER with a selectivity of 93.9%, which is substantially higher than that of 82.2% by DSA (Fig. 2d and Supplementary Fig. 22). The high CER selectivity of Pt−N4 sites was also demonstrated with other $Pt_1/CNT\_X$

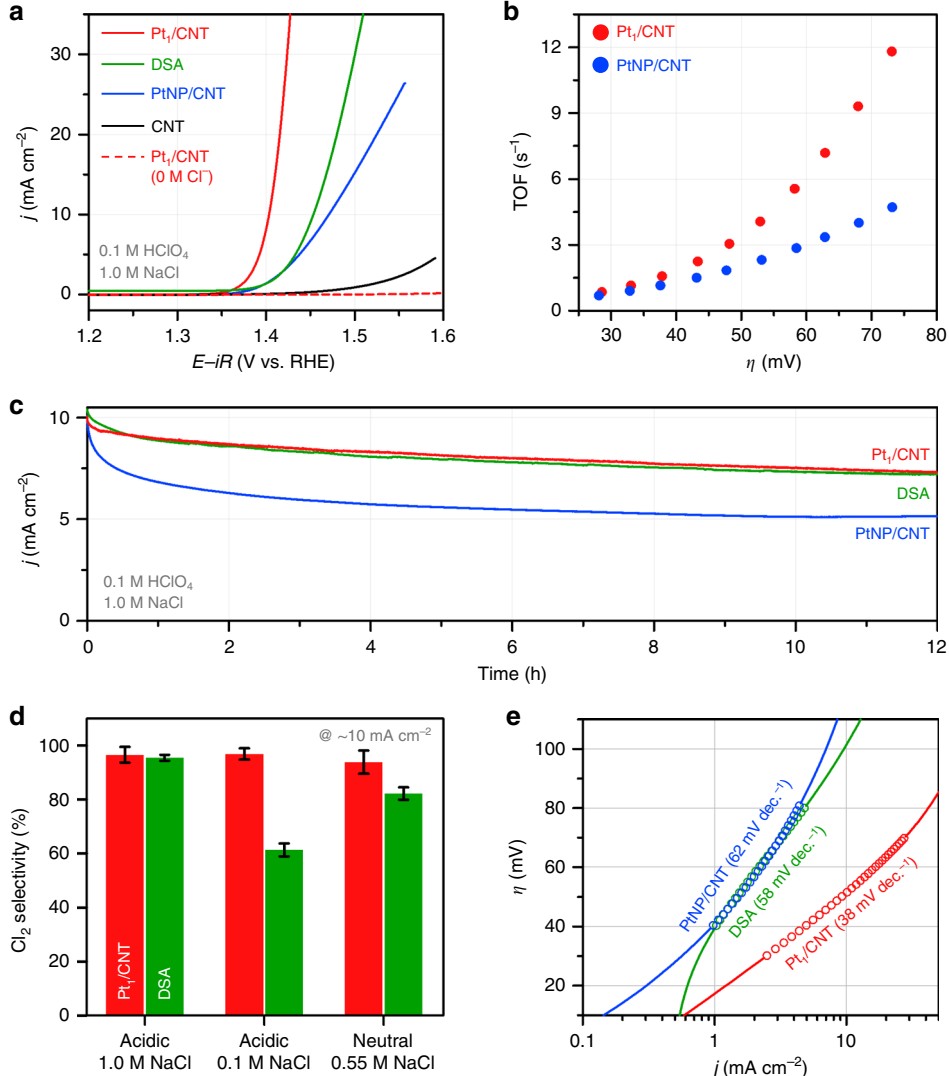

**Fig. 2 CER performance of Pt$_1$/CNT catalyst compared to its NP counterpart and dimensionally stable anode (DSA) catalyst. a** CER polarisation curves of Pt$_1$/CNT, PtNP/CNT, DSA and CNT catalysts obtained in 0.1 M HClO$_4$ + 1.0 M NaCl at an electrode rotation speed of 1600 rpm and a scan rate of 10 mV s$^{-1}$. The polarisation curve of Pt$_1$/CNT catalyst measured in 0.1 M HClO$_4$ is also shown. The DSA catalyst (1 cm × 1 cm) was measured without electrode rotation. **b** Calculated turnover frequencies (TOFs) of Pt$_1$/CNT and PtNP/CNT catalysts corresponding to Fig. 2a. **c** Chronoamperograms of Pt$_1$/CNT and PtNP/CNT catalysts deposited on a carbon paper (1 cm × 1 cm) and DSA catalyst (1 cm × 1 cm) measured in 0.1 M HClO$_4$ + 1.0 M NaCl for 12 h with a stirring speed of 300 rpm. **d** CER selectivity of Pt$_1$/CNT and DSA catalysts measured by iodometric titration under different electrolyte conditions. The error bars indicate the standard deviation of three independent titrations. **e** Tafel plots of Pt$_1$/CNT, PtNP/CNT and DSA catalysts. Their Tafel slopes are denoted in parentheses.

catalysts, suggesting that similarly high CER selectivity was obtained irrespective of the annealing temperatures (Supplementary Fig. 23). The universally high CER selectivity of Pt$_1$/CNT in various pH and Cl$^-$ concentrations indicates that the Pt−N$_4$ sites are less sensitive to water activation, unlike MMOs[5,12,15].

The kinetic information of the catalysts for CER was obtained from Tafel analyses (Fig. 2e). Pt$_1$/CNT showed a Tafel slope of 38 mV dec.$^{-1}$ at an overpotential range of 30–70 mV, whereas those of PtNP/CNT and DSA were higher with 52 mV dec.$^{-1}$ and 60 mV dec.$^{-1}$ in the range of 40–80 mV, respectively. The Tafel analyses suggest that the CER on Pt$_1$/CNT proceeded with faster kinetics than that on PtNP/CNT and DSA. The Tafel slope of 38 mV dec.$^{-1}$ indicates that the CER on Pt$_1$/CNT may proceed via the Volmer-Heyrovsky mechanism[13,14,57,58].

**In situ XANES spectroscopy during the CER.** We conducted in situ electrochemical Pt L$_3$-edge XANES experiments to observe

the interaction between the Pt–N$_4$ sites and Cl$^-$ during the CER. The electrochemical in situ XANES was measured while increasing the applied potentials under CER operation conditions in the absence or presence of NaCl (Fig. 3). Two sets of in situ XANES spectra clearly show different trends. In the absence of NaCl, the XANES spectra of Pt$_1$/CNT changed only marginally with applied potentials (Fig. 3a). However, upon the addition of NaCl, their XANES spectra underwent significant changes in white line intensity, particularly at potentials above 1.40 V where CER takes place (Fig. 3b). The marked increase in white line intensity at 1.40 V and 1.50 V indicates that more Cl$^-$ species are adsorbed on Pt sites for higher Cl$_2$ yields. This distinct difference in the XANES spectra elucidates that Cl$^-$ ions are absorbed on the Pt–N$_4$ sites during the CER.

**Active site identification by DFT calculations.** The origin of excellent CER activity of Pt$_1$/CNT and detailed active site

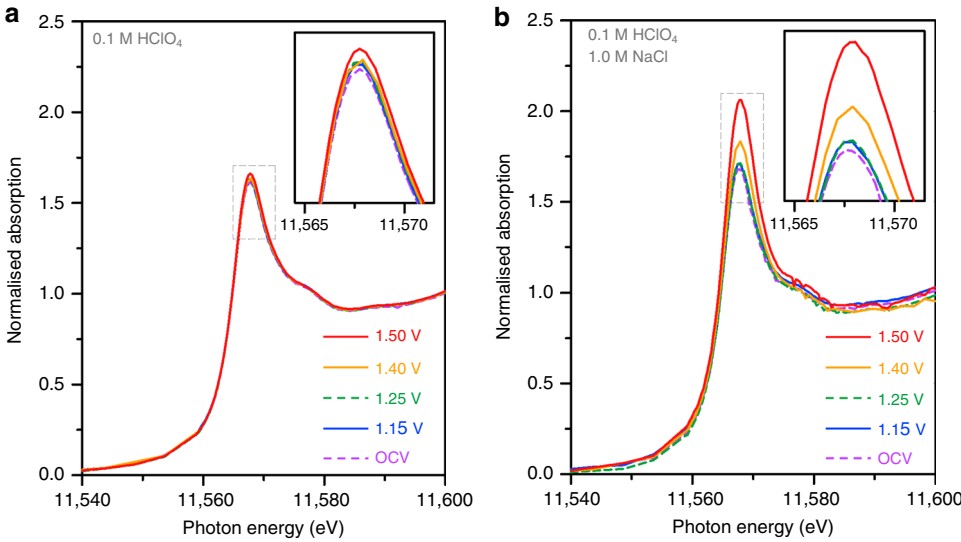

**Fig. 3 In situ electrochemical Pt L$_3$-edge XANES spectra of Pt$_1$/CNT catalyst in the absence and presence of NaCl.** XANES spectra **a** in the absence and **b** in the presence of 0.1 M NaCl taken under sequentially applied potentials without electrode rotation. The insets show a magnified view of the white line regions.

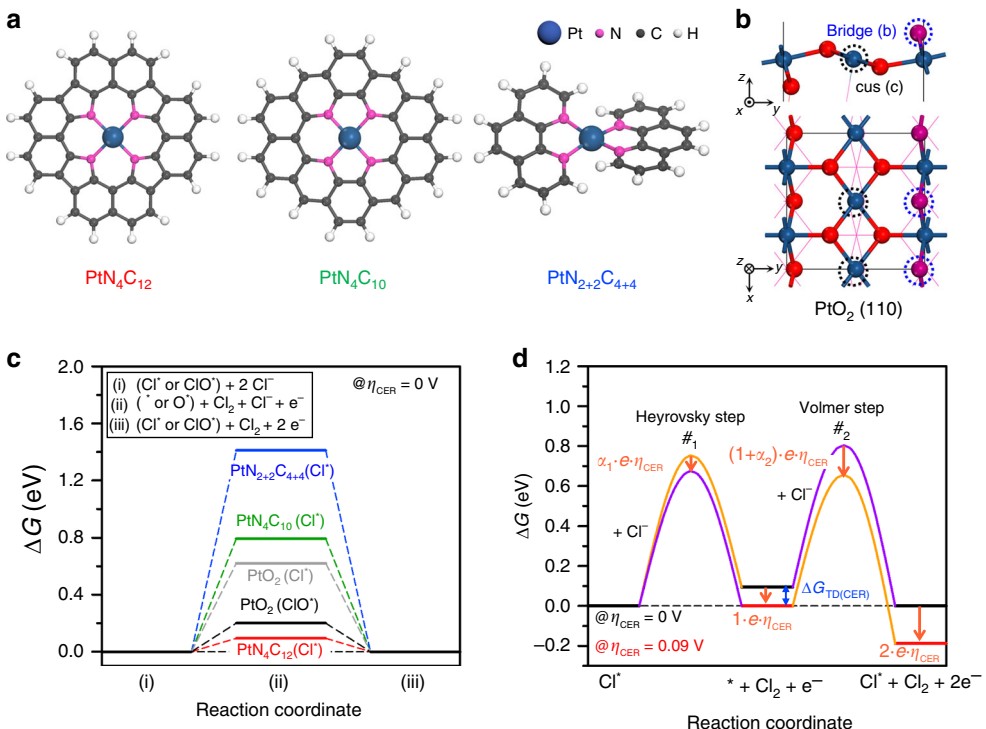

**Fig. 4 DFT calculations of plausible model systems of Pt−N$_4$ sites and PtO$_2$ in CER and OER conditions. a** Model system for possible structural configurations of Pt−N$_4$ sites including PtN$_4$C$_{12}$, PtN$_4$C$_{10}$ and PtN$_{2+2}$C$_{4+4}$ and **b** PtO$_2$ (110) surface. **c** Free energy diagrams for CER over Pt−N$_4$ clusters and PtO$_2$ (110) surface at zero overpotential ($\eta_{CER}$ = 0 V). **d** Full free energy diagram along the reaction coordinate of CER over Pt$_1$/CNT catalyst at the respective overpotential for CER ($\eta_{CER}$) of 0 V (black thick lines) and 0.09 V (red thick lines). $\alpha_1$ and $\alpha_2$ represent the transfer coefficients at each transition state (TS), which are determined as 0.83 and 0.58 from the experimental Tafel plots, respectively. The TS with higher free energy at the respective $\eta_{CER}$ is indicated by purple line. Orange arrows represent the decreased amounts of free energies by applied $\eta_{CER}$ for each state [i.e., first TS (denoted as '#$_1$'), intermediate state, second TS (denoted as '#$_2$'), and final state, respectively]. The free energy change for reaction intermediate at zero overpotential ($\Delta G_{TD}$ $_{(CER)}$) is indicated by blue arrow.

structure were theoretically elucidated by DFT calculations (Supplementary Note 1). We considered three plausible models representing the atomically dispersed Pt−N$_4$ sites (i.e., PtN$_4$C$_{12}$, PtN$_4$C$_{10}$ and PtN$_{2+2}$C$_{4+4}$, Fig. 4a), which were identified as the active sites for CER in Pt$_1$/CNT through the EXAFS and in situ

XANES analyses. We constructed Pourbaix diagrams, which provide the thermodynamically most stable adsorbate structures under applied electrode potential ($U$) and pH, by calculating the adsorption free energies ($\Delta G$'s) of possible adsorbates (i.e., *, ClO*, Cl*, H*, OOH*, O* and OH*) for Pt−N$_4$ sites

(Supplementary Figs. 24–27 and Supplementary Note 2). The Pourbaix diagram was constructed using an ab-initio constrained thermodynamics approach; the reactants were adsorbed on the surface while restricting the subsequent formation of product[59]. Nevertheless, the active adsorbate structure, identified as the thermodynamically most stable structure, can promisingly serve as a starting point for the investigation of mechanistic pathways. CER is pH-independent on the standard hydrogen electrode (SHE) scale and thus appears as a horizontal line at the equilibrium potential of 1.36 V vs. SHE (Supplementary Fig. 26). In contrast, the equilibrium potential for OER exhibits a decreasing trend with a slope of $-59$ mV pH$^{-1}$, starting from $U_{SHE}$ (theoretical SHE potential) $= 1.23$ V and pH $= 0$ (ref. [60]). Considering that the present CER operating condition was acidic for Pt$_1$/CNT (pH $= 1$), Cl$^*$ species were predicted as the most probable adsorbate structures for CER on the Pt–N$_4$ sites.

To model the PtNP/CNT (Supplementary Note 1), we chose the most stable (110) surface of distorted rutile PtO$_2$ (which was found as the thermodynamically most stable phase, Supplementary Fig. 28) as a representing model system for PtNP/CNT (Fig. 4b), following the report that Pt nanoparticles exist in an oxidised form near 1.36 V vs. SHE[61]. For this model, the $\Delta G$'s for all plausible adsorbates and their relevant combinations (i.e., O$_b$ and OH$_b$ at bridge sites and OH$_c$, Cl$_c$, O$_c$, OOH$_c$, O$_{2(c)}$ and ClO$_c$ at coordinatively unsaturated (cus) sites, respectively) were calculated (Supplementary Figs. 29 and 30). The Pourbaix diagrams for the PtO$_2$ (110) surface revealed that both 2O$_b$2Cl$_c$ and 2O$_b$2ClO$_c$ can be active adsorbate structures, especially at the acidic CER operating condition (i.e., $U_{SHE} \approx 1.36$ V, pH $\cong 1$), where they had similar thermodynamic stability by sharing a phase boundary line (Supplementary Fig. 31).

Subsequently, to theoretically evaluate the CER activity with the above-identified active adsorbate structures for Pt–N$_4$ sites and PtO$_2$ (110) surfaces, we calculated the free energy diagrams for CER (Fig. 4c). The thermodynamic overpotential for CER can be defined from $\Delta G$ divided by the elementary charge at zero overpotential (i.e., $\eta_{TD(CER)} = \frac{\Delta G}{e}$), which depends on the reaction intermediates as follows: $\Delta G_{Cl^*}$ for bare structure (*) and Cl$^*$ species, and $\Delta G_{ClO^*} - \Delta G_{O^*}$ for O$^*$ and ClO$^*$ species, respectively (Supplementary Note 3) [62]. Among Pt–N$_4$ sites, PtN$_4$C$_{12}$ was identified as the most plausible structure for the CER owing to its lowest $\eta_{TD(CER)}$ of 0.09 V at zero overpotential. Moreover, the stability against Pt dissolution for the Pt–N$_4$ sites was evaluated based on the electrode potential shift ($\Delta U$), which was gauged with reference to the dissolution on the Pt(111) surface (Supplementary Fig. 32). The positive value of $\Delta U$ implies that the dissolution of Pt on the Pt–N$_4$ sites requires more energetic cost compared to that from Pt(111) surface (Supplementary Note 4). Thus, besides good CER activity, PtN$_4$C$_{12}$ species were found to be thermodynamically more stable against Pt dissolution than others, which further supports that it is feasible for PtN$_4$C$_{12}$ species to exist in Pt$_1$/CNT theoretically. In the case of PtO$_2$ (110) surface (Supplementary Fig. 33), ClO$^*$ species ($\eta_{TD(CER)} = 0.20$ V for 2O$_b$2ClO$_c$) were found to be closer to the thermoneutral state ($\Delta G = 0$) than Cl$^*$ species ($\eta_{TD(CER)} = 0.62$ V for 2O$_b$2Cl$_c$), implying that ClO$^*$ species were identified as the reaction intermediate for the CER on the PtO$_2$ surface of the PtNP/CNT, similar to other precious metal oxides, such as RuO$_2$ and IrO$_2$ (refs. [14,20,58]). In summary, we theoretically demonstrated that the CER activity of Pt$_1$/CNT was superior to that of PtNP/CNT.

By combining the experimental data for kinetics and theoretical data for thermodynamics, we constructed a full free energy diagram along the reaction coordinate of CER over Pt$_1$/CNT (Supplementary Note 5). In this approach, which was recently developed by Exner and co-workers[63,64], the free

energies of the transition states (TS's) were obtained from the experimental Tafel plots, while the free energies of the reaction intermediates were determined from DFT calculations (Fig. 4d and Supplementary Fig. 34). The resulting full free energy diagram of CER over Pt$_1$/CNT revealed that the Heyrovsky step is the first step in the reaction pathway because the PtN$_4$C$_{12}$ species, i.e., active site in Pt$_1$/CNT, already involved the Cl$^*$ species at acidic CER condition (Supplementary Fig. 26a).

$$\text{Heyrovsky step : } Cl^* + Cl^- \rightarrow * + Cl_2 + e^- \quad (5)$$

Subsequently, the Volmer step followed as the second step to close the electrocatalytic cycle, which was identified as the rate determining step (RDS) with higher TS free energy at zero overpotential (i.e., $G_{\#_1} = 0.75$ eV for Heyrovsky step, and $G_{\#_2} = 0.80$ eV for Volmer step, respectively, Fig. 4d).

$$\text{Volmer step : } * + Cl_2 + Cl^- + e^- \rightarrow Cl^* + Cl_2 + 2e^- \quad (6)$$

However, for $\eta_{CER} = 0.09$ V, which corresponded to the thermodynamic optimum of PtN$_4$C$_{12}$ species for the CER (i.e., $\eta_{CER} = \eta_{TD(CER)}$), the Heyrovsky step became the RDS with a slightly higher TS free energy (i.e., $G_{\#_1} = 0.67$ eV for Heyrovsky step, and $G_{\#_2} = 0.66$ eV for Volmer step, shown in Fig. 4d and Supplementary Fig. 35a, respectively). A recent study highlighted that the thermodynamic measure for the activity would be more helpful if it is evaluated at target overpotential ($\eta > 0$), instead of zero overpotential ($\eta = 0$)[65]. Considering that the typical CER overpotentials for the chlor-alkali process are ~0.1 V[5,66], the PtN$_4$C$_{12}$ species in Pt$_1$/CNT could be highly beneficial for industrial chlorine electrocatalysis because they can reach the nearly thermoneutral state at target overpotential (Supplementary Fig. 35b).

To further verify the high CER selectivity as compared to the OER, we obtained the free energy diagrams for OER on the Pt–N$_4$ sites (Supplementary Fig. 36 and Supplementary Note 3). Note that the thermodynamic overpotentials for OER ($\eta_{TD(OER)}$) of all Pt–N$_4$ sites were evaluated at the overpotential of 0.13 V (i.e., $\eta_{OER} = 1.36 - 1.23$ V $= 0.13$ V), to be referenced to the same potential for the CER (i.e., $\eta_{CER} = 0$ V). For the PtN$_4$C$_{12}$ and PtN$_4$C$_{10}$ sites, OH$^*$ adsorption from H$_2$O$_{(l)}$ was found to be the most endothermic among the reaction steps, corresponding to the potential determining step (PDS) with a large $\eta_{TD(OER)}$ above 0.74 V (Supplementary Table 5). Although the PtN$_{2+2}$C$_{4+4}$ site exhibited relatively stable reaction pathways because the value of CN for oxygen-involving intermediates was 4 (Supplementary Fig. 36b) in comparison to other Pt–N$_4$ sites, it was still considered inactive for OER with very high $\eta_{TD(OER)}$ of 0.62 V, corresponding to the formation of OOH$^*$. This result was in good agreement with experimental observation, where OER did not occur even in the absence of Cl$^-$ for Pt$_1$/CNT (Figs. 2a and 3a). Particularly focusing on the PtN$_4$C$_{12}$ site, which most plausibly catalyses the CER in Pt$_1$/CNT, a huge difference between $\eta_{TD(OER)}$ and $\eta_{TD(CER)}$ at the PDS (i.e., $\eta_{TD(OER)} - \eta_{TD(CER)} = 0.99$ V) further corroborates the excellent CER selectivity of Pt$_1$/CNT compared to the OER.

## Discussion

We synthesised Pt$_1$/CNT catalysts comprising dense atomically dispersed Pt–N$_4$ sites via simple high-temperature pyrolysis of a Pt-porphyrin precursor without the agglomeration of Pt. We demonstrated that Pt$_1$/CNT with atomically dispersed Pt–N$_4$ sites could catalyse CER with high activity and selectivity. Pt$_1$/CNT showed superior CER activity compared to PtNP/CNT and commercial Ru/Ir-based DSA. Importantly, Pt$_1$/CNT preserved high CER selectivity in acidic pH with low Cl$^-$ concentration, as well as in neutral pH, where DSA showed

lower selectivity owing to the concomitant generation of $O_2$. By combining the in situ electrochemical XANES and DFT calculations, the atomically dispersed $PtN_4C_{12}$ was identified as the most plausible active site structure for CER. Notably, this work presents the first use of atomically dispersed catalysts for CER. We expect this type of catalyst to be exploited as an alternative to MMO-based catalysts, whose activity and selectivity are intrinsically limited by the scaling relationship between CER and OER. Furthermore, the excellent selectivity of $Pt_1/CNT$ under a wide range of $Cl^-$ concentrations and pH suggest its promising applicability in wastewater and ship ballast water treatments.

## Methods

**Chemicals.** Sodium chloride (NaCl, ≥99%) and sodium iodide (NaI, anhydrous, ≥99.5%) were purchased from Sigma-Aldrich. Multi-walled carbon nanotubes (CNT, MR99, average diameter of 10 nm and average length of 10 μm) were purchased from Carbon Nano-material Technology Co. (Korea). Nitric acid ($HNO_3$, 60%), hydrochloric acid (HCl, 36%), acetone (99.7%), ethanol (94.5%), and anhydrous ethanol (99.9%) were purchased from Samchun Chemicals (Korea). Perchloric acid ($HClO_4$, 70%, Veritas double distilled) was purchased from GFS Chemicals. Hydrogen peroxide ($H_2O_2$, 30%) was purchased from Junsei Chemical. Carbon paper (TGP-H-60, Toray), Sodium perchlorate monohydrate ($NaClO_4·H_2O$, >97%), standardised sodium thiosulfate solution ($Na_2S_2O_3$, 0.01 N), and soluble starch powder (ACS, for iodometry) were purchased from Alfa Aesar. Pt(II) meso-tetraphenylporphine (PtTPP, >95%) were purchased from Frontier Scientific. Dihydrogen hexachloroplatinate(IV) hydrate ($H_2PtCl_6·6H_2O$, 99.95%) was purchased from Umicore. Natural seawater, obtained from Ilsan beach, Ulsan, Republic of Korea (GPS 35.497005, 129.430996), was used after filtration, whose relative ion concentrations can be found in a previous report[67]. Commercial dimensionally stable anode (DSA, Ir/Ru atomic ratio = 2) was provided by Siontech Inc. (Korea). All chemicals were used without further purification, except the CNT.

**Acid treatment of carbon nanotube.** Prior to the synthesis of the catalysts, CNT was treated with heat and acids to remove any metallic impurities. CNT (38.0 g) was calcined in a box furnace at 500 °C for 1 h (ramping rate: 7.9 °C min⁻¹). The heated powder was mixed with a solution of 810 g of 6 M $HNO_3$ (diluted from 60% $HNO_3$), and the mixture was stirred at 80 °C for 12 h. The suspension was filtered and washed with excessive amounts of DI water until the filtrate reached a pH of 7. The powder was subsequently treated with 720 g of 6 M HCl (diluted from 36% HCl) as described above, then dried at 60 °C overnight.

**Synthesis of $Pt_1/CNT\_X$ catalysts.** Five hundred milligram of acid treated CNT and 71 mg of PtTPP were ground in an agate mortar until the colour and texture did not change (for ~20 min). The mixture was heated at a desired temperature between 500 °C and 800 °C under 1 L min⁻¹ $N_2$ flow (99.999%) for 3 h (ramping rate: 2.1 °C min⁻¹). The resulting catalysts were designated as $Pt_1/CNT\_X$ ($X$ = annealing temperature).

**Synthesis of PtNP/CNT catalyst.** Forty four milligram of $H_2PtCl_6·6H_2O$ was dissolved in 5.4 mL of acetone. The solution was mixed with 500 mg of acid treated CNT in a plastic bag by hand scrubbing, then dried at 60 °C overnight. The powder was heated to 200 °C under 0.5 L min⁻¹ $H_2$ flow (99.999%) for 2 h (ramping rate: ~0.6 °C min⁻¹). Subsequently, the temperature was elevated to 350 °C (ramping rate: ~2.5 °C min⁻¹) under 1.0 L min⁻¹ $N_2$ flow (99.999%) and maintained at that temperature for 3 h.

**Characterisation methods.** The HR-TEM and HAADF-STEM images were taken using an Titan³ G2 60-300 microscope (FEI Company) equipped with a double-sided spherical aberration (Cs) corrector operating at an accelerating voltage of 80 kV and 200 kV, respectively. The size distributions of Pt particles were analysed using the Gatan Microscopy Suite 3 Software. A total of 275 and 295 particles were recorded from two HAADF-STEM images of $Pt_1/CNT$ (17 nm × 17 nm and 24 nm × 24 nm) and PtNP/CNT (144 nm × 144 nm and 288 nm × 288 nm), respectively (Supplementary Figs. 1c and f). Scanning electron microscopy (SEM) images were obtained using an S-4800 field emission scanning electron microscope (Hitachi High-Technologies). The XRD patterns were obtained using a high-power X-ray diffractometer (D/MAX2500V/PC, Rigaku) equipped with Cu Kα radiation and operating at 40 kV and 200 mA. The XRD patterns were measured in a 2θ range from 10° to 80° and from 30° to 50° at a scan rate of 2° min⁻¹ and 0.5° min⁻¹, respectively. The XPS measurements were performed with a K-alpha instrument (Thermo Fisher Scientific) equipped with a monochromatic Al Kα X-ray source (1486.6 eV). Pt 4f and N 1s XPS spectra were deconvoluted using the XPSPeak41 software with the mixed (Gaussian 70, Lorentzian 30)-function after a linear (Shirley)-type background correction. The Pt, Ru and Ir contents in the catalysts were analysed using an ICP-OES analyser (700-ES, Varian). The C, H, N

and O contents in the catalysts were measured using a combustion elemental analyser (Flash 2000, Thermo Fisher Scientific).

**X-ray absorption spectroscopy.** XAS was performed at the beamlines of 6D, 8C and 10C of the Pohang Accelerator Laboratory. The storage ring was operated at an energy of 3 GeV and a beam current of 360 mA. The incident beam was filtered by a Si(111) double crystal monochromator and detuned by 20% to remove the high-order harmonics. The incident photon energy was then calibrated using a standard Pt foil where the maximum of the first derivative of absorption of the Pt foil reference is located at 11,564 eV. The powder sample was pressed using a hand-pelletiser to the desired thickness to ensure that the X-ray beam could pass through a large enough number of Pt atoms, resulting in an absorption edge step ranging from 0.4 to 1.0. The background removal and normalisation of the Pt $L_3$-edge XAS spectra were conducted using Athena software[68]. The Fourier transform of $k^3$-weighted extended EXAFS spectra was performed using the Artemis software to obtain the coordination numbers and interatomic distances without phase correction. The fitting was conducted in the $k$ range of 3.0–11.0 Å⁻¹, whereas Pt foil was fitted in the $k$ range of 3.0–14.0 Å⁻¹. All fitting results were obtained under $k^3$ weighting. Crystallographic data of the PtTPP molecule were used for multi-shell fitting with the first shell of Pt−N and the second shell of Pt···C[69]. Throughout the fitting analysis, the amplitude reduction factor ($S_0^2$) of Pt was fixed at 0.85, which was obtained by fitting the EXAFS spectrum of Pt foil.

For in situ electrochemical XAS analyses, a home-made polytetrafluoroethylene spectroelectrochemical cell was used. A catalyst ink was deposited onto the tip of a carbon paper strip (1.0 cm × 1.5 cm) to control the catalyst loading at 2 mg cm⁻². As-deposited film was attached onto the window of the cell with the catalyst layer facing the cell to be in contact with the electrolyte. Subsequently, the window was fully blocked using Kapton tape, and 0.1 M $HClO_4$ + 1.0 M NaCl solution was poured into the cell. The electrolyte was sparged with Ar gas (99.999%) for at least 20 min. A three-electrode system was built using a graphite rod and KCl-saturated Ag/AgCl (RE-1B, ALS) as the counter and reference electrodes, respectively. XAS spectra were obtained after applying open circuit potential (OCP) for 20 min. XAS measurements were conducted sequentially for each applied potential of 1.15, 1.25, 1.4 and 1.5 V (vs. RHE, without $iR$ compensation), after applying each potential for 20 min. The measurement was performed using the fluorescence detection mode. The experiments were repeated under the OER condition described above, except that the concentration of $HClO_4$ electrolyte was 0.1 M.

**Electrochemical cell construction.** A three-electrode system was built using an H-type cell to separate the working electrode from the counter electrode, in which a reference electrode was placed at the compartment of working electrode. Each compartment of the H-type cell was separated by a Nafion 117 membrane (DuPont). Prior to use, the Nafion membrane was pretreated with 5% $H_2O_2$ and heated at 60 °C for 1 h. An E7 RRDE (AFE7R9GCPT, Pine Research Instrumentation, the collection efficiency of 0.37), Pt counter electrode, and KCl-saturated Ag/AgCl electrode were used as the working, counter, and reference electrodes, respectively. An MSR rotator (Pine Research Instrumentation) was used for controlling the rotation speed of the RRDE. The electrolytes were prepared by diluting 70% $HClO_4$ and by adding 99% NaCl in 18.2 MΩ cm Millipore water. For experiments under different NaCl concentrations, $NaClO_4·H_2O$ was added into the electrolyte to compensate the total ionic strength[23]. The pH values of the acidic electrolytes were adjusted to 0.90 ± 0.05 by adding a few drops of 70% $HClO_4$. The pH values of all electrolytes were measured using a digital pH metre (Orion A211, Thermo Fischer Scientific).

**RHE calibration.** RHE conversion was achieved with a two-electrode setup, where the Pt wire and the reference electrode to be calibrated were immersed. Then, OCP was measured with high purity $H_2$ gas (99.9999%), which was sparged into the electrolyte. In this setup, the Pt coil served as the RHE as $H^+/H_2$ equilibrium was established. An OCP was applied for 30 min to obtain a stable potential, i.e., the potential difference between the RHE and reference electrode, which was used as the conversion value.

**General electrochemical methods.** Electrochemical measurements were performed on an electrochemical workstation (CHI760E, CH Instruments) at atmospheric pressure. All potentials were converted to the potential scale of RHE, unless otherwise noted. Geometric current density ($j$, mA cm⁻²) and geometric charge density ($\sigma$, C cm⁻²) were calculated by dividing the measured current and the measured charge by the geometric area of electrode, respectively. Before every measurement, the RRDE was polished on a micro-cloth with aqueous suspensions of 1.0 μm and then 0.3 μm alumina to generate a mirror finish. The catalyst ink was prepared by mixing 2.5 mg of catalyst, 50 μL of DI water, 20 μL of 5% Nafion solution (D521, DuPont), and 530 μL of anhydrous ethanol, and homogenising in an ultrasonic bath (Branson) for at least 40 min. Six microlitre of the catalyst ink was deposited onto a glassy carbon (GC) disk (5.61 mm in diameter) using a micro syringe (Hamilton) and dried at RT. The resulting catalyst loading was 0.1 mg cm⁻². For the stability tests, 24 μL of the catalyst ink was deposited onto the tip of a carbon paper strip (1 cm × 1 cm, catalyst loading: 0.1 mg cm⁻²). Prior to electrochemical measurements, the catalyst film was immersed into an electrolyte of 0.1 M $HClO_4$, which was sparged with Ar gas (99.999%) for at least 20 min. CV

was conducted to clean and make the catalyst fully wet at a scan rate of 500 mV s$^{-1}$ for 50 cycles between 0.05 and 1.2 V. Then, the Pt ring electrode (outer diameter = 7.92 mm and inner diameter = 6.25 mm) was cleaned in the same potential range with a scan rate of 500 mV s$^{-1}$ for 10 cycles. All electrochemical measurements were performed in triplicate, and the averaged values were used.

**Electrochemical CO stripping for active surface area.** For CO stripping experiments, the 0.1 M HClO$_4$ electrolyte was purged for 10 min with CO gas (30.0%, Ar-balanced), while the constant potential of disk electrode was kept at 0.1 V. The sparging gas was changed to Ar for 30 min to remove the CO from the electrolyte while the disk potential was fixed at 0.1 V. The CO monolayer absorbed on the surface of Pt was stripped off by conducting three sequential CVs from 0.1 V to 1.0 V at 20 mV s$^{-1}$. The third CV was displayed as a blank. The electrochemical surface area was calculated using the standard charge of the surface coverage of monolayer CO (420 μC cm$^{-2}$). The number of active sites ($n$) of PtNP/CNT catalyst could be quantified using the CO stripping charge ($Q_{CO}$) with the following equation.

$$n\,(\text{mol}) = \frac{Q_{CO}}{2F} \tag{7}$$

where $F$ represents the Faraday constant. The factor ½ is based on the approximation that two electrons are required for the desorption of one CO molecule during CO stripping. For the Pt$_1$/CNT catalyst, the number of active sites of Pt$_1$/CNT was calculated by the Pt content of Pt$_1$/CNT catalyst loaded on the electrode.

**Calculation of turnover frequency.** The number of active sites ($n$) of Pt$_1$/CNT catalyst was calculated by the following equation.

$$n\,(\text{mol}) = \frac{m_{Pt}}{M_{Pt}} = \frac{w_{Pt} \times \rho_{cat} \times V_{cat}}{100 \times M_{Pt}} \tag{8}$$

where $m_{Pt}$ is the amount of Pt in the catalyst layer, $M_{Pt}$ is the molar mass of Pt (195.084 g mol$^{-1}$), $w_{Pt}$ is the weight percent of Pt in Pt$_1$/CNT (2.7 wt%), $\rho_{cat}$ is the mass concentration of Pt in the catalyst ink (4.17 g L$^{-1}$), and $V_{cat}$ is the volume of the loaded catalyst ink (6 μL). The turnover frequency (TOF) of Pt$_1$/CNT can be calculated as follows.

$$\text{TOF}\,(\text{s}^{-1}) = \frac{i_d}{2 \times n \times F} \tag{9}$$

where $i_d$ is the disk current during CER measurement in 0.1 M HClO$_4$ + 1.0 M NaCl, $n$ is the number of active sites and $F$ is the Faraday constant. The factor ½ is based on the two electrons that are transferred for the oxidation of two Cl$^-$ ions to one Cl$_2$ molecule.

**Rotating ring-disk electrode detection of Cl$_2$ evolution.** For hydrodynamic Cl$_2$ detection[51], the activity of CER was measured in the Ar-saturated 0.1 M HClO$_4$ with controlled concentrations of NaCl. Before measuring the CER activity, electrochemical impedance spectroscopy (EIS) was conducted at a fixed potential of 0.9 V without $iR$ compensation from 100,000 Hz to 1 Hz with a potential amplitude of 10 mV at an electrode rotation speed of 1600 rpm. In a Bode plot, the solution resistance ($R_u$) was determined as the magnitude of impedance, which is closest to zero. The Nyquist plots were obtained at a fixed potential of 1.4 V without $iR$ compensation to measure the charge transfer resistance ($R_{ct}$) after the CER activity of catalysts was consistently obtained. The Nyquist plots were fitted to obtain $R_{ct}$ using the ZView software based on the equivalent circuit shown in Supplementary Fig. 16a, which was built on the previous report[70]. The pseudocapacitance ($C_p$) and its coupled resistance ($R_p$) originate from the adsorption and desorption of Cl$^-$ intermediates[70]. Owing to the uncertain nature of the electrode-solution interface, the double layer capacitance ($C_{dl}$) and $C_p$ were fitted with constant phase elements (CPEs) instead of an ideal capacitor. The charge transfer resistances of the catalysts are summarised in Supplementary Table 4. The $iR$ compensated potentials ($E - iR$) indicate that the potential was corrected by 85% after measurements with the values of $R_u$. CVs were conducted from 1.0 to 1.6 V at a scan rate of 10 mV s$^{-1}$ with an electrode rotation speed of 1600 rpm, while the ring potential was held at 0.95 V.

Five sequential CVs of CER were conducted to obtain steady catalyst performance. The forward scan of fifth CV was used for the representative CER polarisation curve. With the above RRDE setup above, Cl$_2$ generated at the disk electrode was reduced back to Cl$^-$ at the ring electrode. The on-set potential of the catalyst was determined using the ring current indicating Cl$^-$ reduction. The Tafel plots were achieved according to the following Tafel equation.

$$\eta\,(\text{V}) = A \times \log j_d - A \times \log j_0 \tag{10}$$

where $\eta$, A, $j_d$ and $j_0$ represent the overpotential, the Tafel slope, the disk current density, and the exchange current density of the disk, respectively. Tafel slope was obtained in the potential range with the coefficient of determination ($R^2$) over 0.99.

The Cl$_2$ selectivity ($\varepsilon_{CER}$) of catalyst was measured by RRDE chronoamperometry (CA) for 120 s with an electrode rotation speed of 1600 rpm. The CA was sequentially tested for five times with recurring intermittent breaks of 5 min. The applied disk potential was adjusted to generate a current density higher than 10 mA cm$^{-2}$ for 600 s, while the applied ring potential was fixed at 0.95 V. Prior to the RRDE CA of Cl$_2$ evolution, the background currents of disk and ring

were measured by CA with both the disk and ring potential held at constant 0.95 V with an electrode rotation speed of 1600 rpm. The net CER current ($i_{CER}$) on the disk electrode can be calculated by the following relations.

$$i_{CER} = \left|\frac{i_r}{N}\right| \tag{11}$$

where $i_r$, and $N$ denote the background-corrected ring current and collection efficiency, respectively. The Cl$_2$ selectivity was calculated by the following relations.

$$\text{Cl}_2 \text{ selectivity (\%)} = 100 \times \frac{2 \times i_{CER}}{i_d + i_{CER}} = 100 \times \frac{2 \times \left|\frac{i_r}{N}\right|}{i_d + \left|\frac{i_r}{N}\right|} \tag{12}$$

where $i_d$ represents the background-corrected disk current.

**Iodometric titration for Cl$_2$ detection.** The Cl$_2$ selectivity was examined by iodometric titration, as reported by Koper et al.[51]. The Cl$_2$ selectivity was calculated according to the following relation.

$$\text{Cl}_2 \text{ selectivity (\%)} = 100 \times \frac{\text{Experimental yield}}{\text{Theorecital yield}} \tag{13}$$

The experimental condition and setup were the exact same as those used in RRDE studies. Each part of the H-type cell was filled with 45 mL of electrolyte using a micropipette. The electrolyte was sparged with Ar gas for 20 min before Cl$_2$ evolution. CA was then performed with the applied potential, which was adjusted to generate a current density higher than 10 mA cm$^{-2}$ for 120 s. The theoretical yield was calculated from the CA responses using the following relation.

$$\text{Theoretical yield (mol)} = \int \frac{i}{2F} \text{d}t \tag{14}$$

where $i$, $F$ and $t$ represent the current, the Faraday constant, and the time, respectively. The factor ½ is based on that the two electrons that were transferred for the oxidation of two Cl$^-$ ions to one Cl$_2$ molecule.

Immediately after finishing the CA, 10 mL of the anodic electrolyte was moved into 20 mL vial containing a large excess (~100×) of NaI to minimise the equilibrium concentration of volatile I$_2$. Three titrations were conducted for a single CA measurement. The vial was closed air-tight and the colour of the solution was observed to rapidly turn into yellowish brown due to the generation of I$_2$ from the following reaction.

$$\text{Cl}_2 + 2\,\text{NaI} \rightleftharpoons \text{I}_2 + 2\,\text{NaCl} \tag{15}$$

I$_2$ was then titrated with a standardised 0.01 N Na$_2$S$_2$O$_3$ solution. When the yellow colour become pale, few drops of 1% (w/v) starch indicator were added, which changed the colour to the solution to dark blue. The titration was completed with additional Na$_2$S$_2$O$_3$ solution. Two Na$_2$S$_2$O$_3$ molecules are oxidised for the reduction of one I$_2$ molecule. The experimental Cl$_2$ yield (in mole) can be calculated with the volume of the Na$_2$S$_2$O$_3$ solution used in the titration process from the following equation.

$$\text{Experimental yield (mol)} = \frac{0.01\,\text{M} \times \text{Volume of Na}_2\text{S}_2\text{O}_3\,(\text{L})}{2} \tag{16}$$

## Data availability
The data supporting this study are available from the corresponding author upon reasonable request.

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

## Acknowledgements

This work was supported by the National Research Foundation (NRF) of Korea funded by Ministry of Science and ICT (NRF-2014R1A5A1009799, NRF-2017R1A2B2008464, NRF-2017R1A4A1015533, NRF-2019M3E6A1064521 and NRF-2019M3D1A1079306). S.K.K. acknowledges computational resources from UNIST-HPC. The XAS experiment performed at Beamlines 6D, 8C and 10C of the Pohang Accelerator Laboratory (PAL) were supported in part by the MIST and Pohang University of Science of Technology (POSTECH).

## Author contributions

S.H.J. supervised the project. S.H.J., S.K.K., T.L. and G.Y.J. conceived and designed the experiments and calculations. T.L. conducted synthesis, characterisations and electro-chemical analyses. G.Y.J. and S.O.P. performed DFT calculations and analysed the results. J.P. and S.J.K. discussed on the analysis of $Cl_2$ selectivity. H.Y.J. and J.H.K. contributed to TEM analysis. Y.-T.K. commented on the paper. T.L., G.Y.J., S.K.K. and S.H.J. co-wrote the paper.

## Competing interests

The authors declare no competing interests.
