## [Peer Review File · Nature Communications]

Reviewers' comments:

Reviewer #1 (Remarks to the Author):

This manuscript reports an atomically dispersed Pt-N4 sites for the chlorine evolution reaction with high activity and selectivity. A suite of experimental techniques is used to characterize the as prepared catalysts. In general, this is an interesting study. The paper could be accepted with the following minor revisions:

1. The authors used different annealing temperature to prepare Pt1/CNT_X samples, how the annealing temperature affect the activity and selectivity of the samples?
2. The STEM images of other Pt1/CNT_X samples also need to be provided.
3. The authors claim that the Pt1/CNT has high CER selectivity, but based on Supplementary Fig. 14, 15, and 17, PtNP/CNT even demonstrated better selectivity than Pt1/CNT. Why PtNP/CNT also exhibits excellent selectivity?
4. Can the loading mass of Pt be tuned to achieve better activity and selectivity?
5. The high activity Pt1/CNT could be well preserved after durability tests?
- 6 Line 87-88, 90. The unit of "centigrade" is missing. This problem could be found in the whole manuscript.
- 7 The locations of Pt2+ peaks are different in Figure S4. Please explain.
- 8 The equivalent circuit should be displayed and explained in Figure S13.
- 9 As displayed in Figure S12, the CER performance of 700 and 800 showed almost no difference, which is contradicted with the EIS shown in Figure S13. Please explain.
- 10 As shown in Line 234-236, the calculated model of PtNP/CNT is based on "(110) surface of disordered PtO2". Hence it is disordered PtO2, the rationality of atomic arrangement should be explained. And is it able to conclude that PtNP/CNT shows poor selectivity and efficiency towards CER is because of the oxidising of Pt? Hence it is suggested to do further calculations on pure Pt model
- 11 As shown in Line 234-236, the calculated model of PtNP/CNT is based on "(110) surface of disordered PtO2". Hence it is disordered PtO2, the rationality of atomic arrangement should be explained. And is it able to conclude that PtNP/CNT shows poor selectivity and efficiency towards CER is because of the oxidising of Pt? Hence it is suggested to do further calculations on pure Pt model

Reviewer #2 (Remarks to the Author):

In this manuscript, the authors present a detailed analysis of synthesis and characterization of a Pt-based single atom catalyst shown to exhibit superior activity and selectivity for the chlorine evolution reaction (CER) under relevant conditions. This is very impressive and high impact work that, aside from a few points, is quite clear and accurate. A few things that should be addressed prior to publication:

Some conversion issues on lines 87, 88, and 90 (degrees C?) continues throughout manuscript.

In SI Fig. 1, frequency is given but please also provide out of how many counts and over what relevant area to give better understanding of statistics and how well represented the materials are in frames c and f.

XAS implies Pt-N coordination but EELS could show this explicitly (see for instance, Chung, et al., Science 2017 on PGM-free ORR electrocatalysts). This could provide further input on average coordination number.

It is possible to derive the N 1S spectra for specific structures from DFT which could be beneficial for corroborating nature of the active site via DFT beyond just activity comparisons. See for instance

Artyushkova, et al., ChemComm 2013.

Further details of the MWCNT would be beneficial to the reader (average diameter and number of walls, in particular).

Line 152: "all sites were considered active" – Which method for determining final amount of Pt was used in this calculation? Please be clear as identification of number of sites in SAC materials is an important issue especially for comparison across materials.

In the DFT study only planar structures are considered. Given that these are suggested to be hosted in the CNT which is radially strained, shouldn't a radially strained model be applied? This has previously been shown to make a significant impact on binding energies for other electrochemical reactions (ORR in particular) and thus, likely, the activity descriptor used in this study. This must be addressed.

The DFT derived "overpotential" is actually thermodynamic overpotential since no kinetics are included. This must be clarified or at least acknowledged, though this still is a valid descriptor for comparison between sites as determined in extensive HER literature.

In SI Note 4, 1st sentence "Pt atom on the..." seems to be missing a description of the binding site for delEads, though in equations the site is described as "Pt-free".

For the stability from DFT, at 1.36V in acid, Pt bulk is not the relevant low-free-energy state to compare to but rather a solvated Pt²⁺ ionic state would be a more appropriate energetic comparison. This will give a constant energetic shift at constant conditions between the bulk and ionic Pt, however, so using this to evaluate relative stability should give the same result but a change in sign (likely) would indicate that the Pt ion was more stable under such conditions than the PtN₄. This should be addressed since stability against metal dissolution is likely a major contributor to active site loss.

Reviewer #3 (Remarks to the Author):

Atomically dispersed Pt-N₄ sites as efficient and selective electrocatalysts for the chlorine evolution reaction

T. Lim et al.

In the present manuscript, the authors report the synthesis, characterization, and application of atomically dispersed Pt-N₄ on carbon nanotubes for the chlorine evolution reaction (CER), a reaction system of potential interest for chemical industry due to the production of gaseous chlorine in chlor-alkali electrolysis. The measured activity and selectivity results are compared to those of commercial DSA, which are applied as state-of-the-art electrocatalysts in the chlor-alkali process. Density functional theory calculations are employed to gain insights into the high CER activity and selectivity on a molecular level.

The application of atomically dispersed electrocatalysts is innovative and corresponds to an alternative approach compared to commonly studied transition-metal oxide-based electrodes for chlorine electrocatalysis. Therefore, the manuscript might deserve to get published in Nat. Comm., but not in the present form. Before this reviewer can recommend the article for publication, the authors need to carry out a major revision by taking the following 23 points into account:

i) Literature survey: recently, two articles concerning the CER selectivity of precious metal oxides have been published in ChemElectroChem that deserve further attention, but at least should be cited:

10.1002/celec.201900784

10.1002/celec.201900834

ii) Line 33: 60 millions tons of Cl₂ per year is outdated. Nowadays, the annual production amounts to about 70 millions tons or even more. I advise the authors to refer to state-of-the-art reports, such as Chlor Alkali Industry Reviews of Euro Chlor, and to cite those.

iii) Line 36: Equilibrium potential of the CER is missing.

iii) Line 42: The equilibrium potential of the OER is reduced (on the SHE scale) by increasing the pH, resulting in an enhanced overpotential and, hence, a higher catalytic turnover at a constant electrode potential on the SHE scale compared to smaller pH values. The term "thermodynamic barrier", however, is misleading, because the reduction of the OER equilibrium potential directly affects the underlying kinetics.

iv) Line 43: It would be advantageous to provide a reaction equation for the OER, including its equilibrium potential, in comparison to that of the CER (cf. line 36).

v) Line 55: Not correct, about 30% of precious metals, such as Ru, are needed to have sufficient electronic conductivity.

vi) Line 73: "lower Tafel slope": this statement is insufficient. The Tafel slope is overpotential dependent, i.e., the lower the overpotential, the smaller the Tafel slope, and vice versa. Every electrocatalyst exhibits a threshold overpotential, at which the Tafel slope exceeds 60 mV/dec. This is due to the fact that the first elementary reaction step in the free-energy landscape becomes rate determining. I advise the authors to refer to the state-of-the-art literature concerning the connection of Tafel slopes and free-energy diagrams and to revise this statement. If Tafel slopes are reported, it is indispensable to enclose the accompanied overpotential range for the respective Tafel slope.

vii) Line 138: "Onset potentials" are physicochemically not well defined. If the authors want to compare the activity of electrocatalysts at low overpotentials, they should derive and compile exchange current densities.

viii) Line 144/145: Under the reaction conditions of the authors, the equilibrium potential of the CER is likely not equal to the standard equilibrium potential (1.36 V vs. SHE), but rather $U_{\text{CER}} < 1.36 \text{ V vs. SHE}$ due to $a_{\text{Cl}_2} < 1$ is valid. I advise the authors to refer to the literature concerning the determination of the equilibrium potential for the CER. If I am not mistaken, some information can be found in reference 21.

ix) Line 146/147: At which temperatures were the measurements conducted? Room temperature? In chemical industry, CER is carried out at about 80 – 90 °C. Are the proposed Pt/CNT electrocatalysts stable at this temperature? In order to compare the performance of the Pt/CNT electrodes to that of DSA, I think the activity and stability at higher temperature needs to be probed. Otherwise, the authors need to clearly state, which temperature interval was investigated in their study. Throughout the entire manuscript, no temperature concerning the activity or selectivity measurements is reported.

x) Line 158: Supplementary Table 3: I suppose that the overpotential @10 mA/cm² is calculated based on the standard equilibrium potential, $U_{0\text{CER}} = 1.36 \text{ V vs. SHE}$. Please revise referring to point viii). The analysis would be even stronger, if exchange current densities would be provided additionally.

xi) Line 177/178: The comparison of the Pt/CNT electrodes to DSA appears a bit odd, since DSA are stable up to 10 years. A current drop of 28% after already 12 h appears not to be in good agreement with the performance of DSA.

xii) Line 202/203: Please revise the Tafel slope analysis referring to point vi).

xiii) Line 205/206: The Heyrovsky step would be rate determining if chlorine is not already adsorbed on the catalyst's surface when CER takes place. However, if the active surface configuration contains adsorbed chlorine, then the Heyrovsky step is the first step in the reaction mechanism. Correspondingly, the Volmer step, which as second step closes the electrocatalytic cycle, becomes rate determining, as long as the Tafel slope is smaller than 60 mV/dec. Therefore, I suppose that the author need to revise their statement.

xiv) Line 223/224: Pourbaix diagrams rely on a constrained thermodynamics approach, that is, the actual reaction (CER/OER) is suppressed, but the adsorption of CER/OER reactants on the catalyst's surface is allowed to proceed. I advise the authors to refer to the state-of-the-art literature in the field of Pourbaix diagrams for the CER (e.g., 10.1021/acscatal.8b01432, 10.1002/cssc.201900298) and to add an explanatory sentence, why Pourbaix diagrams are a valuable tool to resolve the surface structure of an electrocatalyst under reaction conditions.

xv) Line 226/227: CER is pH-independent on the SHE scale, but not on the RHE scale. The opposite case holds true for the OER. The authors are mixing the potential reference scales throughout the manuscript. I advise the authors to refer all electrode potentials either to the SHE or RHE scale throughout the entire manuscript.

xvi) Line 230-236: Relating to Supplementary Figure 20, Cl* corresponds to the active surface configuration (cf. point xiii).

xvii) Line 245: Figure 4c: I advise the authors to refer to the state-of-the-art literature concerning the construction of free-energy diagrams: Pourbaix diagrams in the initial step are applied to identify the active surface configuration (Cl* as thermodynamically most stable state), from which the reaction mechanism starts. Intermediate states in the free-energy diagram at $\eta = 0$ V can be uphill only, but never downhill.

xviii) Line 246-249: I suppose that ΔG_{Cl^*} is also referred to a reference state (such as the surface without an adsorbate, i.e., *). Please revise. The theoretical/thermodynamic overpotential, corresponding to the concept of Norskov and co-workers, is not a free-energy change, but a free-energy change ($\eta = 0$ V) divided by the elementary charge. This is not backed up in the main text, where the introduction of the theoretical overpotential is confusing. It would be helpful for the reader to cite the corresponding paper of Norskov and co-workers (i.e., 10.1021/jp047349j), where the concept of η_{TD} was introduced. Please name the theoretical overpotential η_{TD} and not η_{CER} to avoid confusion: η_{TD} is not related to any experimental CER overpotential.

xix) Line 249: PtN4C12 is not a thermoneutral catalyst at 1.36 V. For this electrode material, the CER via the thermodynamics of the Volmer-Heyrovsky mechanism is thermoneutral at $\eta = 0.09$ V. Please revise. Actually, there was a recent publication in ACS Catalysis (10.1021/acscatal.9b00732), which suggests that it is beneficial for chlorine electrocatalysis if the thermodynamics of the reaction intermediate is thermoneutral at target overpotential. This finding would be fulfilled for PtN4C12, since typical CER overpotentials, referring to the chlor-alkali process, are about 100 mV.

xx) Line 258: ClO* has also been identified as reaction intermediate in the CER over other transition-metal oxides, such as RuO₂ or IrO₂ (cf. reference 13, reference 20, 10.1021/acscatal.8b01432, 10.1063/1.5051429).

xxi) Line 262: Please revise Figure 4d according to point xvii).

xxii) Line 266: "CN" has not been defined in the main text.

xxiii) The authors compile free-energy diagrams for the CER and OER of the proposed PtN4C12 electrocatalysts; however, the reason for the high CER selectivity on an atomic scale has not been

sufficiently elaborated: a thorough comparison of the energetics for the proposed pathways of CER and OER based on the free-energy landscapes is deeply missing.

<List of Major Changes>

For Main Text

1. Throughout the manuscript, we revised the unit of centigrade ($^{\circ}\text{C}$) and specified the temperature of electrochemical measurements.
2. Throughout the manuscript, we revised the overpotential based on the Nernst equation.
3. Page 3, we added the reaction information of CER and OER regarding Supplementary Note 1 (Chemical reactions and standard potentials for CER and OER).
4. Page 5, we added new sentences regarding newly added Supplementary Fig. 2 (EELS of Pt_1/CNT).
5. Page 6, we added new sentences regarding newly added Supplementary Fig. 6 (HAADF-STEM images of Pt_1/CNT_X samples).
6. Page 6–7, we revised and added the sentences regarding the XPS spectra of Pt_1/CNT_X samples.
7. Page 7, we added new sentences regarding the temperature for all of our electrochemical experiments.
8. Page 7–8, we added the sentences and equations regarding temperature-dependent equilibrium potential of CER.
9. Page 8–9, we specified the calculation for the TOFs and exchange current density.
10. Page 8–9, we added new sentences regarding newly added Supplementary Fig. 13 (CER performance under $80\text{ }^{\circ}\text{C}$).
11. Page 9, we added new sentences regarding newly added Supplementary Fig. 16a and Supplementary Table 4 (equivalent circuit for CER and fitting results of Nyquist plots).
12. Page 10, we added new sentences regarding newly added Supplementary Fig. 17 (CER LSV after 12 h of stability test under $25\text{ }^{\circ}\text{C}$).
13. Page 10–11, we added new sentences regarding newly added Supplementary Fig. 22 and 23 (iodometric titration for CER selectivity in neutral 0.55 M Cl and RRDE CA for Pt_1/CNT_X , respectively).
14. Page 11, we specified the range of Tafel plots to obtain Tafel slope.
15. Page 12, we added and revised sentences regarding Supplementary Fig. 24–27 (Pourbaix diagrams of three Pt-N_4 sites).
16. Page 12–13, we added and revised sentences regarding Supplementary Fig. 28–31 (modelling and Pourbaix diagrams of possible PtO_2 under CER conditions).
17. Page 13, we added and revised sentences regarding Supplementary Fig. 32 and Supplementary Note 5 (stability calculation of Pt-N_4 sites).
18. Page 14–15, we added new paragraph regarding Supplementary Note 6 (Full free energy diagram for CER over Pt_1/CNT).
19. Page 16–25, we added and revised sentences in Method section regarding experimental details in manuscript.
20. Page 26–30, we added 14 new references (reference number: 1, 21, 22, 47, 48, 52, 53, 59, 62, 63, 64, 65, 66, and 70) and rearranged the order of references in accordance with the changes.
21. Page 29, we removed original reference 61.

22. Page 33, we revised Fig. 2b, e (TOFs and Tafel plots) and added relating information on the calculation of TOF and Tafel plots throughout the manuscript.
23. Page 35, we revised Fig. 4c (free energy diagrams for CER) and newly added Fig. 4d (Full free energy diagram along the reaction coordinate of CER over the Pt₁/CNT). The original Fig. 4d was revised and moved to Supplementary Fig 36.

For Supporting Information

1. Page S5, we revised Supplementary Fig. 1c,f (Pt size distribution histograms of Pt₁/CNT and PtNP/CNT).
2. Page S6, we newly added Supplementary Fig. 2 (EELS of Pt₁/CNT).
3. Page S10, we newly added Supplementary Fig. 6 (HAADF-STEM images of Pt₁/CNT_X samples).
4. Page S13, we have revised the Supplementary Fig. 9a (Deconvoluted N 1s XPS spectra of P Pt₁/CNT_X samples).
5. Page S17, we newly added Supplementary Fig. 13 (CER performance under 80 °C).
6. Page S20, we revised Supplementary Fig. 16 (equivalent circuit for CER and fitted Nyquist plots).
7. Page S21, we revised Supplementary Fig. 17 (CER LSV after 12 h of stability test under 25 °C).
8. Page S25, we revised Supplementary Fig. 21 (RRDE CA for CER selectivity for Cl⁻ concentration as low as 0.1 M).
9. Page S26, we newly added Supplementary Fig. 22b (iodometric titration for CER selectivity of PtNP/CNT for 0.55 M Cl⁻ concentration with neutral pH).
10. Page S27, we newly added Supplementary Fig. 23 (RRDE CA for CER selectivity for Pt₁/CNT_X samples.).
11. Page S29, we revised Supplementary Fig. 25 (adsorption free energy for plausible adsorbates on three Pt–N₄ sites).
12. Page S30, we revised Supplementary Fig. 26 (Pourbaix diagram of theoretical standard hydrogen electrode potential vs. pH for three Pt–N₄ sites in equilibrium with H⁺, Cl⁻, and H₂O).
13. Page S31, we newly added Supplementary Fig. 27 (Pourbaix diagram of theoretical standard hydrogen electrode potential vs. pH for three Pt–N₄ sites in equilibrium with H⁺ and H₂O).
14. Page S32, we newly added Supplementary Fig. 28 (atomic structures of α - and β -phases of PtO₂).
15. Page S36, we newly added Supplementary Fig. 32 (difference between adsorption energy of Pt²⁺ ion in the Pt–N₄ clusters and the chlorination energy of Pt²⁺ ion in the [PtCl₄]²⁻ complex).
16. Page S38, we newly added Supplementary Fig. 34 (experimental Tafel plot of Pt₁/CNT exhibiting two linear Tafel regions).
17. Page S39, we newly added Supplementary Fig. 35 (transition state free energy and absolute value of adsorption free energy for the reaction intermediate).
18. Page S40, we newly added Supplementary Fig. 36 (Free energy diagrams for OER on three Pt–N₄ sites at zero overpotential).

19. Page S43, we revised Supplementary Table 3 (comparison table of CER activity and operation condition of Pt₁/CNT catalyst and those of previously reported catalysts in acidic media)
20. Page S44, we newly added Supplementary Table 4 (charge transfer resistance of Pt₁/CNT_X determined by EIS fitting).
21. Page S45, we revised Supplementary Table 5 (absorption free energies of OH^{*}, O^{*}, and OOH^{*} on three Pt–N₄ sites).
22. Page S46, we newly added Supplementary Note 1 regarding the temperature- and Cl₂/Cl[–] concentration-dependent Nernst equations.
23. Page S51–52, we revised Supplementary Note 4 (Free energy diagram of model systems for CER and OER).
24. Page S53, we revised Supplementary Note 5 (Stability calculation of Pt–N₄ sites).
25. Page S54, we newly added Supplementary Note 6 (Full free energy diagram for CER over Pt₁/CNT).
26. In accordance with the changes, the order of equations was rearranged.
27. In accordance with the changes, the order of figures was rearranged.
28. Page S56–57, we added new Supplementary References (3,21–23).

Responses to the Comments of Reviewer 1

(Reviewer's Comments) This manuscript reports an atomically dispersed Pt-N4 sites for the chlorine evolution reaction with high activity and selectivity. A suite of experimental techniques is used to characterize the as prepared catalysts. In general, this is an interesting study. The paper could be accepted with the following minor revisions:

(Authors' Response) We truly appreciate the reviewer's endorsement of our paper.

1. The authors used different annealing temperature to prepare Pt₁/CNT_X samples, how the annealing temperature affect the activity and selectivity of the samples?

We appreciate the reviewer's comment regarding the effect of annealing temperatures on the CER activity and selectivity. The annealing temperature has a major effect on the activity of Pt₁/CNT_X ($X =$ annealing temperature). As we described in the original manuscript (**Supplementary Fig. 15**; original Supplementary Fig. 12), we demonstrated that the higher annealing temperature could induce higher catalytic activity for the CER. The enhanced catalytic activity of Pt₁/CNT_X for the CER with the increased annealing temperature could be explained by their charge transfer resistance (R_{ct}) trend (newly added **Supplementary Table 4**). The R_{ct} values were obtained by fitting Nyquist plots (**Supplementary Fig. 16c**). The lower value of R_{ct} indicates the facile charge transfer, which can lower the overpotential of electrochemical reaction on an electrode-electrolyte interface (**Supplementary Fig. 15**) according to the Butler-Volmer equation, thereby boosting catalytic activity.

Supplementary Fig. 15 | RRDE measurements of CER activities on Pt₁/CNT_X catalysts and unpyrolysed mixture of PtTPP precursor and CNT in acidic media with 1.0 M NaCl. The measurement conditions were Ar-saturated 0.1 M HClO₄, an electrode rotation speed of 1600 rpm, and a scan rate of 10 mV s⁻¹. Top panel indicates disk currents for CER. Bottom panel shows the corresponding ring current for Cl₂ reduction obtained on Pt ring electrode, whose potential was fixed at 0.95 V.

New Supplementary Table 4 | Charge transfer resistances (R_{ct}) of Pt₁/CNT_*X* catalysts determined by the EIS fitting. The error indicates the standard deviation from the corresponding fitting results to the three independent measurements.

Catalysts	R_{ct} (Ω)
Pt ₁ /CNT_mix	445.9 ± 2.7
Pt ₁ /CNT_500	119.3 ± 0.9
Pt ₁ /CNT_600	21.3 ± 0.6
Pt ₁ /CNT_700	14.8 ± 0.4
Pt ₁ /CNT_800	13.3 ± 0.5

Revised Supplementary Fig. 16c | Fitted Nyquist plots with low impedance range of Pt₁/CNT_*X* catalysts and unpyrolysed mixture of PtTPP precursor and CNT in acidic media with 1.0 M NaCl. The measurement conditions were Ar-saturated 0.1 M HClO₄ + 1.0 M NaCl, a fixed potential of 1.4 V, an electrode rotation speed of 1600 rpm, and a frequency range of 1–100,000 Hz. The empty circle and solid line in the Nyquist plots indicate experimental and fitting results, respectively.

Regarding the catalytic selectivity, all Pt₁/CNT_*X* catalysts exhibited similarly high selectivity for the CER (~95%, newly added **Supplementary Fig. 23**). As the selectivity of catalysts is an intrinsic property, this value was not affected by R_{ct} . We note that Pt₁/CNT_mix sample showed over 99 % of CER selectivity but its disk current decayed rapidly. The current decay was gradually relieved with the increased annealing temperature, indicating that highly active and selective Pt–N₄ sites were embedded in the CNT substrate more rigidly, which could be well explained by the van Veen model (newly added **Reference 47** and **Review Only Fig. 1**). The Pt₁/CNT_500 catalyst showed the lowest and fluctuating selectivity due to the possible dissolution of incompletely embedded precursor (**Supplementary Fig. 23b**).

New Supplementary Fig. 23 | Chronoamperograms of Pt₁/CNT_X catalysts measured by RRDE in acidic media with 1.0 M NaCl. a Pt₁/CNT_{mix}, b Pt₁/CNT₅₀₀, c Pt₁/CNT₆₀₀, d Pt₁/CNT₇₀₀, and e Pt₁/CNT₈₀₀. Measurement conditions were Ar-saturated 0.1 M HClO₄ and an electrode rotation speed of 1600 rpm. The potential of the Pt ring electrode was fixed at 0.95 V for Cl₂ reduction.

Review Only Fig. 1 | The schematic van Veen model of the structural change in a porphyrin molecule during heat treatment reproduced from Fig. 9 in *J. Phys. Chem. B* 106, 12993–13001 (2002).

Changes made:

- We have added **Supplementary Fig. 23** on page S27 of the revised supporting information.
 - We have added the following sentence to reflect **Supplementary Fig. 23** on page 11 of the revised manuscript:
New sentence: “The high CER selectivity of Pt–N₄ sites was also demonstrated with other Pt₁/CNT_X catalysts, suggesting that similarly high CER selectivity was obtained regardless of the annealing temperatures (**Supplementary Fig. 23**).”
 - We have added **Supplementary Table 4** on page S44 of the revised supporting information.
 - We have revised the following sentence on page 10 of the revised manuscript to reflect **Supplementary Table 4**:
Original sentence: “As the annealing temperature increased, the smaller semicircles of Pt₁/CNT_X catalysts were observed indicating lower values of R_{ct} and more facile electron transfer for CER.”
Revised sentence: “As the annealing temperature increased, the smaller semicircles (**Supplementary Fig. 16c**) and the lower R_{ct} of Pt₁/CNT_X catalysts were obtained (**Supplementary Table 4**), indicating that more facile charge transfer for CER is achievable through the higher annealing temperature.”
 - We have revised the following sentence on page 6 of the revised manuscript to reflect the van Veen model of **New Reference 47**:
Original sentence: “Deconvoluted N 1s XPS spectra of the series of Pt₁/CNT_X samples (**Supplementary Fig. 7**) indicated that a large portion of N atoms maintained its chemical state of the initial PtTPP precursor (Pt–N_x coordination) during the annealing treatments.”
Revised sentence: “Deconvoluted N 1s XPS spectra of the series of Pt₁/CNT_X samples (**Supplementary Fig. 9**) indicated that a large portion of N atoms maintained its chemical state of the PtTPP precursor with a peak at 399.2 eV (Pt–N_x coordination) during the high-temperature annealing. With the samples annealed above 600 °C, peaks for pyridinic N (398.1 eV), pyrrolic N (400.2 eV), and graphitic N (400.7 eV) species emerged, which could be ascribed to the decomposition of surrounding carbons following the van Veen model⁴⁷.”
 - We have added the following literature in the revised manuscript.
47. Bouwkamp-Wijnoltz, A. L. *et al.* On active-site heterogeneity in pyrolyzed carbon-supported iron porphyrin catalysts for the electrochemical reduction of oxygen: an in situ Mössbauer study. *J. Phys. Chem. B* **106**, 12993–13001 (2002).
2. The STEM images of other Pt₁/CNT_X samples also need to be provided.

We are grateful to the reviewer’s comment to provide STEM images of Pt₁/CNT_X. HAADF-STEM images of Pt₁/CNT_500, 600, and 800 samples (newly added **Supplementary Fig. 6**) show that ultrasmall white dots were uniformly dispersed on the CNT substrate, indicating the formation of atomically dispersed Pt species, just like Pt₁/CNT. The STEM images strengthen our claim that the local structure of Pt–N₄ sites is preserved despite the changes in annealing temperatures.

New Supplementary Fig. 6 | HAADF-STEM images of Pt₁/CNT_*X* catalysts (*X*=annealing temperature). a Pt₁/CNT_500, b Pt₁/CNT_600, and c Pt₁/CNT_800. Scale bars: 2 nm.

Changes made:

- We have added **Supplementary Fig. 6** on page S10 of the revised supporting information.
- We have added the following sentence to reflect **Supplementary Fig. 6** on page 6 of the revised manuscript:
New sentence: “The HAADF-STEM images of Pt₁/CNT_*X* (*X*=500, 600, and 800 °C) (**Supplementary Fig. 6**) clearly indicate the formation of atomically dispersed Pt sites regardless of annealing temperatures, suggesting that the annealing temperature up to 800 °C cannot induce agglomeration of Pt.”
- We have revised the following sentence on page 9 of the revised manuscript to emphasize the thermal stability of Pt–N₄ sites, which is further supported by the HAADF-STEM images of Pt₁/CNT_*X*:
Original sentence: “Thus, the high-temperature annealing enhanced the electrical conductivity between the Pt–N₄ and the CNT substrate, which could play a critical role in enhancing the catalytic activity^{45–48}.”
Revised sentence: “Overall, the high-temperature annealing concomitantly enhanced the structural integrity and electrical conductivity between the Pt–N₄ sites and the CNT^{47–50,54,55}, while preserving the local structure and chemical states of Pt–N₄ sites.”

3. The authors claim that the Pt₁/CNT has high CER selectivity, but based on Supplementary Fig. 14, 15, and 17, PtNP/CNT even demonstrated better selectivity than Pt₁/CNT. Why PtNP/CNT also exhibits excellent selectivity?

We appreciate the reviewer's keen insight into the CER selectivity of PtNP/CNT. The CER takes place above 1.36 V (vs. SHE), in which the oxygen evolution reaction (OER) also occurs. Hence, the OER is generally regarded as the major side reaction for the CER. Pt is one of the least active catalysts for the OER. In acidic media without Cl⁻ ions, polycrystalline Pt showed high overpotential for the OER [1.95 V vs RHE for 5 mA cm⁻², *ChemCatChem* **6**, 2219–2223 (2014), **Review Only Fig. 2a**], and Pt NPs showed even higher overpotential than that of the polycrystalline Pt [*ACS Catal.* **2**, 1765–1772 (2012), **Review Only Fig. 2b**]. These data suggest that Pt catalysts can have intrinsically high selectivity for CER over OER. Our selectivity data indicate that the PtNP/CNT showed slightly higher selectivity (2–3%) than Pt₁/CNT in acidic media. Interestingly, in neutral pH with 0.55 M Cl⁻, PtNP/CNT showed decreased CER selectivity of 87.2% whereas Pt₁/CNT nearly preserved high selectivity with 93.9% (**Revised Supplementary Fig. 22b**).

We suppose that the thermodynamic and kinetic effects of high pH could affect more significantly than Cl⁻ concentration on the CER selectivity of PtNP. However, further insights into the selectivity trend over the PtNP/CNT and Pt₁/CNT catalysts require a more detailed study, which will be conducted as a future work.

In addition, we mistakenly omitted the selectivity data of PtNP/CNT in **Supplementary Fig. 21**. We have therefore revised **Supplementary Fig. 21**. We thank again for the reviewer's careful attention and apologize for our mistake.

Review Only Fig. 2 | a The OER activity of polycrystalline Pt and other metals. Reproduced from Fig. 2a in *ChemCatChem* **6**, 2219–2223 (2014). b The OER activity comparison of polycrystalline bulk and nanoparticles of Pt, Ir, and Ru. Reproduced from Fig. 6 in *ACS Catal.* **2**, 1765–1772 (2012).

Revised Supplementary Fig. 22 | Chronoamperograms for iodometric titration of Cl₂ product on Pt₁/CNT, PtNP/CNT, and DSA catalysts in neutral media with 0.55 M NaCl. a Pt₁/CNT, b PtNP/CNT, and c DSA catalysts. The electrolytes were Ar-saturated.

Revised Supplementary Fig. 21 | Chronoamperograms of Pt₁/CNT and PtNP/CNT catalysts measured by RRDE in acidic media with 0.5 M and 0.1 M NaCl. The CER selectivity of Pt₁/CNT in a 0.5 M NaCl and b 0.1 M NaCl, and PtNP/CNT in c 0.5 M NaCl and d 0.1 M NaCl. The measurement conditions were Ar-saturated 0.1 M HClO₄ and an electrode rotation speed of 1600 rpm. The potential of the Pt ring electrode was fixed at 0.95 V for Cl₂ reduction.

Changes made:

- We have revised the **Supplementary Fig. 22** on page S26 of the revised supporting information.
- We have revised the **Supplementary Fig. 21** on page S25 of the revised supporting information.

4. Can the loading mass of Pt be tuned to achieve better activity and selectivity?

We are grateful to the reviewer's insightful comment on the impact of Pt loading on the CER activity and selectivity. In the original version, we presented the Pt₁/CNT catalyst with 3 wt% nominal Pt loading, and extensively investigated its physicochemical and catalytic properties. In response to the reviewer's comment, we have prepared Pt₁/CNT catalysts with different nominal Pt loadings of 1 wt%, 5 wt%, and 10 wt% (denoted as Pt₁/CNT_X%; X=nominal Pt loading) via the same procedure used for Pt₁/CNT. The actual Pt contents in the samples were analysed by ICP-OES and summarised in **Review Only Table 1**, which suggests that the actual Pt loadings in Pt₁/CNT_X% catalysts were proportional to their nominal Pt loadings. All Pt₁/CNT_X% catalysts comprised atomically dispersed Pt sites on CNT as shown in high-magnification HAADF-STEM

images (**Review Only Fig. 3a, c, and e**). The density of white dots, corresponding to atomically dispersed Pt site, was proportional to their nominal Pt loadings. However, in low-magnification HAADF-STEM images, Pt nanoparticles were observed in Pt₁/CNT_5% and Pt₁/CNT_10% (**Review Only Fig. 3d and f**). In summary, atomically dispersed Pt sites are exclusively formed with nominal Pt loadings of 1 and 3 wt%. However, with Pt loadings above 3 wt%, Pt nanoparticles start to form via dissociation of PtTPP precursor and subsequent aggregation of fragmented Pt species.

Next, we investigated the CER activities of Pt₁/CNT_X% catalysts, which were in general very high, all surpassing those of PtNP/CNT and DSA. Among Pt₁/CNT_X% catalysts, Pt₁/CNT_3% showed the highest activity, followed by Pt₁/CNT_5% > Pt₁/CNT_1% ≈ Pt₁/CNT_10% (**Review Only Fig. 4a**). The better activity of Pt₁/CNT_3% than Pt₁/CNT_1% can be interpreted with higher density of Pt-N₄ sites in the former. The superior activity of Pt₁/CNT_3% than higher Pt loading catalysts can be ascribed to the contribution of less active Pt NPs in Pt₁/CNT_5% and Pt₁/CNT_10% catalysts. The selectivity of Pt₁/CNT_X% was also generally very high (**Review Only Fig. 4b**).

We are grateful again to the reviewer's insightful suggestion to optimize the amount of Pt. We will investigate more detailed analyses of catalytic trends of Pt-loading-controlled samples as a future work.

Review Only Fig. 3 | TEM images of Pt₁/CNT catalysts prepared with different nominal Pt loadings (Pt₁/CNT_X%). High-magnification images of a Pt₁/CNT_1%, c Pt₁/CNT_5%, and e Pt₁/CNT_10%. Low-magnification images of b Pt₁/CNT_1%, d Pt₁/CNT_5%, and f Pt₁/CNT_10%. Scale bars: 2 nm in a, c, e and 20 nm in b, d, f.

Review Only Table 1 | ICP-OES results of Pt₁/CNT_X% samples prepared with different nominal loadings of PtTPP.

Catalysts	Actual Pt (wt%)
Pt ₁ /CNT_1%	0.85
Pt ₁ /CNT_3%	2.7
Pt ₁ /CNT_5%	4.0
Pt ₁ /CNT_10%	8.7

Review Only Fig. 4 | Electrochemical CER performances of Pt₁/CNT_X% catalysts. a CER polarisation curves of Pt₁/CNT_X% catalysts in Ar-saturated 0.1 M HClO₄ + 1.0 M NaCl at a rotation speed of 1600 rpm. Chronoamperograms for measuring CER selectivity of b Pt₁/CNT_1%, c Pt₁/CNT_3%, d Pt₁/CNT_5%, and e Pt₁/CNT_10% at a rotation speed of 1600 rpm. The potential of the Pt ring electrode was fixed at 0.95 V for Cl₂ reduction.

5. The high activity Pt₁/CNT could be well preserved after durability tests?

We appreciate the reviewer's comment. We added the CER polarization curves of Pt₁/CNT after the 12 h of stability test (**Supplementary Fig. 17**). In comparison with the initial LSV, the Pt₁/CNT catalyst maintained 75 % of initial current density after 12 h. This is close to the 72 % of initial current recorded by the 12 h of stability test (**Fig. 2c**).

New Supplementary Fig. 17 | CER polarization curves of Pt₁/CNT catalyst loaded on a carbon paper before and after 12 h of stability test at 10 mA cm⁻² at a scan rate of 10 mV s⁻¹ (Fig. 1c). An electrolyte was stirred at a rotation speed of 300 rpm. The electrolyte was replaced with a fresh electrolyte after the stability test.

Changes made:

- We have added **Supplementary Fig. 17** on page S21 of the revised supporting information.
- We have added the following sentence to reflect the newly added **Supplementary Fig. 17** on page 10 of the revised manuscript:
New sentence: “After the long-term stability test, the polarization curve of Pt₁/CNT exhibited 75% of its initial current at 1.41 V vs. RHE (**Supplementary Fig. 17**).”

6. Line 87-88, 90. The unit of “centigrade” is missing. This problem could be found in the whole manuscript.

We appreciate the reviewer's comment to point out our mistake. It appears that during the conversion from Microsoft Word doc format file to pdf file “centigrade” mark disappeared. We have revised and checked all errors in the revised manuscript and supporting information.

7. The locations of Pt²⁺ peaks are different in Figure S4. Please explain.

We appreciate the reviewer's comment regarding XPS analyses. The binding energy of metal species is sensitive to its surroundings or ligands. Deconvoluted Pt 4f XPS spectrum of PtNP/CNT (**Supplementary Fig. 5**) shows its Pt²⁺ 4f_{7/2} peak at 72.3 eV,

which was fitted using the value of PtO (72.4 eV) [*Anal. Chem.* **47**, 586–588 (1975)]. However, Pt₁/CNT did not comprise Pt NPs and maintained the local structure of Pt–N₄, which is similar to its precursor, Pt(II) meso-tetraphenylporphine (PtTPP). We therefore fitted the Pt²⁺ 4f_{7/2} peak of the Pt₁/CNT spectrum based on the reported value of Pt(II) porphyrin (73.3 eV, **Supplementary Reference 1**), resulting in 73.1 eV as a peak position of Pt²⁺ 4f_{7/2} for Pt₁/CNT catalyst.

Supplementary Fig. 5 | Deconvoluted Pt 4f XPS spectra of Pt₁/CNT and PtNP/CNT catalysts. The spin-orbit splitting and the area ratio for 4f_{5/2} (dashed lines) and 4f_{7/2} (solid lines) peaks are 3.34 eV and 3:4, respectively. The peak of Pt 4f_{7/2} in the spectrum of Pt₁/CNT catalyst was observed at 73.1 eV, which is close to the value of Pt porphyrin in a previous report.

8. The equivalent circuit should be displayed and explained in Figure S13.

We appreciate again the reviewer’s suggestion. We have drawn the equivalent circuit (**Revised Supplementary Fig. 16a**) following the previous work (Newly added **Reference 65**). Thanks to this comment, we could quantify the charge transfer resistance (R_{ct}) values of the catalysts, which is listed in newly added **Supplementary Table 4**.

Revised Supplementary Fig. 16 | Nyquist plots and corresponding fitting results of Pt₁/CNT_X catalysts and unpyrolysed mixture of PtTPP precursor and CNT in acidic media with 1.0 M NaCl. a Equivalent circuit for the EIS fitting. b Nyquist plots with high impedance range. c Nyquist plots with low impedance range. The measurement conditions were Ar-saturated 0.1 M HClO₄ + 1.0 M NaCl, a fixed potential of 1.4 V, an electrode rotation speed of 1600 rpm, and a frequency range of 1–100,000 Hz. The empty circle and solid line in the Nyquist plots indicate experimental and fitting results, respectively.

New Supplementary Table 4 | Charge transfer resistances (R_{ct}) of Pt₁/CNT_X catalysts determined by the EIS fitting. The error indicates the standard deviation from the corresponding fitting results to the three independent measurements.

Catalysts	R_{ct} (Ω)
Pt ₁ /CNT _{mix}	445.9 ± 2.7
Pt ₁ /CNT ₅₀₀	119.3 ± 0.9
Pt ₁ /CNT ₆₀₀	21.3 ± 0.6
Pt ₁ /CNT ₇₀₀	14.8 ± 0.4
Pt ₁ /CNT ₈₀₀	13.3 ± 0.5

Changes made:

- We have revised **Supplementary Fig. 16** on page S20 of the revised supporting information.
- We have added the following sentences on page 23 of the revised manuscript to provide the details of EIS fitting and the corresponding equivalent circuit.

New sentences: “The Nyquist plots were fitted to obtain R_{ct} using the ZView software based on the equivalent circuit shown in **Supplementary Fig. 16a**, which was built on the previous report⁷⁰. A pseudocapacitance (C_p) and its coupled resistance (R_p) originate from the adsorption and desorption of Cl^- intermediates⁷⁰. Due to the uncertain nature of the electrode-solution interface, the double layer capacitance (C_{dl}) and C_p were fitted with constant phase elements (CPEs) instead of an ideal capacitor. The charge transfer resistances of the catalysts are summarized in **Supplementary Table 4.**”

- We have revised the following sentence on page 9 of the revised manuscript to reflect **Revised Supplementary Fig. 16:**

Original sentence: “To clarify the annealing temperature-dependent activity trend in Pt₁/CNT_X catalysts, the Nyquist plots at the potential of 1.4 V (vs. RHE) were obtained to assess the charge transfer resistance (R_{ct} , Supplementary Fig. 13).”

Revised sentence: “To clarify the annealing temperature-dependent activity trend in Pt₁/CNT_X catalysts, the Nyquist plots at the potential of 1.4 V (vs. RHE) were obtained and fitted with an equivalent circuit to assess the charge transfer resistance (R_{ct} , **Supplementary Fig. 16**).”

- We have added the following literature in the revised manuscript.

70. Silva, J. F., Dias, A. C., Araújo, P., Brett, C. M. A. & Mendes, A. Electrochemical cell design for the impedance studies of chlorine evolution at DSA[®] anodes. *Rev. Sci. Instrum.* **87**, 085113 (2016).

9. As displayed in Figure S12, the CER performance of 700 and 800 showed almost no difference, which is contradicted with the EIS shown in Figure S13. Please explain.

We thank the reviewer for pointing out why the Pt₁/CNT_700 and Pt₁/CNT_800 catalysts showed the similar activity, which is contrasting with their EIS results. As referred in the response to the Comment 1, we have newly fitted the EIS results with the corresponding equivalent circuit. The fitted R_{ct} values revealed that the difference between the Pt₁/CNT_700 and Pt₁/CNT_800 catalysts is only marginal (1.5 ohm, newly added **Supplementary Table 4**). We therefore suppose that the minimum charge transfer resistance was achieved by the annealing at 700 °C, which could be sufficient to overcome the free energy barrier of CER at an electrode-electrolyte interface. We thank again the reviewer’s insight to match the equivalent circuit to our electrocatalytic system.

New Supplementary Table 4 | Charge transfer resistances (R_{ct}) of Pt₁/CNT-*X* catalysts determined by the EIS fitting. The error indicates the standard deviation from the corresponding fitting results to the three independent measurements.

Catalysts	R_{ct} (Ω)
Pt ₁ /CNT_mix	445.9 ± 2.7
Pt ₁ /CNT_500	119.3 ± 0.9
Pt ₁ /CNT_600	21.3 ± 0.6
Pt ₁ /CNT_700	14.8 ± 0.4
Pt ₁ /CNT_800	13.3 ± 0.5

10. As shown in Line 234-236, the calculated model of PtNP/CNT is based on “(110) surface of disordered PtO₂”. Hence it is disordered PtO₂, the rationality of atomic arrangement should be explained. And is it able to conclude that PtNP/CNT shows poor selectivity and efficiency towards CER is because of the oxidising of Pt? Hence it is suggested to do further calculations on pure Pt model.

We appreciate the reviewer’s comment regarding the rationality of atomic arrangement for PtNP/CNT. As we referred in the original manuscript, we chose the most stable (110) surface of distorted rutile PtO₂ as a representing model system for PtNP/CNT, following the previous report that Pt nanoparticles exist in an oxidized form near 1.36 V vs SHE (**Revised Reference 61**). However, as the reviewer pointed out, the atomic structure of PtO₂ under CER condition should have been identified to rationalize our calculation models. There exist two representative phases for PtO₂, referred to CdI₂-type α -PtO₂ (space group *P-3m1*) and (distorted rutile) CaCl₂-type β -PtO₂ (space group *Pnnm*), respectively [*Phys. Rev. B* **84**, 100102 (2011)]. By calculating the Gibbs free energy of formation (ΔG_f) for each phase, we found that β -PtO₂ phase was thermodynamically more stable, implying that β -PtO₂ predominantly could exist in the PtNP/CNT under CER condition (**New Supplementary Fig. 28**).

New Supplementary Fig. 28 | Atomic structures of α - and β -phases of PtO₂. Crystalline system, space group, and Gibbs free energy of formation (ΔG_f) for each phase are given below the unit cells. The ΔG_f 's are determined with respect to the bulk Pt and molecular oxygen (i.e., $\Delta G_f = G(\text{PtO}_2) - G(\text{Pt-bulk}) - G(\text{O}_{2(g)})$). Colour legends – Pt: dark-blue; O: red.

In addition, we would like to note that PtNP/CNT showed slightly higher CER selectivity (2-3%) than that of Pt₁/CNT as referred in the Comment 3 (**Revised Supplementary Fig. 21**), indicating that oxidising of Pt might not mitigate the selectivity of CER. To further resolve the reviewer's concerns regarding the activity, we calculated the free energy diagrams for CER on pure Pt(111) surface by considering two possible pathways mediated by Cl* or ClO* intermediates (**Review Only Fig. 5**). As a result, both cases exhibited very large thermodynamic overpotentials for CER (i.e., $\eta_{\text{CER(TD)}} = \Delta G/e$, where $\Delta G = \Delta G_{\text{Cl}^*}$ or $(\Delta G_{\text{ClO}^*} - \Delta G_{\text{O}^*})$, respectively) exceeding 0.8 V. It indicated that pure Pt might not be active for CER. Considering that PtO₂(110) surface, which represents the PtNP/CNT, exhibited still better CER activity (i.e., $\eta_{\text{CER(TD)}} = 0.2$ V) compared to that of pure Pt(111) surface, we anticipated that Pt oxidation would not be the origin of hampering the CER activity.

Revised Supplementary Fig. 21 | Chronoamperograms of Pt₁/CNT and PtNP/CNT catalysts measured by RRDE in acidic media with 0.5 M and 0.1 M NaCl. The CER selectivity of Pt₁/CNT in a 0.5 M NaCl and b 0.1 M NaCl and PtNP/CNT in c 0.5 M NaCl and d 0.1 M NaCl. The measurement conditions were Ar-saturated 0.1 M HClO₄ and an electrode rotation speed of 1600 rpm. The potential of the Pt ring electrode was fixed at 0.95 V for Cl₂ reduction.

Review Only Fig. 5 | DFT calculations for CER activity on pure Pt(111) surface. a Model systems for Pt(111) surface with possible reaction intermediates of Cl*, O*, and ClO* species. Colour legends – Pt(top-most): dark-blue; Pt(second-most from top): black; Pt(third-most from top): gray; Pt(bottom-most): light-gray; Cl: yellow-green; O: red. b,c Free energy diagrams for CER on Pt(111) surface including b Cl* and c ClO* intermediates.

Changes made:

- We have added **Supplementary Fig. 28** on page S32 of the revised supporting information.
- We have revised the following sentence on page 12 of the revised manuscript to reflect **Supplementary Fig. 28**.

Original sentence: “For modelling the PtNP/CNT (Supplementary Note 1), we chose the most stable (110) surface of disordered rutile PtO₂ as a representing model system for PtNP/CNT (Fig. 4b), following the report that Pt nanoparticles exist in an oxidised form near 1.36 V.”

Revised sentence: “For modelling the PtNP/CNT (**Supplementary Note 2**), we chose the (110) surface of distorted rutile PtO₂ (which was found as the thermodynamically most stable phase, **Supplementary Fig. 28**) as a representing model system for PtNP/CNT (**Fig. 4b**), following the report that Pt nanoparticles exist in an oxidised form near 1.36 V vs. SHE⁶¹.”

11. As shown in Line 234-236, the calculated model of PtNP/CNT is based on “(110) surface of disordered PtO₂”. Hence it is disordered PtO₂, the rationality of atomic arrangement should be explained. And is it able to conclude that PtNP/CNT shows poor selectivity and efficiency towards CER is because of the oxidising of Pt? Hence it is suggested to do further calculations on pure Pt model.

We found that the comment 11 is identical to the comment 10. We have replied this issue at the comment 10.

Responses to the Comments of the Reviewer 2

In this manuscript, the authors present a detailed analysis of synthesis and characterization of a Pt-based single atom catalyst shown to exhibit superior activity and selectivity for the chlorine evolution reaction (CER) under relevant conditions. This is very impressive and high impact work that, aside from a few points, is quite clear and accurate. A few things that should be addressed prior to publication:

We are truly grateful to the reviewer's high evaluation of our paper.

1. Some conversion issues on lines 87, 88, and 90 (degrees C?) continues throughout manuscript.

We appreciate the reviewer's comment to point out our mistake. It appears that during the conversion from Microsoft Word doc file format to pdf file "centigrade" mark disappeared. We have revised and checked all errors in the revised manuscript and supporting information.

2. In SI Fig. 1, frequency is given but please also provide out of how many counts and over what relevant area to give better understanding of statistics and how well represented the materials are in frames c and f.

We thank the reviewer's constructive suggestion to provide the exact number of counted dots and the specific area for our counting. We counted total 275 white dots from two HAADF-STEM images of Pt₁/CNT (**Review Only Fig. 6a and b**), and 295 particles from PtNP/CNT (**Review Only Fig. 6c and d**). We have added total count numbers in size distribution histograms in **Supplementary Fig. 1c and f** of the revised supplementary information.

Review Only Fig. 6 | HAADF-STEM images for particle size distribution histograms. a High-magnification and b Low-magnification images of Pt₁/CNT catalyst. c High-magnification and d Low-magnification images of PtNP/CNT catalyst. Scale bars: 2 nm in a, b, 10 nm in c, and 20 nm in e.

Revised Supplementary Fig. 1 | Additional TEM images and Pt particle size distribution histograms of Pt₁/CNT and PtNP/CNT catalysts. a Low-magnification HR-TEM image, b HAADF-STEM image, and c Pt particles size distribution histogram of Pt₁/CNT catalyst. d Low-magnification HR-TEM image, e HAADF-STEM image, and f Pt particles size distribution histogram of PtNP/CNT catalyst. Scale bars: 20 nm in a, d and 2 nm in b, e.

Changes made:

- We have inserted the exact number of counted particles in **Supplementary Fig. 1c and f** on page S5 of the revised supporting information.
- We have shortened the following sentence on page 18 of the revised manuscript for specific description:
Original sentence: “Size distributions of Pt particles were analysed using a Gatan Microscopy Suite 3 Software by counting at least 200 particles.”
Revised sentence: “Size distributions of Pt particles were analysed using a Gatan Microscopy Suite 3 Software.”
- We have added a new sentence on page 18 of the revised manuscript:
New sentence: “Total 275 and 295 particles were counted from two HAADF-STEM images of Pt₁/CNT (17 nm × 17 nm and 24 nm × 24 nm) and PtNP/CNT (144 nm × 144 nm and 288 nm × 288 nm), respectively (**Supplementary Figs. 1c and f**).”

3. XAS implies Pt-N coordination but EELS could show this explicitly (see for instance, Chung, et al., Science 2017 on PGM-free ORR electrocatalysts). This could provide further input on average coordination number.

We are grateful to the reviewer's suggestion on the possibility of EELS data for providing information on the Pt-N coordination number. In response, we obtained EELS spectrum of Pt₁/CNT catalyst, which was acquired from a limited area of ~5 Å² in HAADF-STEM image that includes an atomically dispersed Pt site (newly added **Supplementary Fig. 2**). From EELS spectrum, we could identify the presence of Pt and N species. However, due to the weak signal of N, determining the coordination number of Pt-N₄ centre was rather difficult. However, as a future work, we will make efforts to obtain EELS spectra with better resolution, so that, in combination with EXAFS and other characterization data, deterministic quantification of coordination number around the central metal can be made possible. We thank again for the reviewer's valuable suggestion.

New Supplementary Fig. 2 | EELS spectrum of Pt₁/CNT. a High-magnification HAADF-STEM image. Scale bar: 1 nm. b EELS spectrum of Pt₁/CNT taken on the red dotted box (~5 Å²) in a.

Changes made:

- We have added **Supplementary Fig. 2** on page S6 of the revised supporting information.
- We have added new sentences on page 5 of the revised manuscript:

New sentences: “The electron energy loss spectrum (EELS) of Pt₁/CNT, obtained from a limited area (~5 Å²) comprising an atomically dispersed Pt site in the HAADF-STEM image (**Supplementary Fig. 2**), indicated the presence of Pt and N.”

4. It is possible to derive the N 1S spectra for specific structures from DFT which could be beneficial for corroborating nature of the active site via DFT beyond just activity comparisons. See for instance Artyushkova, et al., ChemComm 2013.

We appreciate the reviewer's suggestion on the use of DFT calculations for understanding the nature of the active sites.

In an elegant XPS study by Artyushkova and co-workers [*Chem. Commun.* **49**, 2539–2541 (2013)], they combined carefully measured XPS spectra with DFT calculations of binding energy (BE) shifts of N 1s spectra, and compared and evaluated DFT predicted BE shifts to experimentally curve-fitted spectra of model compounds. Finally, they used the information obtained from DFT calculations as input for curve-fitting of in situ XPS spectra, which could lead to interpretation of XPS spectra to

the point of distinguishing Me-N₂ and Me-N₄ species. This is truly a remarkable work that can provide very details of otherwise inaccessible active site structure. One important conclusion of this work was the shift of peak position for metal-N_x sites from those of pyridinic N and pyrrolic N species.

In response, we measured and compared XPS spectra of unpyrolysed TPP (without metal centre), unpyrolysed PtTPP, and Pt₁/CNT_700 (**Review Only Fig. 7**). While XPS spectrum of TPP showed two peaks at 398.1 eV and 400.1 eV for pyridinic N and pyrrolic N species, respectively, PtTPP showed a single XPS peak at 399.2 eV corresponding to Pt-N_x species. This result is in accordance with that of Artyushkova and co-workers with clear shift of Pt-N_x peak from those of pyridinic N and pyrrolic N species. Pt₁/CNT_700 showed three deconvoluted peaks for pyridinic N, Pt-N_x, and graphitic N species. In addition, following Artyushkova's work, we have newly deconvoluted the N 1s XPS spectra of Pt₁/CNT_X catalysts by adding the pyrrolic N peak (**Revised Supplementary Fig. 9a**).

We note that our XPS spectra show inferior resolution compared to those of Artyushkova and co-workers, hampering in-depth XPS peak analyses and deciphering of active sites. As a future work, we will follow the method of Artyushkova and co-workers to deduce the active site structure for the CER.

Review Only Fig. 7 | Deconvoluted N 1s XPS spectra of Pt₁/CNT catalyst, PtTPP, and TPP precursor.

Revised Supplementary Fig. 9a | Deconvoluted N 1s XPS spectra of Pt₁/CNT_X catalysts and PtTPP precursor.

Changes made:

- We have revised the **Supplementary Fig. 9a** on page S13 of the revised supporting information.
 - We have added a following literature in the revised manuscript.
48. Artyushkova, K. et al. Density functional theory calculations of XPS binding energy shift for nitrogen-containing graphene-like structures. *Chem. Commun.* **49**, 2539–2541 (2013).
 - We have added the following sentence on page 7 of the revised manuscript:
New sentence: “We note that the position of Pt²⁺-ligated N species was clearly different from those of pyridinic N and pyrrolic N species, which is consistent with previously reported, heat-treated M–N/C catalysts comprising atomically dispersed Fe–N_x or Co–N_x sites^{47–50}.”
5. Further details of the MWCNT would be beneficial to the reader (average diameter and number of walls, in particular).

We thank the reviewer’s comment to provide the details of MWCNT. The average diameter and length of MWCNT are around 10 nm (5–15 nm) and 10 μm, respectively, which were provided by the manufacturer (Carbon Nano-material Technology Co., Korea). The number of walls was not provided by manufacturer. We have therefore counted the wall numbers in four different bright-field transmission electron microscope (BF-TEM) images of CNT (**Review Only Fig. 8**), which varied from 8 to 27 with an average value of 16.

Review Only Fig. 8 | Additional BF-STEM images for counting the number of walls of MWCNT. a,b BF-STEM images of Pt₁/CNT catalyst. c,d BF-STEM images of PtNP/CNT catalyst. Scale bars: all 5 nm.

Changes made:

- We have revised the following sentence on page 16 of the revised manuscript:
Original sentence: “Multi-walled carbon nanotubes (CNT, MR 99) were purchased from Carbon Nano-material Technology Co. (Korea).”
Revised sentence: “Multi-walled carbon nanotubes (CNT, MR 99, average diameter of 10 nm, average length of 10 μm, and average 16 walls) were purchased from Carbon Nano-material Technology Co. (Korea).”
6. Line 152: “all sites were considered active” – Which method for determining final amount of Pt was used in this calculation? Please be clear as identification of number of sites in SAC materials is an important issue especially for comparison across materials.

We appreciate the reviewer’s constructive suggestion about the quantification of Pt. In the original manuscript, we quantified the amount of Pt by inductively coupled plasma optical emission spectroscopy (ICP-OES) and used the obtained data for the turnover frequency (TOF) calculation. We have specified the equation for calculating the actual amount (mol) of Pt in Pt₁/CNT catalyst layer used for the electrochemical measurements.

Changes made:

- We have revised the following sentences on page 8 of the revised manuscript:
Original sentence: “For the calculation of TOFs, all Pt sites were considered as active sites for Pt₁/CNT, whereas CO stripping method was used to calculate electrochemically active surface sites for PtNP/CNT (**Supplementary Fig. 9**).”
Revised sentence: “For the calculation of TOFs, all Pt sites in the catalyst layer were considered as active sites for Pt₁/CNT and calculated as 3.46 nmol based on the quantification of Pt by ICP-OES analysis (see Equations 6 and 7 in **Methods** and **Supplementary Table 1**).”

- We have added the following sentence on page 8 of the revised manuscript:
New sentence: “The CO stripping method was used to calculate electrochemically active surface sites for PtNP/CNT, and the calculated value was 0.96 nmol (see Equations 5 and 7 in **Methods** and **Supplementary Fig. 11**).”

- We have added the following sentences on page 22 of the revised manuscript:
New sentences: “The n of Pt₁/CNT catalyst was calculated by the following equation:

$$n \text{ (mol)} = \frac{m_{\text{Pt}}}{M_{\text{Pt}}} = \frac{w_{\text{Pt}} \cdot \rho_{\text{cat}} \cdot V_{\text{cat}}}{100 \cdot M_{\text{Pt}}} \quad (6)$$

where m_{Pt} is the amount of Pt in the catalyst layer, w_{Pt} is the weight percent of Pt in the Pt₁/CNT (2.7 wt%), ρ_{cat} is the mass concentration of Pt in the catalyst ink (4.17 g L⁻¹), V_{cat} is the volume of the loaded catalyst ink (6 μL), and M_{Pt} is the molar mass of Pt (195.084 g mol⁻¹).”

7. In the DFT study only planar structures are considered. Given that these are suggested to be hosted in the CNT which is radially strained, shouldn't a radially strained model be applied? This has previously been shown to make a significant impact on binding energies for other electrochemical reactions (ORR in particular) and thus, likely, the activity descriptor used in this study. This must be addressed.

We appreciate the reviewer's comment regarding the strain effect on the CER activity for Pt₁/CNT. We calculated the adsorption free energy of Cl* species (ΔG_{Cl^*}) on radially strained models for PtN₄C₁₂, which was identified as the most plausible active adsorbate structure from our investigations (**Review Only Fig. 9a**). When the radial strain (ϵ) was applied by 10 %, the structural instability linearly increased (**Review Only Fig. 9b**), but the ΔG_{Cl^*} negligibly changed (i.e., $\Delta(\Delta G_{\text{Cl}^*}) < 0.02$ eV, **Review Only Fig. 9c**). Based on the results, we expect that the effect of radial strain on the CER activity of Pt₁/CNT is negligible.

In addition, the maximum radial strain will be $\epsilon = 1.2\%$ by combining with the width of PtN₄C₁₂ (13.35 Å) and the minimum diameter of MWCNT (5 nm) mentioned in the response to Comment 7. If we only consider the active site (Pt–N₄), the maximum radial strain becomes much smaller, $\epsilon = 0.1\%$. The maximum diameter of a Pt–N₄ site can be estimated as long as 4.01 Å according to the L₃-edge EXAFS fitting results of Pt₁/CNT (**Supplementary Table 2**), which would be 9.2° bent on the surface of the thinnest MWCNT.

Review Only Fig. 9 | DFT calculations for the effect of radial strain (ϵ) on the Pt₁/CNT. a Model systems for Pt₄C₁₂ clusters under applied ϵ , ranging from 0% to 10%. The end-to-end distances for $\epsilon = 0\%$ (pristine) and $\epsilon = 10\%$ are indicated below. The red dotted lines and circles indicate the terminal hydrogen atoms at both ends. Colour legends – Pt: dark-blue; H: white; N: pink; Cl: yellow-green. b Relative total energy of Pt₄C₁₂ clusters under applied radial strain (%), ranging from 0% to 10%. c Adsorption free energy of Cl* species (ΔG_{Cl^*}) for Pt₄C₁₂ clusters under applied radial strain (%), ranging from 0% to 10%.

Supplementary Table 2 | Summary of EXAFS fitting parameters of Pt₁/CNT catalyst, PtTPP precursor, PtNP/CNT catalyst, and Pt foil.

Sample	Shell	CN	R (Å)	σ^2 (10^{-3} \AA^{-2})	ΔE_0 (eV)	R factor (%)
Pt ₁ /CNT	Pt–N	3.9 ± 0.7	2.00 ± 0.01	3.44 ± 1.49	12.51 ± 2.13	1.8
	Pt–C	4.0 ± 2.7	3.00 ± 0.03	5.03 ± 6.26		
PtTPP	Pt–N	4*	2.01 ± 0.01	1.09 ± 0.46	11.44 ± 1.80	0.9
	Pt–C	8*	3.04 ± 0.01	1.89 ± 0.99		
	\angle Pt–C	16*	3.30 ± 0.04	1.28 ± 6.78		

PtNP/CNT	Pt-C	3.3 ± 0.9	2.04 ± 0.02	5.47 ± 2.48	4.25 ± 3.07	2.0
	Pt-Pt (1 st)	7.7 ± 3.7	2.70 ± 0.02	17.33 ± 4.23		
Pt foil	Pt-Pt (1 st)	12	2.77 ± 0.00	4.82 ± 0.10	8.28 ± 0.28	0.1
	Pt-Pt (2 nd)	6	3.89 ± 0.01	4.79 ± 0.49		

Pt-N indicates a single scattering path of first shell. Pt-C and ∠Pt-C indicate a single scattering path of second shell and an obtuse triangle path of Pt-C, respectively (**Shell** column). The **CN** is coordination number obtained from the amplitude reduction factor (S_0^2) of 0.85. * denotes a fixed constant value of **CN** obtained from the crystallographic data in a previous report². **R** indicates bond distance. σ^2 indicates Debye-Waller factor. ΔE_0 indicates energy shift. **R factor** obtained from the best fit for the respective catalysts.

- The DFT derived “overpotential” is actually thermodynamic overpotential since no kinetics are included. This must be clarified or at least acknowledged, though this still is a valid descriptor for comparison between sites as determined in extensive HER literature.

We thank the reviewer for the very insightful comment. As the reviewer pointed out, the theoretical overpotential, which was invented by Nørskov and co-workers [*J. Phys. Chem. B* **108**, 17886-17992 (2004)], is a thermodynamic descriptor for the activity, where kinetics are not included. To clarify this, we referred the DFT-derived overpotential for CER or OER to the “thermodynamic overpotential” (i.e., $\eta_{TD(CER)}$ and $\eta_{TD(OER)}$ for CER and OER, respectively) to distinguish from the experimental overpotential (η) throughout the revised manuscript.

Changes made:

- We have revised the following sentences on page 13 of the revised manuscript:

Original sentences: “The theoretical overpotential for CER (η_{CER}) can be defined from the ΔG depending on the reaction intermediates as follows: ΔG_{Cl^*} for Cl^* species, and $\Delta G_{ClO^*} - \Delta G_{O^*}$ for ClO^* species, respectively (Supplementary Note 3). Among Pt-N₄ sites, PtN₄C₁₂ was identified as the most plausible structure for CER due to their thermo-neutral state at 1.36 V with $\eta_{CER} = 0.09$ V.”

Revised sentences: “The thermodynamic overpotential for CER can be defined from the ΔG divided by the elementary charge at zero overpotential (i.e., $\eta_{TD(CER)} = \frac{\Delta G}{e}$), which depends on the reaction intermediates as follows: ΔG_{Cl^*} for bare structure (*) and Cl^* species, and $\Delta G_{ClO^*} - \Delta G_{O^*}$ for O^* and ClO^* species, respectively (Supplementary Note 4)⁶². Among Pt-N₄ sites, PtN₄C₁₂ was identified as the most plausible structure for CER due to its lowest $\eta_{TD(CER)}$ of 0.09 V at zero overpotential.”
- We have revised the following phrases on page 13 of the revised manuscript:

Original phrase: “ $\eta_{CER} = 0.20$ V for 2O_b2ClO_c”

Revised phrase: “ $\eta_{\text{TD}(\text{CER})} = 0.20 \text{ V}$ for $2\text{O}_b2\text{ClO}_c$ ”

Original phrase: “ $\eta_{\text{CER}} = 0.62 \text{ V}$ for $2\text{O}_b2\text{Cl}_c$ ”

Revised phrase: “ $\eta_{\text{TD}(\text{CER})} = 0.62 \text{ V}$ for $2\text{O}_b2\text{Cl}_c$ ”

- We have revised the following phrases on page 15 of the revised manuscript:

Original phrase: “with large theoretical overpotentials for OER (i.e., $\eta_{\text{OER}} \geq 0.87 \text{ V}$, ...)”

Revised phrase: “with a large $\eta_{\text{TD}(\text{OER})}$ above 0.74 V (**Supplementary Table 5**).”

Original phrase: “with very high overpotential of 0.75 V ”

Revised phrase: “with very high $\eta_{\text{TD}(\text{OER})}$ of 0.62 V , corresponding to the formation of OOH^* ”

- We have revised the following sentences in the **Supplementary Note 4** on pages S50 of the revised supporting information:

Original sentences: “Then, the theoretical overpotential for CER (η_{CER}) can be defined as follows,

(I) For Cl^* intermediate:

$$\eta_{\text{CER}} = \frac{|\Delta G_{\text{Cl}^*}|}{e} \quad (1)$$

(II) For ClO^* intermediate:

$$\eta_{\text{CER}} = \frac{|\Delta G_{\text{ClO}^*} - \Delta G_{\text{O}^*}|}{e} \quad (2)$$

To investigate the theoretical overpotential for OER (η_{OER}) of Pt–N₄ sites, we assumed the conventional four-electron pathway for the OER as follows,”

Revised sentences: “Then, the thermodynamic overpotential for CER at zero overpotential ($\eta_{\text{TD}(\text{CER})}$) can be defined as follows,

(I) For O^* and Cl^* species:

$$\eta_{\text{TD}(\text{CER})} = \frac{|\Delta G_{\text{Cl}^*}|}{e} \quad (3)$$

(II) For O^* and ClO^* species:

$$\eta_{\text{TD}(\text{CER})} = \frac{|\Delta G_{\text{ClO}^*} - \Delta G_{\text{O}^*}|}{e} \quad (4)$$

To investigate the thermodynamic overpotential for OER at zero overpotential ($\eta_{\text{TD}(\text{OER})}$) of Pt–N₄ sites, we assumed the conventional four-electron pathway for the OER as follows,”

- We have revised the following sentences in the **Supplementary Note 4** on pages S51 of the revised supporting information:

Original sentences: “Finally, η_{OER} can be defined by

$$\eta_{\text{OER}} = \frac{\max[\Delta G_1, \Delta G_2, \Delta G_3, \Delta G_4]}{e} - U_{\text{eq}} [V] \quad (5)$$

where U_{eq} indicates the equilibrium potential for OER (i.e., 1.23 V).”

Revised sentences: “Finally, $\eta_{\text{TD(OER)}}$ can be defined by

$$\eta_{\text{TD(OER)}} = \frac{\max[\Delta G_1, \Delta G_2, \Delta G_3, \Delta G_4]}{e} - U_{\text{eq}} [V] \quad (6)$$

where U_{eq} indicates the equilibrium potential for OER (i.e., 1.23 V vs SHE).”

9. In SI Note 4, 1st sentence “Pt atom on the....” seems to be missing a description of the binding site for delEads, though in equations the site is described as “Pt-free”.

We thank the reviewer to point out our mistake. As a response to the following Comment 10, we extensively revised the **Supplementary Note 5**, including the missing description that the reviewer pointed out in this comment.

Changes made:

- We have revised the following phrase in **Supplementary Note 5** on page S53 of the revised supporting information:

Original phrase: “by comparing the energy difference between the adsorption energy of Pt atom on the (ΔE_{ads}) and ...”

Revised phrase: “by comparing the energy difference between the adsorption energy of Pt^{2+} ion in the Pt– N_4 sites (ΔE_{ads}) and ...”

10. For the stability from DFT, at 1.36V in acid, Pt bulk is not the relevant low-free-energy state to compare to but rather a solvated Pt^{2+} ionic state would be a more appropriate energetic comparison. This will give a constant energetic shift at constant conditions between the bulk and ionic Pt, however, so using this to evaluate relative stability should give the same result but a change in sign (likely) would indicate that the Pt ion was more stable under such conditions than the PtN4. This should be addressed since stability against metal dissolution is likely a major contributor to active site loss.

We appreciate the reviewer’s valuable suggestion regarding the stability against metal dissolution. We agree to the reviewer’s opinion that a solvated Pt^{2+} ionic state would be the more relevant low-free-energy state to compare. In this regard, there was a report that Pt dissolution in the presence of chlorides proceeded via the formation of Pt-chloro complex under acidic CER conditions [*Electrochim. Acta* **179**, 24-31 (2015)]. To reflect this situation, we changed the relevant reference state to compare from the Pt bulk into a Pt-chloro complex in water (i.e., particularly in the form of $[\text{PtCl}_4]^{2-}$). We compared the energy difference between the adsorption energy of Pt^{2+} ion in the Pt– N_4 sites (ΔE_{ads}) and the Pt^{2+} chlorination energy in the $[\text{PtCl}_4]^{2-}$ complex (ΔE_{Cl^-}) (i.e., $\Delta E_{\text{ads}} - \Delta E_{\text{Cl}^-}$, **Revised Supplementary Fig. 32**).

We found that the energetic shift was not exactly same in comparison with our calculated

data based on Pt bulk state (i.e., $\Delta E_{\text{ads}} - \Delta E_{\text{coh}}$, Original Supplementary Fig. 25). This was because the reference states for ΔE_{ads} were changed from both neutral (i.e., $E_{\text{Pt-atom}}$ for Pt atom, and $E_{\text{Pt-free}}$ for Pt–N₄ clusters without Pt atom) into positively and negatively charged states, respectively (i.e., $E_{\text{Pt}^{2+}}$ for a solvated Pt²⁺ ion, and $E_{\text{Pt-free}}$ for Pt–N₄ clusters without Pt²⁺ ion in a reduced state), leading to the uneven changes in the respective energy states.

As a result, all Pt–N₄ sites were identified to be stable against metal dissolution as indicated by a negative sign ($\Delta E_{\text{ads}} - \Delta E_{\text{Cl}^-} < 0$). Also, the general tendency for the relative stability remained unchanged (in the order of PtN₄C₁₂, PtN₄C₁₀, and PtN₂₊₂C₄₊₄), implying that PtN₄C₁₂ species were still thermodynamically more stable against metal dissolution than other ones. Therefore, we expect that the loss of active site by the metal dissolution would rarely occur in the Pt₁/CNT catalyst.

Revised Supplementary Fig. 32 | a The difference between adsorption energy of Pt²⁺ ion in the Pt–N₄ sites (ΔE_{ads}) and the chlorination energy of Pt²⁺ ion in the [PtCl₄]²⁻ complex (ΔE_{Cl^-}). b Optimized structure of [PtCl₄]²⁻ complex, used for ΔE_{Cl^-} calculation. Colour legends – Pt: dark-blue; Cl: yellow-green.

Changes made:

- We have revised the following sentences on page 14 of the revised manuscript to reflect **Revised Supplementary Fig. 32**:

Original sentences: “Moreover, we evaluated the thermodynamic stability of the Pt–N₄ sites by calculating the difference in adsorption energy (ΔE_{ads}) and cohesive energy (ΔE_{coh}) (Supplementary Fig. 25). The negative value of $\Delta E_{\text{ads}} - \Delta E_{\text{coh}}$ implies that embedding Pt atom into the adjacent network is more stable than it leaching or aggregating with other Pt atoms (Supplementary Note 5).”

Revised sentences: “Moreover, we evaluated the thermodynamic stability of the Pt–N₄ sites by calculating the difference in adsorption energy (ΔE_{ads}) and chlorination energy (ΔE_{Cl^-}) of Pt²⁺ ion (Supplementary Fig. 32). The negative value of $\Delta E_{\text{ads}} - \Delta E_{\text{Cl}^-}$ implies that embedding Pt²⁺ ion into the adjacent network is more stable against Pt dissolution, which may proceed via the formation of Pt-chloro complex (Supplementary Note 5).”

- We have extensively revised the **Supplementary Note 5** in the revised supporting information:

Original sentences: “By comparing the energy difference between the adsorption energy of Pt atom on the (ΔE_{ads}) and the Pt bulk cohesive energy (ΔE_{coh}), we evaluated the stability of Pt–N₄ sites (Supplementary Fig. 25)²⁰. The ΔE_{ads} and ΔE_{coh} were

calculated by,

$$\Delta E_{\text{ads}} = E_{\text{Pt-N}_4} - E_{\text{Pt-free}} - E_{\text{Pt-atom}} \quad (48)$$

$$\Delta E_{\text{coh}} = E_{\text{Pt-bulk}} - n \times E_{\text{Pt-atom}} \quad (49)$$

where $E_{\text{Pt-N}_4}$, $E_{\text{Pt-free}}$, $E_{\text{Pt-atom}}$, and $E_{\text{Pt-bulk}}$ represent the DFT-optimised energies of Pt-N₄ sites, Pt-N₄ sites where the Pt atom is not included, an isolated Pt atom, and bulk Pt unit cell, respectively. n is 4, indicating the formula unit.”

Revised sentences: “The stability of Pt-N₄ sites against Pt dissolution was theoretically evaluated (**Supplementary Fig. 32**). It was previously reported that Pt dissolution in the presence of Cl⁻ proceeds through the formation of metastable Pt-chloro complex such as [PtCl₄]²⁻ (ref. ²¹). To reflect this situation, we investigated the relative stability of Pt-N₄ sites by comparing the energy difference between the adsorption energy of Pt²⁺ ion in the Pt-N₄ sites (ΔE_{ads}) and the chlorination energy of Pt²⁺ ion in the [PtCl₄]²⁻ complex (ΔE_{Cl^-}). The ΔE_{ads} and ΔE_{Cl^-} were calculated by,

$$\Delta E_{\text{ads}} = E_{\text{Pt-N}_4} - E_{\text{Pt-free}} - E_{\text{Pt}^{2+}} \quad (7)$$

$$\Delta E_{\text{Cl}^-} = E_{[\text{PtCl}_4]^{2-}} - E_{\text{Pt}^{2+}} - 4 E_{\text{Cl}^-} \quad (8)$$

where $E_{\text{Pt-N}_4}$, $E_{\text{Pt-free}}$, $E_{\text{Pt}^{2+}}$, $E_{[\text{PtCl}_4]^{2-}}$, and E_{Cl^-} represent the DFT-optimised energies of Pt-N₄ sites, bare Pt-N₄ sites without Pt²⁺ ion in a reduced state, a solvated Pt²⁺ ion in water, [PtCl₄]²⁻ complex, and a solvated Cl⁻ ion in water, respectively.”

- We replaced **Supplementary Reference 21** with the following literature on page S56 of the revised supporting information:

21. Geiger, S., Cherevko, S., Mayrhofer, K.J.J. Dissolution of platinum in presence of chloride traces. *Electrochim. Acta* **179**, 24-31 (2015).

Responses to the Comments of the Reviewer 3

Atomically dispersed Pt-N4 sites as efficient and selective electrocatalysts for the chlorine evolution reaction

T. Lim et al.

In the present manuscript, the authors report the synthesis, characterization, and application of atomically dispersed Pt-N4 on carbon nanotubes for the chlorine evolution reaction (CER), a reaction system of potential interest for chemical industry due to the production of gaseous chlorine in chlor-alkali electrolysis. The measured activity and selectivity results are compared to those of commercial DSA, which are applied as state-of-the-art electrocatalysts in the chlor-alkali process. Density functional theory calculations are employed to gain insights into the high CER activity and selectivity on a molecular level.

The application of atomically dispersed electrocatalysts is innovative and corresponds to an alternative approach compared to commonly studied transition-metal oxide-based electrodes for chlorine electrocatalysis. Therefore, the manuscript might deserve to get published in Nat. Comm., but not in the present form. Before this reviewer can recommend the article for publication, the authors need to carry out a major revision by taking the following 23 points into account:

We truly appreciate the reviewer's endorsement of our paper.

1. Literature survey: recently, two articles concerning the CER selectivity of precious metal oxides have been published in ChemElectroChem that deserve further attention, but at least should be cited:

10.1002/celec.201900784

10.1002/celec.201900834

We appreciate the reviewer's recommendation of the relevant literature to our work. We have added the cited literature in the references.

Changes made:

- We have added the following references in the revised manuscript.
 21. Wintrich, D. *et al.* Enhancing the selectivity between oxygen and chlorine towards chlorine during the anodic chlorine evolution reaction on a dimensionally stable anode. *ChemElectroChem* **6**, 3108–3112 (2019).
 22. Exner, K. S. Controlling stability and selectivity in the competing chlorine and oxygen evolution reaction over transition metal oxide electrodes. *ChemElectroChem* **6**, 3401–3409 (2019).
2. Line 33: 60 millions tons of Cl₂ per year is outdated. Nowadays, the annual production amounts to about 70 millions tons or even more. I advise the authors to refer to state-of-the-art reports, such as Chlor Alkali Industry Reviews of Euro Chlor, and to cite those.

We thank the reviewer’s suggestion to update the annual production amounts of Cl₂. We have changed the value to 75 million tons according to a report from the World Chlorine Council published in 2017.

Changes made:

- We have added the following reference in the revised manuscript.
 1. World Chlorine Council, *Sustainable progress* (World Chlorine Council, 2017).
- We have revised the following sentence on page 2 of the revised manuscript:
Original sentence: “Chlorine (Cl₂) is one of the most important industrial chemicals with an annual production of around 60 million tons¹.”
Revised sentence: “Chlorine (Cl₂) is one of the most important industrial chemicals with an annual production of around 75 million tons worldwide¹.”

3. Line 36: Equilibrium potential of the CER is missing.

We thank the reviewer’s suggestion to provide details of the CER. For easy comparison, we have newly added **Supplementary Note** including the reaction equation and the equilibrium potential of the CER and OER.

Changes made:

- We have revised the following sentence on page 2 of the revised manuscript:
Original sentence: “The electrochemical chlorine evolution reaction (CER, $2 \text{Cl}^-_{(\text{aq})} \rightarrow \text{Cl}_{2(\text{aq})} + 2 \text{e}^-$) plays a pivotal role in the chlor-alkali process as the anodic reaction.”
Revised sentence: “The electrochemical chlorine evolution reaction (CER) plays a pivotal role in the chlor-alkali process as the anodic reaction⁵⁻⁸.”
- We have newly added **Supplementary Note 1** on page S46 of the revised Supporting Information:
New Paragraph: “On the potential scale of reversible hydrogen electrode (RHE), the CER and OER occur via the following reactions (1) and (2) with their standard reversible electrode potentials (E^0), respectively:
 $2 \text{Cl}^- \rightleftharpoons \text{Cl}_2 + 2 \text{e}^-$
 $E^0_{\text{CER}} = \left(1.358 + \frac{RT}{F} \cdot 2.303 \cdot \text{pH} \right) \text{ V vs. RHE} \quad (1)$
 $2 \text{H}_2\text{O} \rightleftharpoons \text{O}_2 + 4 \text{H}^+ + 4 \text{e}^-$
 $E^0_{\text{OER}} = 1.229 \text{ V vs. RHE} \quad (2)$
where T , R , and F represent the temperature ($T = 298.15 \text{ K}$), the universal gas constant, and the Faraday constant, respectively.

4. Line 42: The equilibrium potential of the OER is reduced (on the SHE scale) by increasing the pH, resulting in an enhanced overpotential and, hence, a higher catalytic turnover at a constant electrode potential on the SHE scale compared to smaller pH values. The term “thermodynamic barrier”, however, is misleading, because the reduction of the OER equilibrium potential directly affects the underlying kinetics.

We appreciate and consent to the reviewer’s concern over the use of the term “thermodynamic barrier”. This term did not consider the underlying kinetic effect of pH on the OER. We have therefore revised the sentence by changing “thermodynamic barrier”

to the general term “overpotential”, which includes both thermodynamic and kinetic effects of pH on the OER.

Changes made:

- We have revised the following sentence on page 2 of the revised manuscript.
Original sentence: “The AC generation is typically conducted in neutral pH, where the thermodynamic barrier of the oxygen evolution reaction (OER), the side reaction of CER, becomes lower than that in acidic pH.”
Revised sentence: “The AC generation is typically conducted in neutral pH, where the oxygen evolution reaction (OER), the side reaction of CER, shows lower overpotential than that in acidic pH^{13,14}.”
5. Line 43: It would be advantageous to provide a reaction equation for the OER, including its equilibrium potential, in comparison to that of the CER (cf. line 36).

As we responded to the Comment 3, we have revised the related sentence following the reviewer’s recommendation.

Changes made:

- We have newly added **Supplementary Note 1** on page S46 of the revised Supporting Information:
New Paragraph: “On the potential scale of reversible hydrogen electrode (RHE), the CER and OER occur via the following reactions (1) and (2) with their standard reversible electrode potentials (E^0), respectively:
 $2 \text{Cl}^- \rightleftharpoons \text{Cl}_2 + 2\text{e}^-$
 $E_{\text{CER}}^0 = \left(1.358 + \frac{RT}{F} \cdot 2.303 \cdot \text{pH} \right) \text{ V vs. RHE} \quad (1)$
 $2 \text{H}_2\text{O} \rightleftharpoons \text{O}_2 + 4 \text{H}^+ + 4 \text{e}^-$
 $E_{\text{OER}}^0 = 1.229 \text{ V vs. RHE} \quad (2)$
where T , R , and F represent the temperature ($T = 298.15 \text{ K}$), the universal gas constant, and the Faraday constant, respectively.”
6. Line 55: Not correct, about 30% of precious metals, such as Ru, are needed to have sufficient electronic conductivity.

We appreciate the reviewer’s correction on the role of precious metals in MMO electrocatalysts. We have revised the related sentence in the revised manuscript.

Changes made:

- We have revised the following sentence on page 3 of the revised manuscript.
Original sentence: “Nevertheless, high amounts (around 30 at%) were required to maintain CER activity.”
Revised sentence: “Nevertheless, high amounts (around 30 at%) were required to maintain sufficient electronic conductivity for the CER⁵.”

7. Line 73: “lower Tafel slope”: this statement is insufficient. The Tafel slope is overpotential dependent, i.e., the lower the overpotential, the smaller the Tafel slope, and vice versa. Every electrocatalyst exhibits a threshold overpotential, at which the Tafel slope exceeds 60 mV/dec. This is due to the fact that the first elementary reaction step in the free-energy landscape becomes rate determining. I advise the authors to refer to the state-of-the-art literature concerning the connection of Tafel slopes and free-energy diagrams and to revise this statement. If Tafel slopes are reported, it is indispensable to enclose the accompanied overpotential range for the respective Tafel slope.

We are grateful to the reviewer’s keen insight into Tafel slope. Following the suggestion of the reviewer, we have newly added **Reference 64** [Exner, K. S., Sohrabnejad-Eskan, I. & Over, H. *ACS Catal.* **8**, 1864–1879 (2018)], which describes the relation between Tafel slopes and free-energy diagrams. Also, the Tafel slope of PtNP/CNT and DSA catalysts were newly obtained in the potential range of 40–80 mV (**Revised Fig. 2b and e**). We note that the Tafel slopes of PtNP/CNT and DSA were obtained in different range of potential, compared to that used for Pt₁/CNT to consider their different CER activities. **Revised Fig. 2e** displays newly obtained Tafel slopes of these catalysts for CER, but the resulting values were not significantly different from the previously calculated values [Pt₁/CNT (38 mV dec.⁻¹), PtNP/CNT (52 mV dec.⁻¹), and DSA (60 mV dec.⁻¹)]. We have specified the potential ranges used for the calculation of Tafel slopes in the revised sentences. Note that the newly established E_{CER} (vs. RHE) based on the Nernst equation is used in **Revised Fig. 2e**, as mentioned in the response to the Comment 3.

Revised Fig. 2e | Tafel plots of Pt₁/CNT, PtNP/CNT, and DSA catalysts. Their Tafel slopes are denoted in parentheses.

Changes made:

- We have revised the **Fig. 2e** in page 33 of the revised manuscript.
- We have revised the following sentence on page 4 of the revised manuscript.

Original sentence: “The Pt₁/CNT catalyst exhibited superior CER activity to Pt nanoparticles on CNT (PtNP/CNT) and commercial Ru/Ir-based DSA catalysts in acidic media, with its lower overpotential and lower Tafel slope.”

Revised sentence: “The Pt₁/CNT catalyst exhibited superior CER activity to Pt nanoparticles on CNT (PtNP/CNT) and commercial Ru/Ir-based DSA catalysts in acidic media.”

- We have revised the following sentence on page 11 of the revised manuscript to specify the range of potential for the corresponding Tafel slope.

Original sentence: “Pt₁/CNT showed a Tafel slope of 38 mV dec.⁻¹, which is lower than those of PtNP/CNT (62 mV dec.⁻¹) and DSA (58 mV dec.⁻¹), suggesting faster CER kinetics of the former.”

Revised sentence: “Pt₁/CNT showed a Tafel slope of 38 mV dec.⁻¹ at the overpotential range of 30–70 mV, whereas those of PtNP/CNT and DSA were higher with 52 mV dec.⁻¹ and 60 mV dec.⁻¹ at the range of 40–80 mV, respectively. The Tafel analyses suggest that the CER on Pt₁/CNT proceeded with faster kinetics than on PtNP/CNT and DSA.”

8. Line 138: “Onset potentials” are physicochemically not well defined. If the authors want to compare the activity of electrocatalysts at low overpotentials, they should derive and compile exchange current densities.

We thank the reviewer for suggesting exchange current densities instead of on-set potentials for comparing the activity of electrocatalysts. In response, we have removed the term “on-set potential” from the manuscript. For more clear comparison of the catalytic activity, we have calculated the exchange current densities (j_0) for the three main catalysts, Pt₁/CNT, PtNP/CNT, and DSA.

Changes made:

- We have revised the following sentence on page 8 of the revised manuscript.

Original sentence: “Pt₁/CNT started to catalyse the CER at an on-set potential of 1.38 V, which is merely 20 mV higher than the standard electrode potential of CER (1.36 V vs. reversible hydrogen electrode, RHE).”

Revised sentence: “Pt₁/CNT started to catalyse the CER at a potential of 1.38 V, which is 30 mV higher than the E_{CER} (1.35 V vs. RHE for 25 °C).”

- We have added a new sentence on page 8 of the revised manuscript:

New sentence: “The intrinsic catalytic activity of electrocatalysts for the CER was also assessed in terms of exchange current density (j_0) using Tafel analyses (see Equation 8 in **Methods**). The j_0 of Pt₁/CNT was 0.43 mA cm⁻² at the overpotential range of 30–70 mV, whereas those of PtNP/CNT and DSA were 0.23 mA cm⁻² and 0.20 mA cm⁻² at the overpotential range of 40–80 mV, respectively. Pt₁/CNT loaded on a carbon paper showed j_0 of 0.44 mA cm⁻² similar to the value obtained on a RRDE, suggesting that j_0 was invariant on the type of electrode substrates.”

9. Line 144/145: Under the reaction conditions of the authors, the equilibrium potential of the CER is likely not equal to the standard equilibrium potential (1.36 V vs. SHE), but rather $U_{\text{CER}} < 1.36$ V vs. SHE due to $a_{\text{Cl}_2} < 1$ is valid. I advise the authors to refer to the literature concerning the determination of the equilibrium potential for the CER. If I am not mistaken, some information can be found in reference 21.

We appreciate the reviewer’s critical comment on the equilibrium potential of CER (E_{CER}^0). We agree to the comment and have calculated the E_{CER} under our experimental conditions, following the information provided in **Reference 24** in the revised manuscript

(Reference 21 in the original manuscript). We note that we stated all values of experimental potentials in the RHE scale (see **Methods**). We found that the equilibrium potential of the CER is 1.35 V under our experimental conditions. Accordingly, the overpotential values were also re-established throughout the revised manuscript and supporting information (e.g., **Fig. 2b and e**).

Revised Fig. 2b and e | b Calculated TOFs of Pt₁/CNT and PtNP/CNT catalysts. e Tafel plots of Pt₁/CNT, PtNP/CNT, and DSA catalysts. Their Tafel slopes are denoted in parentheses.

Changes made:

- We have added new sentences on page 7 of the revised manuscript.

New sentences: “In the RHE scale, the equilibrium potential of CER (E_{CER}) is dependent on pH, temperature, concentration of Cl^- ions, and partial pressure of Cl_2 of the electrolyte. The E_{CER} was derived by the Nernst equation^{24,52}:

$$E_{\text{CER}}(T, a(\text{Cl}_2), a(\text{Cl}^-)) \text{ vs. RHE} \\ = E_{\text{CER}}^0 - \frac{RT}{F} \cdot \ln a(\text{Cl}^-) + \frac{RT}{2F} \cdot \ln a(\text{Cl}_2) \quad (1)$$

where a , T , R , and F represent the chemical activity, the temperature, the universal gas constant, and the Faraday constant, respectively. The $a(\text{Cl}_2)$ was assumed as 0.01 for a partial pressure of evolving Cl_2 under Ar purging. The $a(\text{Cl}^-)$ was determined by the experimental conditions (i.e., $a(\text{Cl}^-) = 1.0$ for 1.0 M NaCl)⁵². The temperature dependence of the E_{CER}^0 vs. RHE can be calculated from the following equation (2)⁵³.

$$E_{\text{CER}}^0 = (1.358 \text{ V} + \frac{RT}{F} \cdot 2.303 \cdot \text{pH}) - (0.001248 \frac{\text{dV}}{\text{dK}}) \cdot (T - 298.15 \text{ K}) \quad (2)$$

- We have revised the following sentences on page 8 of the revised manuscript:

Original sentences: “As a result, Pt₁/CNT showed 2.6 times higher TOF than PtNP/CNT at an overpotential of 60 mV (**Fig. 2b**). The CER activities were also tested on a carbon paper (1 cm × 1 cm). Pt₁/CNT coated on a carbon paper required an overpotential of 60 mV to reach a current density of 10 mA cm⁻² (**Supplementary Fig. 10**).”

Revised sentences: “As a result, Pt₁/CNT showed 2.6 times higher TOF than PtNP/CNT at an overpotential of 70 mV (**Fig. 2b**). The CER activities were also tested on a carbon paper (1 cm × 1 cm). Pt₁/CNT coated on a carbon paper required an overpotential of 70 mV to reach a current density of 10 mA cm⁻² (**Supplementary Fig. 10**).”
- We have revised the **Fig. 2b and e** in page 33 of the revised manuscript.

10. Line 146/147: At which temperatures were the measurements conducted? Room temperature? In chemical industry, CER is carried out at about 80 – 90 °C. Are the proposed Pt/CNT electrocatalysts stable at this temperature? In order to compare the performance of the Pt/CNT electrodes to that of DSA, I think the activity and stability at higher temperature needs to be probed. Otherwise, the authors need to clearly state, which temperature interval was investigated in their study. Throughout the entire manuscript, no temperature concerning the activity or selectivity measurements is reported.

We thank the reviewer's important comment concerning temperature of CER experiments. All of our electrochemical experiments were conducted at room temperature (~25 °C). In response to the reviewer's comment we measured the CER activity of Pt₁/CNT and DSA catalysts under conditions relevant to the industrial chlor-alkali process (~80 °C). Under this condition, the overpotentials of Pt₁/CNT and DSA were 50 mV and 80 mV at a current density of 10 mA cm⁻², respectively (newly added **Supplementary Fig. 13**). These activity results are similar to the trend obtained at 25 °C. In CER stability tests (**Review Only Fig. 10**), Pt₁/CNT retained 53% of initial current after 12 hours of operating, whereas DSA showed high stability preserving 95% of initial current. Overall, Pt₁/CNT preserved superior CER activity at 80 °C but its stability was deteriorated. We will investigate the origin of this temperature-dependent activity and stability trends of Pt₁/CNT as a future work.

We note, however, that the CER under room temperature is also important for small-scale active chlorine (AC) generation, such as water disinfection and wastewater treatment devices. We thank again for the reviewer's valuable suggestion.

New Supplementary Fig. 13 | CER polarization curves of Pt₁/CNT catalyst loaded on a carbon paper and DSA in 0.1 M HClO₄ + 1.0 M NaCl at 80 °C. The equilibrium potential of CER (E_{CER}) depends on the temperature (see Methods in the manuscript).

Review Only Fig. 10 | Chronoamperograms of Pt₁/CNT and DSA catalysts loaded on a carbon paper and DSA catalyst measured in 0.1 M HClO₄ + 1.0 M NaCl under 80 °C for 12 h with a stirring speed of 300 rpm.

Changes made:

- We have added the new sentence on page 7 of the revised manuscript.
New sentences: “All electrochemical measurements were conducted at room temperature (~25 °C) unless otherwise specified.”
- We have newly added the **Supplementary Fig. 13** in page 17 of the revised manuscript.
- We have added the new sentence on page 9 of the revised manuscript to reflect newly added **Supplementary Fig. 13**.
New sentences: “Since the practical chlor-alkali process is operated at temperatures of 80–90 °C, we also measured the CER activity of Pt₁/CNT and DSA catalysts at 80 °C, which indicated a similar activity trend to that obtained at 25 °C (**Supplementary Fig. 13**)”
- We have added new sentences on page 7 of the revised manuscript.
New sentences: “In the RHE scale, the equilibrium potential of CER (E_{CER}) is dependent on pH, temperature, concentration of Cl⁻ ions, and partial pressure of Cl₂ of the electrolyte. The E_{CER} was derived by the Nernst equation^{24,52}:

$$E_{\text{CER}}(T, a(\text{Cl}_2), a(\text{Cl}^-)) \text{ vs. RHE} \\ = E_{\text{CER}}^0 - \frac{RT}{F} \cdot \ln a(\text{Cl}^-) + \frac{RT}{2F} \cdot \ln a(\text{Cl}_2) \quad (1)$$

where a , T , R , and F represent the chemical activity, the temperature, the universal gas constant, and the Faraday constant, respectively. The $a(\text{Cl}_2)$ was assumed as 0.01 for a partial pressure of evolving Cl₂ under Ar purging. The $a(\text{Cl}^-)$ was determined by the experimental conditions (i.e., $a(\text{Cl}^-) = 1.0$ for 1.0 M NaCl)⁵². The temperature dependence of the E_{CER}^0 vs. RHE can be calculated from the following equation (2)⁵³.

$$E_{\text{CER}}^0 = (1.358 \text{ V} + \frac{RT}{F} \cdot 2.303 \cdot \text{pH}) - (0.001248 \frac{\text{dV}}{\text{dK}}) \cdot (T - 298.15 \text{ K}) \quad (2)$$

11. Line 158: Supplementary Table 3: I suppose that the overpotential @10 mA/cm² is calculated based on the standard equilibrium potential, $U_{0_CER} = 1.36 \text{ V}$ vs. SHE. Please revise referring to point viii). The analysis would be even stronger, if exchange current densities would be provided additionally.

We are grateful to the reviewer’s constructive suggestion to re-calculate the overpotential following the Nernst equation and to include exchange current densities in the Table. As

we responded to the Comment 9, we have re-established the equilibrium potential under our electrochemical measurement conditions. In addition, we have added the values of the exchange current densities (j_0) in revised **Supplementary Table 3**. As most of previously reported papers did not provide the exchange current density values, we could have only added the exchange current of RuO₂ (110) in the comparison table [newly added **Supplementary Reference 3**, *ACS Catal.* **7**, 2403–2411 (2017)].

Revised Supplementary Table 3 | Comparison of CER activity and operation condition of Pt₁/CNT catalyst and those of previously reported catalysts in acidic media.

Catalysts	Overpotential @10 mA cm ⁻² (mV)	Exchange current density (mA cm ⁻²)	CER operation conditions	Precious metal contents	Ref.
Pt ₁ /CNT (RRDE method)	50	0.43	0.1 M HClO ₄ + 1.0 M NaCl (pH 0.9, 25 °C)	2.7 wt% Pt (0.17 at% Pt)	This work (Fig. 2a)
Pt ₁ /CNT (carbon paper)	70	0.44			This work (Supplementary Fig. 12)
PtNP/CNT (RRDE method)	120	0.23	0.1 M HClO ₄ + 1.0 M NaCl (pH 0.9, 25 °C)	2.9 wt% Pt (0.18 at% Pt)	This work (Fig. 2a)
Commercial DSA, Ru–Ti–Ir/Ti (Siontech, Korea)	105	0.20	0.1 M HClO ₄ + 1.0 M NaCl (pH 0.9, 25 °C)	NA	This work (Supplementary Fig. 12)
RuO ₂ (110)	140*	4.5×10 ⁻³	0.1 M HClO ₄ + 1.0 M NaCl (pH 0.9, 25 °C)	NA	Supplementary Ref. ³
Commercial DSA, Ru–Ti–Ir/Ti (Covestro, Germany)	140*	NA	3.0 M NaNO ₃ + 1.0 M NaCl (pH 3.0, 25 °C)	NA	Supplementary Ref. ⁴
Commercial DSA, Ru _{0.3} Ti _{0.7} O ₂ /Ti (Covestro, Germany)	80	NA	3.5 M NaCl (pH 3.0, 80 °C)	30 at% Ru	Supplementary Ref. ⁵
Mesoporous Ru–Ir/TiO ₂	135*	NA	4.0 M NaCl (pH 3.0, 25 °C)	7.5 wt% Ru 7.5 wt% Ir	Supplementary Ref. ⁶
RuTiO _x /SbSnO _x	120	NA	5.0 M NaCl (pH 2.0, 25 °C)	NA	Supplementary Ref. ⁷

* denotes the value of overpotential without considering *iR* compensation; **CER operation conditions** column indicates Cl⁻ concentration, pH value, and temperature.

Changes made:

- We have revised the **Supplementary Table 3** in the page S43 of the revised supplementary information.
- We have added a new sentence on page 8 of the revised manuscript:
New sentence: “The intrinsic catalytic activity of electrocatalysts for the CER was also assessed in terms of exchange current density (j_0) using Tafel analyses (see Equation 8 in **Methods**). The j_0 of Pt₁/CNT was 0.43 mA cm⁻² at the overpotential range of 30–70 mV, whereas those of PtNP/CNT and DSA were 0.23 mA cm⁻² and 0.20 mA cm⁻² at the overpotential range of 40–80 mV, respectively. Pt₁/CNT loaded on a carbon paper showed j_0 of 0.44 mA cm⁻² similar to the value obtained on a RRDE, suggesting that j_0 was invariant on the type of electrode substrates.”

12. Line 177/178: The comparison of the Pt/CNT electrodes to DSA appears a bit odd, since DSA are stable up to 10 years. A current drop of 28% after already 12 h appears not to be in good agreement with the performance of DSA.

We consent to the reviewer’s comments that DSA are stable enough to be operated as an anode over a decade for a chlor-alkali membrane electrolyzer at high temperature [Best Available Techniques (BAT) Reference Document for the Production of Chlor-alkali. *Tech. rep.* (2014)]. However, in the lab-scale chrono-analyses (chronoamperometry or chronopotentiometry) at room temperature, the initial performance decay of mixed metal oxides electrocatalysts (i.e., decrease in current or increase in potential) can be found in many literature (three representative examples displayed in **Review Only Fig. 11**). The origin of this initial degradation can be ascribed by the insufficient diffusion or underlying kinetics of low temperature, as referred in the response to the Comment 10.

Review Only Fig. 11 | a The CER chronoamperograms of four different commercial DSAs prepared by the

same method (Bayer Materials, Germany). Reproduced from Fig. 1a in *Electrochim. Acta* **82**, 408–414 (2012). b The CER chronoamperograms of three different DSAs prepared by different preparation methods. Reproduced from Fig. 3d in *Small* **13**, 1602240 (2017). c The CER chronoamperograms of the thermally prepared Ru/TiO_x on an antimony-doped tin oxide (ATO) film using different preparation methods. Reproduced from Fig. 2c in *Energy Environ. Sci.* **12**, 1241–1248 (2019).

13. Line 202/203: Please revise the Tafel slope analysis referring to point vi).

As we responded to the Comment 7, we have revised the sentence following the reviewer's suggestion.

Changes made:

- We have revised a following sentence on page 11 of the revised manuscript to specify the range of potential for the corresponding Tafel slope.
- **Original sentence:** “Pt₁/CNT showed a Tafel slope of 38 mV dec.⁻¹, which is lower than those of PtNP/CNT (62 mV dec.⁻¹) and DSA (58 mV dec.⁻¹), suggesting faster CER kinetics of the former.”
- **Revised sentence:** “Pt₁/CNT showed a Tafel slope of 38 mV dec.⁻¹ at the overpotential range of 30–70 mV, whereas those of PtNP/CNT and DSA were higher with 52 mV dec.⁻¹ and 60 mV dec.⁻¹ at the range of 40–80 mV, respectively. The Tafel analyses suggest that the CER on Pt₁/CNT proceeded with faster kinetics than on PtNP/CNT and DSA.”

14. Line 205/206: The Heyrovsky step would be rate determining if chlorine is not already adsorbed on the catalyst's surface when CER takes place. However, if the active surface configuration contains adsorbed chlorine, then the Heyrovsky step is the first step in the reaction mechanism. Correspondingly, the Volmer step, which as second step closes the electrocatalytic cycle, becomes rate determining, as long as the Tafel slope is smaller than 60 mV/dec. Therefore, I suppose that the author need to revise their statement.

We are grateful to the reviewer's critical comment, which lead us to further explore the mechanistic pathways for the CER in our study. Following the recent publications by Exner and co-workers [*ChemElectroChem* **4**, 2902-2908 (2017) and *ACS Catal.* **8**, 1864-1879 (2018)], we constructed a full free energy diagram (**Revised Fig. 4d**) to include the discussions regarding the reaction mechanism of CER over the Pt₁/CNT. As the reviewer pointed out, the Heyrovsky and Volmer steps were identified as the first and second step for electrocatalytic cycle over Pt₁/CNT, respectively. Also, the rate-determining step was changed from Volmer to Heyrovsky step with increasing overpotential (Newly added **Supplementary Figs. 34 and 35**). Thus, we thoroughly revised the manuscript and supporting information according to these findings.

In addition, we agree with the reviewer's concern and have removed the phrase regarding the rate determining step. The sentence was written based on the previously reported mechanism for the general Volmer-Heyrovsky mechanism, not specifically for our results.

Revised Fig. 4d | Full free energy diagram along the reaction coordinate of CER over the Pt₁/CNT at respective overpotential for CER (η_{CER}) of 0 V (black thick lines) and 0.09 V (red thick lines). α_1 and α_2 represent the transfer coefficients at each transition state (TS), which are determined as 0.83 and 0.58 from experimental Tafel plots, respectively. The TS with higher free energy at the respective η_{CER} is indicated by purple line. Orange arrows represent the decreased amounts of free energies by applied η_{CER} for each state [i.e., first TS (denoted as '#1'), intermediate state, second TS (denoted as '#2'), and final state, respectively]. The free energy change for reaction intermediate at zero overpotential ($\Delta G_{\text{TD}(\text{CER})}$) is indicated by blue arrow.

New Supplementary Fig. 34 | Experimental Tafel plot exhibiting two linear Tafel regions with Tafel slopes of 38 mV dec⁻¹ and 79 mV dec⁻¹. The fitting ranges of overpotential for CER (η_{CER}) are indicated. The exchange current density j_0 is given by the extrapolation.

New Supplementary Fig. 35 | a Transition state (TS) free energy corresponding to the step #_i ($G_{\#i}(\eta_{\text{CER}})$) as

a function of applied overpotential η_{CER} in the Pt₁/CNT. The first step ($\#_1$) corresponds to the Heyrovsky step, while the second step ($\#_2$) corresponds to the Volmer step, respectively. The black dotted line indicates the thermodynamic optimum of PtN₄C₁₂ species, where the η_{CER} equals to the thermodynamic overpotential for CER (i.e., $\eta_{\text{CER}} = \eta_{\text{TD(CER)}} (= 0.09 \text{ V})$). **b** Absolute value of adsorption free energy for the reaction intermediate ($|\Delta G(\eta_{\text{CER}})|$) of PtN₄C₁₂ species as a function of applied overpotential, η_{CER} . The black dotted line indicates the thermodynamic optimum ($|\Delta G(\eta_{\text{CER}})| = 0$), where the η_{CER} equals to the thermodynamic overpotential for CER (i.e., $\eta_{\text{CER}} = \eta_{\text{TD(CER)}} = 0.09 \text{ V}$).

Changes made:

- We have revised the **Fig. 4d** in page 35 of the revised manuscript.
- We have added a new paragraph on page 14 of the revised manuscript

New paragraph: “By combining the experimental data for kinetics and theoretical data for thermodynamics, we constructed a full free energy diagram along the reaction coordinate of CER over the Pt₁/CNT (**Supplementary Note 6**). In this approach, which was recently developed by Exner and co-workers^{63,64}, the free energies of the transition states (TS’s) are obtained from the experimental Tafel plots, while the free energies of the reaction intermediates are determined from DFT calculations (**Fig. 4d** and **Supplementary Fig. 34**). The resulting full free energy diagram of CER over the Pt₁/CNT revealed that the Heyrovsky step is the first step (denoted as ‘ $\#_1$ ’) in the reaction pathway since the PtN₄C₁₂ species, i.e., active site in the Pt₁/CNT, already involved the Cl* species at acidic CER condition (**Supplementary Fig. 26a**).

Subsequently, the Volmer step followed as the second step (denoted as ‘ $\#_2$ ’) to close the electrocatalytic cycle, which was identified as the rate determining step (RDS) with higher TS free energy at zero overpotential (i.e., $G_{\#_1} = 0.75 \text{ eV}$ for Heyrovsky step, and $G_{\#_2} = 0.80 \text{ eV}$ for Volmer step, respectively, **Fig. 4d**).

However, for $\eta_{\text{CER}} = 0.09 \text{ V}$, which corresponded to the thermodynamic optimum of PtN₄C₁₂ species for the CER (i.e., $\eta_{\text{CER}} = \eta_{\text{TD(CER)}}$), the Heyrovsky step became the RDS with a slightly higher TS free energy (i.e., $G_{\#_1} = 0.67 \text{ eV}$ for Heyrovsky step, and $G_{\#_2} = 0.66 \text{ eV}$ for Volmer step, respectively) (**Fig. 4d** and **Supplementary Fig. 35a**). A recent study highlighted that the thermodynamic measure for the activity would be more helpful if it is evaluated at target overpotential ($\eta > 0 \text{ V}$), not at zero overpotential ($\eta = 0 \text{ V}$)⁶⁵. Considering that typical CER overpotentials for chlor-alkali process are about $\sim 0.1 \text{ V}$ ^{5,66}, PtN₄C₁₂ species in the Pt₁/CNT would be more beneficial for industrial chlorine electrocatalysis since they can reach nearly thermoneutral state at target overpotential (**Supplementary Fig. 35b**).”

- We have added new references on page 29 of the revised manuscript.
63. Exner, K. S., Sohrabnejad-Eskan, I., Anton, J., Jacob, T. & Over, H. Full free energy diagram of an electrocatalytic reaction over a single-crystalline model electrode. *ChemElectroChem* **4**, 2902–2908 (2017).
64. Exner, K. S., Sohrabnejad-Eskan, I. & Over, H. A universal approach to determine the free energy diagram of an electrocatalytic reaction. *ACS Catal.* **8**, 1864–1879 (2018).

65. Exner, K. S. Is thermodynamics a good descriptor for the activity? Re-investigation of Sabatier's principle by the free energy diagram in electrocatalysis. *ACS Catal.* **9**, 5320–5329 (2019).

66. Over, H. Atomic scale insights into electrochemical versus gas phase oxidation of HCl over RuO₂-based catalysts: a comparative review *Electrochim. Acta* **93**, 314–333 (2013).

- We have newly added **Supplementary Fig. 34** on page S38 in the revised supporting information.
- We have newly added **Supplementary Fig. 35** on page S39 in the revised supporting information.
- We have newly added **Supplementary Note 6** regarding the construction of full free energy diagram for CER over Pt₁/CNT (**New Fig. 4d**) on pages S54–55 of the revised supporting information.

New paragraph: “By combining the experimental data for the kinetics and theoretical data for thermodynamics, a full free energy diagram along the reaction coordinate of CER over the Pt₁/CNT was constructed. Details regarding the definition and derivation of this approach are fully given in the earlier works by Exner and co-workers^{22,23}. Within the Butler-Volmer formalism, the Tafel slope, b , is defined by following equation,

$$b = \frac{k_B T \ln(10)}{e \cdot (\gamma + r_{\text{rds}} \alpha_k)} \quad (52)$$

where k_B is Boltzmann's constant, T is temperature, e is elementary charge, γ is integer number of transferred electrons before the rate-determining step (rds), r_{rds} is 0 for chemical (i.e., no charge transfer) and 1 for electrochemical step, and α_k is transfer coefficient of the considered reaction step k , respectively. With increasing overpotential for the CER (η_{CER}), the experimental Tafel plot reveals two linear Tafel regions with b of 38 mV dec.⁻¹ ($30 \text{ mV} \leq \eta_{\text{CER}} \leq 70 \text{ mV}$), and 79 mV dec.⁻¹ ($75 \text{ mV} \leq \eta_{\text{CER}} \leq 102 \text{ mV}$), respectively (**Supplementary Fig. 34**). At room temperature, the respective γ , r_{rds} , and α_k for each Tafel region were determined as $\gamma = 0$, $r_{\text{rds}} = 1$, $\alpha_1 = 0.83$ (for first region, $k = 1$) and $\gamma = 1$, $r_{\text{rds}} = 1$, $\alpha_2 = 0.58$ (for second region, $k = 2$), respectively. The overall current density (j) can be expressed as a function of η_{CER} ,

$$\log(j(\eta_{\text{CER}})) = \log\left(\frac{k_B T \cdot 2e \Gamma_{\text{act}}}{h}\right) - \frac{G_{\text{rds}}^{\#}}{k_B T \ln(10)} + \frac{\eta_{\text{CER}}}{b} = \log(j_0) + \frac{\eta_{\text{CER}}}{b} \quad (53)$$

where h is the Plank's constant, Γ_{act} is the number of active sites per area, $G_{\text{rds}}^{\#}$ is the free energy of transition state (TS) at the rds, and j_0 is the exchange current density, respectively. The Γ_{act} was obtained from the equation (54):

$$\Gamma_{\text{act}} = m \times N_A \quad (54)$$

where m is the molar number of Pt-catalyst loaded on the electrode (i.e., 14.00 nmol cm⁻² for Pt₁/CNT), and N_A is Avogadro's number (6.022×10^{23}). From the $\log(j_0)$, we can determine the $G_{\text{rds}}^{\#}$ as follows,

$$G_{rds}^{\#} = k_B T \ln(10) \left(\log \left(\frac{k_B T \cdot 2e\Gamma_{act}}{h} \right) - \log(j_0) \right) \quad (55)$$

The $G_{rds}^{\#}$'s were determined as 0.75 (for first TS) and 0.80 eV (for second TS) in the Pt₁/CNT. When the η_{CER} is applied ($\eta_{CER} > 0$), the free energies of each state are affected by the number of transferred electrons, z (i.e., $z = 0, 1,$ and 2 for initial state (IS), intermediate state (IM), and final state (FS)) and α_k (i.e., $\alpha_1 = 0.83$ for first TS, and $(1+\alpha_2) = 1.58$ for second TS, respectively). Resultingly, with increasing η_{CER} , the free energies for the first TS, IM, second TS, and FS along the reaction coordinate of CER over Pt₁/CNT were lowered by $0.83 \cdot e \cdot \eta_{CER}$, $1 \cdot e \cdot \eta_{CER}$, $1.58 \cdot e \cdot \eta_{CER}$, and $2 \cdot e \cdot \eta_{CER}$, respectively (**Fig. 4d**).

The TS free energies of step $\#_i$ ($G_{\#_i}(\eta_{CER})$) showed that with increasing η_{CER} , the rds was switched from the first TS (Heyrovsky step, $i = 1$) to the second TS (Volmer step, $i = 2$) due to the larger decrease of free energies by η_{CER} in the Volmer step (indicated by the slope, $dG_{\#_i}(\eta_{CER})/d\eta_{CER}^{-1}$, in the **Supplementary Fig. 35a**). Additionally, the absolute value of free energy for the IM ($|\Delta G(\eta_{CER})|$) showed that it reached the thermoneutral state at the point where the η_{CER} equals to the thermodynamic overpotential for CER (i.e., $\eta_{TD(CER)} = 0.09$ V for PtN₄C₁₂ species, **Supplementary Fig. 35b**).

- We have added new supplementary references on page S57 of the revised supporting information.
 22. Exner, K. S., Sohrabnejad-Eskan, I., Anton, J., Jacob, T. & Over, H. Full free energy diagram of an electrocatalytic reaction over a single-crystalline model electrode. *ChemElectroChem* **4**, 2902–2908 (2017).
 23. Exner, K. S., Sohrabnejad-Eskan, I. & Over, H. A Universal approach to determine the free energy diagram of an electrocatalytic reaction. *ACS Catal.* **8**, 1864–1879 (2018).
15. Line 223/224: Pourbaix diagrams rely on a constrained thermodynamics approach, that is, the actual reaction (CER/OER) is suppressed, but the adsorption of CER/OER reactants on the catalyst's surface is allowed to proceed. I advise the authors to refer to the state-of-the-art literature in the field of Pourbaix diagrams for the CER (e.g., 10.1021/acscatal.8b01432, 10.1002/cssc.201900298) and to add an explanatory sentence, why Pourbaix diagrams are a valuable tool to resolve the surface structure of an electrocatalyst under reaction conditions.

We thank the reviewer for the valuable comment. As remarked by the reviewer, Pourbaix diagrams are constructed under a constrained condition, at which actual reaction is suppressed. From our understanding based on the referred literature [*ACS Catal.* **8**, 9034-9042 (2018) and *ChemSusChem* **12**, 2330-2344 (2019)], Pourbaix diagrams can provide the most plausible structure for active surface configurations at the respective condition (i.e., electrode potential and pH). Thus, the identified active surface structures can promisingly serve as a starting point for the mechanistic investigations. Following the reviewer's comment, we have added the explanatory sentences and relevant references to strengthen the meaning of Pourbaix diagram.

Changes made:

- We have revised sentence on pages 12 of the revised manuscript:
Original sentence: “We constructed Pourbaix diagrams, which provide the thermodynamically most stable adsorbate structures under varying potential (U) and pH, by calculating the adsorption free energies (ΔG 's) of possible adsorbates (i.e., ClO^* , Cl^* , H^* , OOH^* , O^* , and OH^*) for Pt–N₄ sites (Supplementary Figs. 19–21 and Supplementary Note 2).
Revised sentence: “We constructed Pourbaix diagrams, which provide the thermodynamically most stable adsorbate structures under applied electrode potential (U) and pH, by calculating the adsorption free energies (ΔG 's) of possible adsorbates (i.e., * , ClO^* , Cl^* , H^* , OOH^* , O^* , and OH^*) for Pt–N₄ sites (**Supplementary Figs. 24–27 and Supplementary Note 3**).”
 - We have added new sentences on pages 12 of the revised manuscript:
New sentences: “The Pourbaix diagram was constructed from an *ab-initio* constrained thermodynamics approach; the reactants are adsorbed on surface while the subsequent formation of product does not proceed⁵⁹. Nevertheless, the active adsorbate structure, which is identified as the thermodynamically most stable one, can promisingly serve as a starting point for the investigation of mechanistic pathways.”
 - We added the following new reference on page 30 of the revised manuscript.
59. Exner, K. A. Recent advancements towards closing the gap between electrocatalysis and battery science communities: the computational lithium electrode and activity-stability Volcano plots. *ChemSusChem* **12**, 2330–2344 (2019).
 - We have revised a sentence on page 13 of the revised manuscript:
Original sentence: “The Pourbaix diagrams for PtO₂ (110) surface revealed that both 2O_b2Cl_c and 2O_b2ClO_c can be possible intermediates, especially at the acidic CER operating condition, i.e., $U_{\text{SHE}} \approx 1.36$ V, pH \cong 1 (Supplementary Fig. 24).”
Revised sentence: “The Pourbaix diagrams for PtO₂ (110) surface revealed that both 2O_b2Cl_c and 2O_b2ClO_c can be possible active adsorbate structures, especially at the acidic CER operating condition (i.e., $U_{\text{SHE}} \approx 1.36$ V, pH \cong 1), where they had similar thermodynamic stability by sharing a phase boundary line (**Supplementary Fig. 31**).”
16. Line 226/227: CER is pH-independent on the SHE scale, but not on the RHE scale. The opposite case holds true for the OER. The authors are mixing the potential reference scales throughout the manuscript. I advise the authors to refer all electrode potentials either to the SHE or RHE scale throughout the entire manuscript.

We thank the reviewer’s comment regarding the potential reference scales. As we responded to the Comment 9, we experimentally calibrated all values of experimental potential in the RHE scale before all electrochemical experiments (see **Methods**). We restrictively used the SHE scale in the DFT calculation section (“Active site identification by DFT calculations” in the revised manuscript) because the Pourbaix diagrams were constructed based on the SHE scale, which has been typically used in most theoretical studies. To avoid confusion, we specified the SHE scale in the DFT section of the revised manuscript and supporting information.

Changes made:

- We have revised sentence on page 12 of the revised manuscript:
Original sentence: “The CER is pH-independent and thus appears as a horizontal line at the equilibrium potential of 1.36 V (**Supplementary Fig. 21**).”
Revised sentence: “The CER is pH-independent on the standard hydrogen electrode (SHE) scale and thus appears as a horizontal line at the equilibrium potential of 1.36 V vs. SHE (**Supplementary Figs. 26**).”
- We have revised a phrase on page 12, line 271 of the revised manuscript:
Original phrase: “potential of standard hydrogen electrode”
Revised phrase: “theoretical SHE potential”
- We have revised a phrase on page 12, line 277 of the revised manuscript:
Original phrase: “in an oxidised form near 1.36 V”
Revised phrase: “in an oxidised form near 1.36 V vs. SHE⁶¹”
- We have revised a phrase on page S52 of the revised supporting information:
Original phrase: “(i.e., 1.23 V)”
Revised phrase: “(i.e., 1.23 V vs. SHE)”

17. Line 230-236: Relating to Supplementary Figure 20, Cl^* corresponds to the active surface configuration (cf. point xiii).

We thank the reviewer’s careful comment regarding the active surface configurations. We agree with that Cl^* corresponds to the active adsorbate structures for all Pt–N₄ sites since they were identified as the thermodynamically most stable phase in the Pourbaix diagram (**Revised Supplementary Fig. 26**). Based on this active adsorbate structure for PtN₄C₁₂ site, we have added related discussion by in-depth investigations on the mechanistic pathways, as referred in our response to the Comment 14 (i.e., point xiii by reviewer). Also, to clarify the meaning of active surface configurations, we have revised a few phrases to refer the “active adsorbate structures”, which were identified from the Pourbaix diagrams, in the revised manuscript.

Revised Supplementary Fig. 26 | Pourbaix diagram of theoretical standard hydrogen electrode potential (U_{SHE}) vs. pH for three Pt- N_4 sites in equilibrium with H^+ , Cl^- and H_2O at $T = 298$ K. a PtN_4C_{12} , c PtN_4C_{10} , and e $PtN_{2+2}C_{4+4}$. Red dashed line and blue dashed line represent the equilibrium potential for CER ($U_{\text{eq}} = 1.36$ V) and for OER ($U_{\text{eq}} = 1.23$ V -0.059 pH) in the SHE scale, respectively. Black solid lines represent the phase boundary where two adsorbate species exist in equilibrium.

Revised Fig. 4c | Free energy diagrams for CER over Pt- N_4 clusters and PtO_2 (110) surface at zero overpotential ($\eta_{\text{CER}} = 0$ V).

Changes made:

- We have revised the **Supplementary Fig. 26** in page S30 of the revised supplementary information.

- We have revised following sentences on page 13 of the revised manuscript:

Original sentence: “The Pourbaix diagrams for PtO₂ (110) surface revealed that both 2O_b2Cl_c and 2O_b2ClO_c can be possible intermediates, especially at the acidic CER operating condition, (i.e., $U_{SHE} \approx 1.36$ V, pH ≈ 1), where they had similar thermodynamic stability by sharing a phase boundary line (Supplementary Fig. 24)”

Revised sentence: “The Pourbaix diagrams for PtO₂ (110) surface revealed that both 2O_b2Cl_c and 2O_b2ClO_c can be possible active adsorbate structures, especially at the acidic CER operating condition, (i.e., $U_{SHE} \approx 1.36$ V, pH ≈ 1), where they had similar thermodynamic stability by sharing a phase boundary line (**Supplementary Fig. 31**)”

Original sentence: “Subsequently, to theoretically evaluate the CER activity with the above-identified reaction intermediates for Pt–N₄ sites and PtO₂ (110) surfaces, we calculated the free energy diagrams for mechanistic pathways of CER (Fig. 4c).”

Revised sentence: “Subsequently, to theoretically evaluate the CER activity with the above-identified active adsorbate structures for Pt–N₄ sites and PtO₂ (110) surfaces, we calculated the free energy diagrams for CER (**Fig. 4c**).”

18. Line 245: Figure 4c: I advise the authors to refer to the state-of-the-art literature concerning the construction of free-energy diagrams: Pourbaix diagrams in the initial step are applied to identify the active surface configuration (Cl* as thermodynamically most stable state), from which the reaction mechanism starts. Intermediate states in the free-energy diagram at $\eta = 0$ V can be uphill only, but never downhill.

We are grateful to the reviewer’s suggestion regarding the construction of free energy diagram. According to the comment, we changed the initial steps as active adsorbate structures (i.e., Cl* species for all Pt–N₄ sites (i.e., PtN₄C₁₂, PtN₄C₁₀, and PtN₂₊₂C₄₊₄), and both 2O_b2Cl_c (including Cl* species) and 2O_b2ClO_c (including ClO* species) for PtO₂ (110) surface, respectively). Correspondingly, the intermediate states in the revised free energy diagram at zero overpotential were uphill for all Pt–N₄ sites and PtO₂ (110) surface (**Revised Fig. 4c**).

Revised Fig. 4c | Free energy diagrams for CER over Pt–N₄ clusters and PtO₂ (110) surface at zero overpotential ($\eta_{CER} = 0$ V).

Changes made:

- We have revised the **Fig. 4c** in page 35 of the revised manuscript.
19. Line 246-249: I suppose that ΔG_{Cl^*} is also referred to a reference state (such as the surface without an adsorbate, i.e., *). Please revise. The theoretical/thermodynamic overpotential, corresponding to the concept of Norskov and co-workers, is not a free-energy change, but a free-energy change (@ $\eta = 0$ V) divided by the elementary charge. This is not backed up in the main text, where the introduction of the theoretical overpotential is confusing. It would be helpful for the reader to cite the corresponding paper of Norskov and co-workers (i.e., 10.1021/jp047349j), where the concept of η_{TD} was introduced. Please name the theoretical overpotential η_{TD} and not η_{CER} to avoid confusion: η_{TD} is not related to any experimental CER overpotential.

We thank the reviewer's constructive comments regarding the insufficient descriptions in the original manuscript. Below, we responded to the comments in a point-by-point manner.

First, we fully agree with the reviewer's point that the bare structure without an adsorbate (*) can also be a reference state for ΔG_{Cl^*} . At this point, we noticed that the data for bare structure (*) was mistakenly omitted in the original version of adsorption free energy diagrams and Pourbaix diagrams for Pt-N₄ sites (Original Supplementary Figs. 20 and 21). Thus, we have added the bare structure without an adsorbate (*) into the Pourbaix diagram. Notably, the revised Pourbaix diagrams also revealed that the Cl* species still corresponded to the active adsorbate structures for all Pt-N₄ sites at acidic CER condition (**Revised Supplementary Figs. 25 and 26**). Therefore, the addition of the state (*) does not change our previous descriptions for the results.

Second, following the reviewer's suggestion, we referred the DFT-derived overpotential as the "thermodynamic overpotential" (η_{TD}) to distinguish from the experimental overpotential, η , throughout the revised manuscript and supporting information. Also, we have clarified the definition of the η_{TD} (i.e., $\eta_{TD} = \frac{\Delta G}{e}$) and added relevant reference in the revised manuscript.

Revised Supplementary Fig. 25 | The adsorption free energy for plausible adsorbates on three Pt-N₄ sites. Seven plausible adsorbates (i.e., bare (*), Cl⁻, O⁻, ClO⁻, OH⁻, H⁺, and OOH⁻) on Pt-N₄ sites were considered as a function of the theoretical standard hydrogen electrode potential (U_{SHE}) at pH = 0: a PtN₄C₁₂, b PtN₄C₁₀, and c PtN₂₊₂C₄₊₄.

Revised Supplementary Fig. 26 | Pourbaix diagram of theoretical standard hydrogen electrode potential (U_{SHE}) vs. pH for three Pt-N₄ sites in equilibrium with H⁺, Cl⁻ and H₂O at $T = 298$ K. a PtN₄C₁₂, c PtN₄C₁₀, and e PtN₂₊₂C₄₊₄. Red dashed line and blue dashed line represent the equilibrium potential for CER ($U_{\text{eq}} = 1.36$ V) and for OER ($U_{\text{eq}} = 1.23$ V - 0.059 pH) in the SHE scale, respectively. Black solid lines represent the phase boundary where two adsorbate species exist in equilibrium.

Changes made:

- We added the bare surface without an adsorbate (*) on page 12, line 262 in the revised manuscript.

Original phrase: (i.e., ClO^{*}, Cl^{*}, H^{*}, OOH^{*}, O^{*}, and OH^{*})

Revised phrase: (i.e., *, ClO^{*}, Cl^{*}, H^{*}, OOH^{*}, O^{*}, and OH^{*})

- We have revised **Supplementary Fig. 25** to include the data for bare structure (*) on page S29 of the revised supporting information.
- We have revised **Supplementary Fig. 26** to include the data for bare structure (*) on page S30 of the revised supporting information.
- We deleted the irrelevant sentence on page 12 of the revised manuscript to reflect **Revised Supplementary Fig. 26**.

Deleted sentence: “Overall, for all Pt–N₄ sites, the Cl* species were identified as the most stable at low and moderate potentials ($U_{\text{SHE}} \leq \sim 1.5$ V) over the wide range of pH ($0 \leq \text{pH} \leq 14$).”

- We have revised the following sentences on pages 13 of the revised manuscript:

Original sentences: “The theoretical overpotential for CER (η_{CER}) can be defined from the ΔG depending on the reaction intermediates as follows: ΔG_{Cl^*} for Cl* species, and $\Delta G_{\text{ClO}^*} - \Delta G_{\text{O}^*}$ for ClO* species, respectively (Supplementary Note 3). Among Pt–N₄ sites, PtN₄C₁₂ was identified as the most plausible structure for CER due to their thermo-neutral state at 1.36 V with $\eta_{\text{CER}} = 0.09$ V.”

Revised sentences: “The thermodynamic overpotential for CER can be defined from the ΔG ’s divided by the elementary charge at zero overpotential (i.e., $\eta_{\text{TD(CER)}} = \frac{\Delta G}{e}$), which depends on the reaction intermediates as follows: ΔG_{Cl^*} for bare structure (*) and Cl* species, and $\Delta G_{\text{ClO}^*} - \Delta G_{\text{O}^*}$ for O* and ClO* species, respectively (Supplementary Note 4)⁶². Among Pt–N₄ sites, PtN₄C₁₂ was identified as the most plausible structure for CER due to its lowest $\eta_{\text{TD(CER)}}$ of 0.09 V at zero overpotential.”
- We have revised the following phrases on page 13, lines 306 and 307 in the revised manuscript:

Original phrase: “ $\eta_{\text{CER}} = 0.20$ V for 2O_b2ClO_c”

Revised phrase: “ $\eta_{\text{TD(CER)}} = 0.20$ V for 2O_b2ClO_c”

Original phrase: “ $\eta_{\text{CER}} = 0.62$ V for 2O_b2Cl_c”

Revised phrase: “ $\eta_{\text{TD(CER)}} = 0.62$ V for 2O_b2Cl_c”
- We have revised the following phrases on page 15, lines 342 and 345 in the revised manuscript:

Original phrase: “with large theoretical overpotentials for OER (i.e., $\eta_{\text{OER}} \geq 0.87$ V, ...)”

Revised phrase: “with a large $\eta_{\text{TD(OER)}}$ above 0.74 V”

Original phrase: “with very high overpotential of 0.75 V”

Revised phrase: “with very high $\eta_{\text{TD(OER)}}$ of 0.62 V,”
- We have revised the following sentences on page S51 of the revised supporting information:

Original sentences: “Then, the theoretical overpotential for CER (η_{CER}) can be defined as follows,

(I) For Cl* intermediate:

$$\eta_{\text{CER}} = \frac{|\Delta G_{\text{Cl}^*}|}{e} \quad (9)$$

(II) For ClO* intermediate:

$$\eta_{\text{CER}} = \frac{|\Delta G_{\text{ClO}^*} - \Delta G_{\text{O}^*}|}{e} \quad (10)$$

To investigate the theoretical overpotential for OER (η_{OER}) of Pt–N₄ sites, we assumed the conventional four-electron pathway for the OER as follows,”

Revised sentences: “Then, the thermodynamic overpotential for CER at zero overpotential ($\eta_{\text{TD}(\text{CER})}$) can be defined as follows,

(I) For * and Cl* species:

$$\eta_{\text{TD}(\text{CER})} = \frac{|\Delta G_{\text{Cl}^*}|}{e} \quad (11)$$

(II) For O* and ClO* species:

$$\eta_{\text{TD}(\text{CER})} = \frac{|\Delta G_{\text{ClO}^*} - \Delta G_{\text{O}^*}|}{e} \quad (12)$$

To investigate the thermodynamic overpotential for OER at zero overpotential ($\eta_{\text{TD}(\text{OER})}$) of Pt-N₄ sites, we assumed the conventional four-electron pathway for the OER as follows,”

- We have revised the following sentences on page S52 of the revised supporting information:

Original sentences: “Finally, η_{OER} can be defined by

$$\eta_{\text{OER}} = \frac{\max[\Delta G_1, \Delta G_2, \Delta G_3, \Delta G_4]}{e} - U_{\text{eq}} \text{ [V]} \quad (13)$$

where U_{eq} indicates the equilibrium potential for OER (i.e., 1.23 V).”

Revised sentences: “Finally, $\eta_{\text{TD}(\text{OER})}$ can be defined by

$$\eta_{\text{TD}(\text{OER})} = \frac{\max[\Delta G_1, \Delta G_2, \Delta G_3, \Delta G_4]}{e} - U_{\text{eq}} \text{ [V]} \quad (14)$$

where U_{eq} indicates the equilibrium potential for OER (i.e., 1.23 V vs SHE).”

- We added the following new reference on page 29 of the revised manuscript.
62. Nørskov, J. K. *et al.* Origin of the overpotential for oxygen reduction at a fuel-cell cathode. *J. Phys. Chem. B* **108**, 17886-17892 (2004).

20. Line 249: PtN₄C₁₂ is not a thermoneutral catalyst at 1.36 V. For this electrode material, the CER via the thermodynamics of the Volmer-Heyrovsky mechanism is thermoneutral at $\eta = 0.09$ V. Please revise. Actually, there was a recent publication in ACS Catalysis (10.1021/acscatal.9b00732), which suggests that it is beneficial for chlorine electrocatalysis if the thermodynamics of the reaction intermediate is thermoneutral at target overpotential. This finding would be fulfilled for PtN₄C₁₂, since typical CER overpotentials, referring to the chlor-alkali process, are about 100 mV.

We thank for the reviewer’s careful comment regarding the thermoneutral state. We have revised the sentence by removing the phrase, “thermo-neutral state at 1.36 V” in the revised manuscript. Also, we agree to the reviewer’s point that PtN₄C₁₂ would be more beneficial for industrial chlorine electrocatalysis by achieving the nearly thermoneutral state at target overpotential (~ 100 mV). Following the recent publication by Exner [ACS

Catal. **9**, 5320-5329 (2019)], we newly added the **New Supplementary Fig. 35b** representing the nearly thermoneutral state of PtN₄C₁₂ species at target overpotential (~100 mV) in the revised supporting information.

New Supplementary Fig. 35 | b Absolute value of adsorption free energy for the reaction intermediate ($|\Delta G(\eta_{\text{CER}})|$) of PtN₄C₁₂ species as a function of applied overpotential, η_{CER} . The black dotted line indicates the thermodynamic optimum ($|\Delta G(\eta_{\text{CER}})| = 0$), where the η_{CER} equals to the thermodynamic overpotential for CER (i.e., $\eta_{\text{CER}} = \eta_{\text{TD}(\text{CER})} = 0.09$ V).

Changes made:

- We have revised the following sentence on page 13 of the revised manuscript.
Original sentence: “Among Pt–N₄ sites, PtN₄C₁₂ was identified as the most plausible structure for CER due to their thermo-neutral state at 1.36 V with $\eta_{\text{CER}} = 0.09$ V.”
Revised sentence: Among Pt–N₄ sites, PtN₄C₁₂ was identified as the most plausible structure for CER due to its lowest $\eta_{\text{TD}(\text{CER})}$ of 0.09 V at zero overpotential.
- We have added the following sentences on page 14 of the revised manuscript.
New sentences: “A recent study highlighted that the thermodynamic measure for the activity would be more helpful if it is evaluated at target overpotential ($\eta > 0$), not at zero overpotential ($\eta = 0$)⁶⁵. Considering that typical CER overpotentials for chlor-alkali process are about ~0.1 V^{5,66}, PtN₄C₁₂ species in the Pt₁/CNT could be highly beneficial for industrial chlorine electrocatalysis since they can reach nearly thermoneutral state at target overpotential (**Supplementary Fig. 35b**).”

21. Line 258: ClO* has also been identified as reaction intermediate in the CER over other transition-metal oxides, such as RuO₂ or IrO₂ (cf. reference 13, reference 20, 10.1021/acscatal.8b01432, 10.1063/1.5051429).

We appreciated the reviewer’s keen insight regarding ClO* as a reaction intermediate of CER on PtO₂. We have reflected the reviewer’s comment to the sentence and have added related references.

Changes made:

- We have newly added the footnotes of the recommended references to the sentence (**Revised References 14, 20, and 60**).
- We have revised the following sentence on page 13 of the revised manuscript to avoid confusion.

Original sentence: “In the case of PtO₂ (110) surface (**Supplementary Fig. 26**), ClO* species ($\eta_{\text{CER}} = 0.20$ V for 2O_b2ClO_c) were found to be closer to the thermoneutral state ($\Delta G = 0$) than Cl* species ($\eta_{\text{CER}} = 0.62$ V for 2O_b2Cl_c), implying that ClO* species play a major role for the CER in the PtNP/CNT.”

Revised sentence: “In the case of PtO₂ (110) surface (**Supplementary Fig. 33**), ClO* species ($\eta_{\text{TD(CER)}} = 0.20$ V for 2O_b2ClO_c) were found to be closer to the thermoneutral state ($\Delta G = 0$) than Cl* species ($\eta_{\text{TD(CER)}} = 0.62$ V for 2O_b2Cl_c), implying that ClO* species were identified as the reaction intermediate for the CER on the PtO₂ surface of the PtNP/CNT, similar to other precious metal oxides, such as RuO₂ and IrO₂ (refs. ^{14,20,60}).”

22. Line 262: Please revise Figure 4d according to point xvii).

We fully agree to the reviewer’s suggestion that active adsorbate structure should be the first step in the free energy diagram (referred in the **Comment 18**, i.e., point xvii by the reviewer). As a precursor step, we further investigated the Pourbaix diagrams of Pt–N₄ sites in the absence of Cl[–] condition (i.e., in equilibrium with H⁺ and H₂O, newly added **Supplementary Fig. 27**). Correspondingly, we have revised the free energy diagrams of OER for Pt–N₄ sites, where the initial steps are different depending on the Pt–N₄ sites (i.e., bare structure, * for PtN₄C₁₂ and PtN₄C₁₀, and O* species for PtN₂₊₂C₄₊₄, respectively, newly added **Supplementary Fig. 36**).

New Supplementary Fig. 27 | Pourbaix diagram of theoretical standard hydrogen electrode potential (U_{SHE}) vs. pH for three Pt-N₄ sites in equilibrium with H⁺ and H₂O at $T = 298$ K. a PtN₄C₁₂, b PtN₄C₁₀, and c PtN₂₊₂C₄₊₄. Blue dashed line represents the equilibrium potential for OER in the SHE scale (i.e., $U_{\text{eq}} = 1.23 \text{ V} - 0.059 \text{ pH}$). Black solid lines represent the phase boundary where two adsorbate species exist in equilibrium.

New Supplementary Fig. 36 | Free energy diagrams for OER on Pt-N₄ sites at zero overpotential ($\eta_{\text{OER}} = 0 \text{ V}$); a PtN₄C₁₂, PtN₄C₁₀ and b PtN₂₊₂C₄₊₄. The active adsorbate structures for each model were employed as initial steps (i.e., bare structure (*) for PtN₄C₁₂ and PtN₄C₁₀, and O* for PtN₂₊₂C₄₊₄, respectively). The black dotted line represents the thermoneutral state (i.e., $\Delta G = 0$).

Changes made:

- We have newly added the **Supplementary Fig. 27** on page S31 of the revised supporting information.
- We have deleted the **Fig. 4d** in the original manuscript and newly added the **Supplementary Fig. 36** on page S40 of the revised supporting information.

23. Line 266: “CN” has not been defined in the main text.

We appreciate the reviewer’s comment on the use of abbreviation. We denoted “coordination number (CN)” at its first usage in page 5 in the original manuscript (please check **page 6 in the revised manuscript**).

24. The authors compile free-energy diagrams for the CER and OER of the proposed PtN₄C₁₂ electrocatalysts; however, the reason for the high CER selectivity on an atomic scale has not been sufficiently elaborated: a thorough comparison of the energetics for the proposed pathways of CER and OER based on the free-energy landscapes is deeply missing.

We appreciate the reviewer’s constructive suggestion to provide a thorough comparison of the energetics for the CER and OER regarding the origin of high CER selectivity. For this investigation, we rescaled the thermodynamic overpotentials for OER ($\eta_{TD(OER)}$) to be referenced to the same potential for the CER (i.e., $\eta_{CER} = 0$ V) by reducing 0.13 V (i.e., 1.36 – 1.23 V) for all values (**Revised Supplementary Table 5**). From a large difference of η_{TD} for the OER and CER adsorbate (i.e., $\eta_{TD(OER)} - \eta_{TD(CER)} = 0.99$ V) at the potential determining step (PDS), we could further corroborate the excellent CER selectivity of PtN₄C₁₂ site compared to the OER. According to the comment, we have revised and added the sentences to explain the PDS for each Pt–N₄ site, and the comparison of η_{TD} for the CER and OER in the revised manuscript.

Revised Supplementary Table 5 | Adsorption free energies (ΔG_s) of OH*, O*, and OOH* on Pt–N₄ sites (i.e., PtN₄C₁₂, PtN₄C₁₀, and PtN₂₊₂C₄₊₄) and thermodynamic overpotentials for OER ($\eta_{TD(OER)}$) at the overpotential (η_{OER}) of 0 and 0.13 V.

Sites	ΔG_{OH^*} (eV)	ΔG_{O^*} (eV)	ΔG_{OOH^*} (eV)	$\eta_{TD(OER)}$ (V) (@ $\eta_{OER} = 0$ V)	$\eta_{TD(OER)}$ (V) (@ $\eta_{OER} = 0.13$ V)
PtN ₄ C ₁₂	2.45	4.35	5.31	1.22	1.09
PtN ₄ C ₁₀	2.10	4.12	5.07	0.87	0.74
PtN ₂₊₂ C ₄₊₄	0.82	1.97	3.95	0.75	0.62

Changes made:

- We have revised the following **Supplementary Table 5** on page S45 of the revised supporting information.

- We have newly added the following sentence on page 15 of the revised manuscript.
New sentence: “Note that the thermodynamic overpotentials for OER ($\eta_{TD(OER)}$) of all Pt–N₄ sites were evaluated at the overpotential of 0.13 V (i.e., $\eta_{OER} = 1.36 \text{ V} - 1.23 \text{ V} = 0.13 \text{ V}$), to be referenced to the same potential for the CER (i.e., $\eta_{CER} = 0 \text{ V}$).”
- We have revised a following sentence on page 15 of the revised manuscript.
Original sentence: “Apart from the PtN₂₊₂C₄₊₄ model, OH* adsorption from H₂O_(l) was found to be the most endothermic among reaction steps, corresponding to the potential determining step with large theoretical overpotentials for OER (i.e., $\eta_{OER} \geq 0.87 \text{ V}$, **Supplementary Table 4**).”
Revised sentence: “For PtN₄C₁₂ and PtN₄C₁₀ sites, OH* adsorption from H₂O_(l) was found to be the most endothermic among reaction steps, corresponding to the potential determining step (PDS) with a large $\eta_{TD(OER)}$ above 0.74 V (**Supplementary Table 5**)”
- We have revised a following phrase on page 15 of the revised manuscript.
Original phrase: “with very high overpotential of 0.75 V”
Revised phrase: “with very high $\eta_{TD(OER)}$ of 0.62 V, corresponding to the formation of OOH*”
- We have newly added the following sentence on page 15 of the revised manuscript.
New sentence: “Particularly focusing on the PtN₄C₁₂ site, which most plausibly catalyses the CER in the Pt₁/CNT, a huge difference between $\eta_{TD(OER)}$ and $\eta_{TD(CER)}$ at the PDS (i.e., $\eta_{TD(OER)} - \eta_{TD(CER)} = 0.99 \text{ V}$) further corroborates the excellent CER selectivity of Pt₁/CNT compared to the OER.”

REVIEWERS' COMMENTS:

Reviewer #1 (Remarks to the Author):

The manuscript has been revised according to the reviewers' comments. I recommend that the paper could be accepted in the present form.

Reviewer #2 (Remarks to the Author):

The authors have done an admirable job in responding to all reviewer comments but there are still some issues with the stability model. Equation (50) is not charge balanced and addition of electrons would mean potential dependence which is not at all addressed. DFT-based treatment of electrochemical dissolution reactions for such cases have been previously addressed in, e.g., DOI: 10.1103/PhysRevB.85.235438 and doi:10.1038/s41529-019-0088-z. Either this should be properly treated (yielding a dissolution potential for a 2 electron reaction) or the previous use of a Pt bulk reference state should be maintained with some explanation that the choice is to avoid complications with modeling charged systems. Otherwise, the authors have improved on an already very nice manuscript.

Reviewer #3 (Remarks to the Author):

Atomically dispersed Pt-N4 sites as efficient and selective electrocatalysts for the chlorine evolution reaction
T. Lim et al.

The authors made a thorough revision and addressed all 23 points that I was raising when initially reviewing the manuscript. The manuscript is much stronger after the revision, particularly as the authors extended the experimental measurements to elevated temperatures as present in the chlor-alkali process. From my side there are no further obstacles for publication and without any further ado I recommend this contribution to be published in Nature Communications.

I have the following minor remarks that could be considered in the finally accepted version of the manuscript:

- i) The English language is ought to be polished at certain parts in the manuscript; probably, the editors will take care of this.
- ii) Page 8: "The j_0 of Pt1/CNT was 0.43 mA cm^{-2} at the overpotential range of 30–70 mV, whereas those of PtNP/CNT and DSA were 0.23 mA cm^{-2} and 0.20 mA cm^{-2} at the overpotential range of 40–80 mV, respectively." - I am a bit confused, why the authors state overpotential ranges for the exchange current density, j_0 . j_0 is evaluated at zero overpotential by extrapolating the corresponding linear Tafel lines to $\eta_{\text{CER}} = 0 \text{ V}$ (as shown in Fig. 34 of the supplementary). I think the overpotential ranges in this sentence should be deleted to avoid confusion.

Exner

<List of Major Changes>

For Main Text

1. Throughout the manuscript and the supporting information, we checked and revised our writings.
2. Throughout Result and Methods section, we have redacted all subheadings following the guideline.
3. Page 2, we have revised the sentences in the present tense describing the results of the current study in Abstract section.
4. Page 2, we have labelled Abstract section.
5. Page 2, we have revised the sentence describing Cl₂ production.
6. Page 3, we have newly added the sentences according to the RHE scale information, which was previously described as Supplementary Note 1.
7. Page 4, we have revised the final paragraph of Introduction section in the present tense.
8. Page 8, we have redacted the sentence regarding the exchange current density.
9. Page 13, we have revised the sentences regarding the theoretical stability against Pt dissolution for the Pt–N₄ site.
10. Page 21, we have revised the sentence to note that three experiments were independently conducted.
11. Page 25, we have newly added Data availability section.
12. Page 26, we have formatted Reference 4 in Nature style.
13. Page 31, we have revised Competing interests section.
14. Page 33, we have stated what the error bars of Fig. 2d indicate.

For Supporting Information

1. Throughout the supporting information, we have removed the line numbering
2. Page 2, we have removed the Table of contents list.
3. Page 33, we revised Supplementary Fig. 32.
4. Page 43, we have moved Supplementary Note 1 to the Introduction section in manuscript.
5. Page 49, we have revised Supplementary Note 4.
6. Page 53, we have replaced Supplementary Reference 21 and added new Supplementary Reference 22.
7. Page 52–53, we have rearranged the order of references in accordance with the changes.

Responses to the Comments of Reviewer 1

(Reviewer's Comments) The manuscript has been revised according to the reviewers' comments. I recommend that the paper could be accepted in the present form.

We sincerely appreciate the reviewer's advice of our paper.

Responses to the Comments of the Reviewer 2

The authors have done an admirable job in responding to all reviewer comments but there are still some issues with the stability model. Equation (50) is not charge balanced and addition of electrons would mean potential dependence which is not at all addressed. DFT-based treatment of electrochemical dissolution reactions for such cases have been previously addressed in, e.g., DOI: 10.1103/PhysRevB.85.235438 and doi:10.1038/s41529-019-0088-z. Either this should be properly treated (yielding a dissolution potential for a 2 electron reaction) or the previous use of a Pt bulk reference state should be maintained with some explanation that the choice is to avoid complications with modeling charged systems. Otherwise, the authors have improved on an already very nice manuscript.

We appreciate the reviewer's valuable comment. We agree with the reviewer's point on that our previous energetic comparison, based on the previous equations (48) and (49) in the original manuscript, did not fully describe the dependence of potential originated from the charged model systems. To reflect the concept of Pt dissolution, we further investigated the chemical potential of Pt atom on the outermost layer of Pt(111) surface (i.e., $\mu_{\text{Pt}(111)}$). Subsequently, following the previous approach regarding the Pt dissolution [*J. Phys. Chem. C* **117**, 7107-7113 (2013) and *Phys. Chem. Chem. Phys.* **18**, 14234-14243 (2016)], we investigated the electrode potential shift (ΔU) for reaction of $\text{Pt} \rightarrow \text{Pt}^{2+} + 2\text{e}^-$, which was gauged with the difference in chemical potential of Pt atom in the Pt-N₄ sites and that on the Pt(111) surface [i.e., $\mu_{\text{Pt-N}_4} - \mu_{\text{Pt}(111)}$]. The positive value of ΔU indicates that the dissolution of Pt from the Pt-N₄ sites would occur at a higher electrode potential compared to that from Pt(111) surface. We note that the equations for $\mu_{\text{Pt-N}_4}$ and $\mu_{\text{Pt}(111)}$ are charge balanced with reference to the Pt bulk state. As a result, PtN₄C₁₂ species required the largest energetic cost for Pt dissolution among Pt-N₄ sites (i.e., $\Delta U = 1.59$ V, **Revised Supplementary Fig. 32**), implying that PtN₄C₁₂ species were still thermodynamically most stable against Pt dissolution than other ones.

Revised Supplementary Fig. 32 | **a** Electrode potential shift (ΔU), which was gauged with the difference in chemical potential of Pt atom in the Pt-N₄ sites and that on the Pt(111) surface [i.e., $\mu_{\text{Pt-N}_4} - \mu_{\text{Pt}(111)}$]. The black dotted line represents the dissolution potential of Pt(111) surface. **b** Schematics for leaching of a Pt atom on the Pt(111) surface, used for $\mu_{\text{Pt}(111)}$ calculation. Colour legends – Pt(top-most): dark-blue; Pt(second-most from top): black; Pt(third-most from top): grey; Pt(bottom-most): light-grey. The red dotted circle indicates the leaching site on the outermost layer.

Changes made:

- We have revised the title of **Supplementary Note 4** to clarify the meaning in the revised supporting information:

Original title: “Stability calculation of Pt–N₄ sites.”

Revised title: “Stability against Pt dissolution for the Pt–N₄ sites.”

- We have extensively revised the **Supplementary Note 4** in the revised supporting information:

Original sentences: “The stability of Pt–N₄ sites against Pt dissolution was theoretically evaluated (**Supplementary Fig. 32**). It was previously reported that Pt dissolution in the presence of Cl[−] proceeds through the formation of metastable Pt-chloro complex such as [PtCl₄]^{2−} (ref. ²¹). To reflect this situation, we investigated the relative stability of Pt–N₄ sites by comparing the energy difference between the adsorption energy of Pt²⁺ ion in the Pt–N₄ sites (ΔE_{ads}) and the chlorination energy of Pt²⁺ ion in the [PtCl₄]^{2−} complex (ΔE_{Cl^-}). The ΔE_{ads} and ΔE_{Cl^-} were calculated by,

$$\Delta E_{\text{ads}} = E_{\text{Pt-N}_4} - E_{\text{Pt-free}} - E_{\text{Pt}^{2+}} \quad (50)$$

$$\Delta E_{\text{Cl}^-} = E_{[\text{PtCl}_4]^{2-}} - E_{\text{Pt}^{2+}} - 4 E_{\text{Cl}^-} \quad (51)$$

where $E_{\text{Pt-N}_4}$, $E_{\text{Pt-free}}$, $E_{\text{Pt}^{2+}}$, $E_{[\text{PtCl}_4]^{2-}}$, and E_{Cl^-} represent the DFT-optimised energies of Pt–N₄ sites, bare Pt–N₄ sites without Pt²⁺ ion in a reduced state, a solvated Pt²⁺ ion in water, [PtCl₄]^{2−} complex, and a solvated Cl[−] ion in water, respectively.”

Revised sentences: “The extent of dissolution of Pt from Pt–N₄ sites relative to that from Pt(111) surface (**Supplementary Fig. 32**) could be gauged with the electrode potential shift (ΔU) for reaction $\text{Pt} \rightarrow \text{Pt}^{2+} + 2\text{e}^-$, by following the previous reports^{21,22}. The ΔU is defined by,

$$\Delta U = -[\mu_{\text{Pt-N}_4} - \mu_{\text{Pt(111)}}]/2e \quad (50)$$

where $\mu_{\text{Pt-N}_4}$ and $\mu_{\text{Pt(111)}}$ are the chemical potentials of Pt atoms on the Pt–N₄ sites and Pt(111) surface, respectively. Note that the dissolutions of Pt atoms are assumed to only occur on the outermost layer of the surface. Assuming the reference state as Pt bulk state, the $\mu_{\text{Pt-N}_4}$ and $\mu_{\text{Pt(111)}}$ were calculated as follows,

$$\mu_{\text{Pt-N}_4} = E_{\text{Pt-N}_4} - E_{\text{Pt-free}} - E_{\text{Pt-bulk}} \quad (51)$$

$$\mu_{\text{Pt(111)}} = E_{\text{Pt(111)}} - E_{\text{Pt-leached}} - E_{\text{Pt-bulk}} \quad (52)$$

where $E_{\text{Pt-N}_4}$, $E_{\text{Pt-free}}$, $E_{\text{Pt-bulk}}$, $E_{\text{Pt(111)}}$, and $E_{\text{Pt-leached}}$ represent the DFT-optimised energies of Pt–N₄ sites, Pt–N₄ sites where the Pt atom is not included, a Pt atom in the bulk unit cell, perfect Pt(111) surface, and Pt(111) surface where a single Pt atom on the outermost surface is leached, respectively.

- We have revised the following sentences on page 13 of the revised manuscript to reflect **Revised Supplementary Fig. 32**:

Original sentences: “Moreover, we evaluated the thermodynamic stability of the Pt–N₄ sites by calculating the difference in adsorption energy (ΔE_{ads}) and chlorination energy

(ΔE_{Cl^-}) of Pt^{2+} ion (**Supplementary Fig. 32**). The negative value of $\Delta E_{\text{ads}} - \Delta E_{\text{Cl}^-}$ implies that embedding Pt^{2+} ion into the adjacent network is more stable against Pt dissolution, which may proceed via the formation of Pt-chloro complex (**Supplementary Note 4**). Thus, besides good CER activity, $\text{PtN}_4\text{C}_{12}$ species were found to be thermodynamically more stable than other ones, which further supports that $\text{PtN}_4\text{C}_{12}$ species are feasible to exist in Pt_1/CNT on a theoretical viewpoint.”

Revised sentences: “Moreover, the stability against Pt dissolution for the Pt-N_4 sites was evaluated based on the electrode potential shift (ΔU), which was gauged with reference to the dissolution on the $\text{Pt}(111)$ surface (**Supplementary Fig. 32**). The positive value of ΔU implies that the dissolution of Pt on the Pt-N_4 sites requires more energetic cost compared to that from $\text{Pt}(111)$ surface (**Supplementary Note 4**). Thus, besides good CER activity, $\text{PtN}_4\text{C}_{12}$ species were found to be thermodynamically more stable against Pt dissolution than others, which further supports that it is feasible for $\text{PtN}_4\text{C}_{12}$ species to exist in Pt_1/CNT theoretically.”

Responses to the Comments of the Reviewer 3

Atomically dispersed Pt-N4 sites as efficient and selective electrocatalysts for the chlorine evolution reaction

T. Lim et al.

The authors made a thorough revision and addressed all 23 points that I was raising when initially reviewing the manuscript. The manuscript is much stronger after the revision, particularly as the authors extended the experimental measurements to elevated temperatures as present in the chlor-alkali process. From my side there are no further obstacles for publication and without any further ado I recommend this contribution to be published in Nature Communications.

We sincerely appreciate the reviewer's advice of our paper.

1. The English language is ought to be polished at certain parts in the manuscript; probably, the editors will take care of this.

We appreciate the reviewer's comment to refine the English language throughout the manuscript. We have edited the manuscript and the supporting information thoroughly.

2. Page 8: "The j_0 of Pt₁/CNT was 0.43 mA cm⁻² at the overpotential range of 30–70 mV, whereas those of PtNP/CNT and DSA were 0.23 mA cm⁻² and 0.20 mA cm⁻² at the overpotential range of 40–80 mV, respectively." - I am a bit confused, why the authors state overpotential ranges for the exchange current density, j_0 . j_0 is evaluated at zero overpotential by extrapolating the corresponding linear Tafel lines to $\eta_{\text{CER}} = 0$ V (as shown in Fig. 34 of the supplementary). I think the overpotential ranges in this sentence should be deleted to avoid confusion.

We are grateful the reviewer's recommendation to avoid confusion regarding the exchange current. We have removed the potential range in which the exchange current was experimentally obtained.

Changes made:

- We have redacted the following sentence on page 8 of the revised manuscript.
Original sentence: "The j_0 of Pt₁/CNT was 0.43 mA cm⁻² at the overpotential range of 30–70 mV, whereas those of PtNP/CNT and DSA were 0.23 mA cm⁻² and 0.20 mA cm⁻² at the overpotential range of 40–80 mV, respectively."
Revised sentence: "The j_0 of Pt₁/CNT was 0.43 mA cm⁻², whereas those of PtNP/CNT and DSA were 0.23 mA cm⁻² and 0.20 mA cm⁻², respectively."